# Kronecker-Factored Approximate Curvature for Physics-Informed Neural Networks

**Felix Dangel**[*]
Vector Institute
Toronto
Canada
fdangel@vectorinstitute.ai

**Johannes Müller**[*]
Chair of Mathematics of Information Processing
RWTH Aachen University
Aachen, Germany
mueller@mathc.rwth-aachen.de

**Marius Zeinhofer**[*]
Seminar for Applied Mathematics, ETH Zürich,
Department of Nuclear Medicine, University Hospital Freiburg
marius.zeinhofer@uniklinik-freiburg.de

## Abstract

Physics-informed neural networks (PINNs) are infamous for being hard to train. Recently, second-order methods based on natural gradient and Gauss-Newton methods have shown promising performance, improving the accuracy achieved by first-order methods by several orders of magnitude. While promising, the proposed methods only scale to networks with a few thousand parameters due to the high computational cost to evaluate, store, and invert the curvature matrix. We propose Kronecker-factored approximate curvature (KFAC) for PINN losses that greatly reduces the computational cost and allows scaling to much larger networks. Our approach goes beyond the established KFAC for traditional deep learning problems as it captures contributions from a PDE's differential operator that are crucial for optimization. To establish KFAC for such losses, we use Taylor-mode automatic differentiation to describe the differential operator's computation graph as a forward network with shared weights. This allows us to apply KFAC thanks to a recently developed general formulation for networks with weight sharing. Empirically, we find that our KFAC-based optimizers are competitive with expensive second-order methods on small problems, scale more favorably to higher-dimensional neural networks and PDEs, and consistently outperform first-order methods and LBFGS.

## 1 Introduction

Neural network-based approaches to numerically solve partial differential equations (PDEs) are growing at an unprecedented speed. The idea to train network parameters to minimize the residual of a PDE traces back to at least Dissanayake & Phan-Thien [15], Lagaris et al. [28], but was only recently popularized under the name *deep Galerkin method* (DGM) and *Physics-informed neural networks* (PINNs) through the works of Sirignano & Spiliopoulos [52], Raissi et al. [50]. PINNs are arguably one of the most popular network-based approaches to the numerical solution of PDEs as they are easy to implement, seamlessly incorporate measurement data, and promise to work well in high dimensions. Despite their immense popularity, PINNs are notoriously difficult to optimize [57] and fail to provide satisfactory accuracy when trained with first-order methods, even for simple problems [64, 41]. Recently, second-order methods that use the function space geometry to design preconditioners

---

[*]Equal contribution

38th Conference on Neural Information Processing Systems (NeurIPS 2024).

have shown remarkable promise in addressing the training difficulties of PINNs [64, 41, 14, 24, 42]. However, these methods require solving a linear system in the network's high-dimensional parameter space at cubic computational iteration cost, which prohibits scaling such approaches. To address this, we build on the idea of Kronecker-factored approximate curvature (KFAC) and apply it to Gauss-Newton matrices of PINN losses which greatly reduces the computational cost:

- We use higher-order forward (Taylor) mode automatic differentiation to interpret the computation graph of a network's input derivatives as a larger net with weight sharing (§3.1).

- We use this weight sharing view to propose KFAC for Gauss-Newton matrices of objectives with differential operators, like PINN losses (§3.3 and eq. (14)). Thanks to the generality of Taylor-mode and KFAC for weight sharing layers [17], our approach is widely applicable.

- We show that, for specific differential operators, the weight sharing in Taylor-mode can be further reduced by absorbing the reduction of partial derivatives into the forward propagation, producing a more efficient scheme. For the prominent example of the Laplace operator, this recovers and generalizes the *forward Laplacian* framework [29] (§3.2 and eq. (9)).

- Empirically, we find that our KFAC-based optimizers are competitive with expensive second-order methods on small problems, scale more favorably to higher-dimensional neural networks and PDEs, and consistently outperform first-order methods and LBFGS (§4).

**Related work**  Various approaches were developed to improve the optimization of PINNs such as adaptive re-weighting of loss terms [57, 56, 59], different sampling strategies for discretizing the loss [34, 43, 13, 63, 58, 61], and curriculum learning [26, 58]. While LBFGS is known to improve upon first-order optimizers [35], recently, other second-order methods that design meaningful preconditioners that respect the problem's geometry have significantly outperformed it [64, 41, 14, 33, 24, 7, 62]. Müller & Zeinhofer [42] provide a unified view on these approaches which greatly improve the accuracy of PINNs, but come with a significant per-iteration cost as one needs to solve a linear system in the network's high-dimensional parameter space, which is only feasible for small networks when done naively. One approach is to use matrix-free methods to approximately compute Gauss-Newton directions by introducing an inner optimization loop, see [51, 36] for supervised learning problems and [64, 4, 24, 62] for PINNs. Instead, our KFAC-based approach uses an explicit structured curvature representation which can be updated over iterations and inverted more cheaply.

We build on the literature on Kronecker-factored approximate curvature (KFAC), which was initially introduced in Heskes [22], Martens [36] as an approximation of the per-layer Fisher matrix to perform approximate natural gradient descent. Later, KFAC was extended to convolutional [20], recurrent [39], attention [47, 44, 21], and recently to general linear layers with weight sharing [17]. These works do not address preconditioners for losses with contributions from differential operators, as is the case for PINN losses. Our interpretation via Taylor-mode makes the computation graph of such losses explicit, and allows us to establish KFAC based on its generalization to linear weight sharing layers [17].

## 2 Background

For simplicity, we present our approach for multi-layer perceptrons (MLPs) consisting of fully-connected and element-wise activation layers. However, the generality of Taylor-mode automatic differentiation and KFAC for linear layers with weight sharing allows our KFAC to be applied to such layers (e.g. fully-connected, convolution, attention) in arbitrary neural network architectures.

**Flattening & Derivatives**  We vectorize matrices using the *first-index-varies-fastest* convention, i.e. column-stacking (row index varies first, column index varies second) and denote the corresponding flattening operation by vec. This allows to reduce derivatives of matrix- or tensor-valued objects back to the vector case by flattening a function's input and output before differentiation. The Jacobian of a vector-to-vector function $\boldsymbol{a} \mapsto \boldsymbol{b}(\boldsymbol{a})$ has entries $[\mathrm{J}_{\boldsymbol{a}}\boldsymbol{b}]_{i,j} = \partial b_i / \partial a_j$. For a matrix-to-matrix function $\boldsymbol{A} \mapsto \boldsymbol{B}(\boldsymbol{A})$, the Jacobian is $\mathrm{J}_{\boldsymbol{A}}\boldsymbol{B} = \mathrm{J}_{\mathrm{vec}\,\boldsymbol{A}}\,\mathrm{vec}\,\boldsymbol{B}$. A useful property of vec is $\mathrm{vec}(\boldsymbol{A}\boldsymbol{X}\boldsymbol{B}) = (\boldsymbol{B}^\top \otimes \boldsymbol{A})\,\mathrm{vec}\,\boldsymbol{X}$ for matrices $\boldsymbol{A}, \boldsymbol{X}, \boldsymbol{B}$ which implies $\mathrm{J}_{\boldsymbol{X}}(\boldsymbol{A}\boldsymbol{X}\boldsymbol{B}) = \boldsymbol{B}^\top \otimes \boldsymbol{A}$.

**Sequential neural nets**  Consider a *sequential neural network* $u_{\boldsymbol{\theta}} = f_{\boldsymbol{\theta}^{(L)}} \circ f_{\boldsymbol{\theta}^{(L-1)}} \circ \ldots \circ f_{\boldsymbol{\theta}^{(1)}}$ of depth $L \in \mathbb{N}$. It consists of layers $f_{\boldsymbol{\theta}^{(l)}} \colon \mathbb{R}^{h^{(l-1)}} \to \mathbb{R}^{h^{(l)}}, \boldsymbol{z}^{(l-1)} \mapsto \boldsymbol{z}^{(l)} = f_{\boldsymbol{\theta}^{(l)}}(\boldsymbol{z}^{(l-1)})$ with

trainable parameters $\boldsymbol{\theta}^{(l)} \in \mathbb{R}^{p^{(l)}}$ that transform an input $\boldsymbol{z}^{(0)} \coloneqq \boldsymbol{x} \in \mathbb{R}^{d := h^{(0)}}$ into a prediction $u_{\boldsymbol{\theta}}(\boldsymbol{x}) = \boldsymbol{z}^{(L)} \in \mathbb{R}^{h^{(L)}}$ via intermediate representations $\boldsymbol{z}^{(l)} \in \mathbb{R}^{h^{(l)}}$. In the context of PINNs, we use networks with scalar outputs ($h^{(L)} = 1$) and denote the concatenation of all parameters by $\boldsymbol{\theta} = (\boldsymbol{\theta}^{(1)\top}, \dots, \boldsymbol{\theta}^{(L)\top})^{\top} \in \mathbb{R}^{D}$. A common choice is to alternate fully-connected and activation layers. Linear layers map $\boldsymbol{z}^{(l-1)} \mapsto \boldsymbol{z}^{(l)} = \boldsymbol{W}^{(l)} \boldsymbol{z}^{(l-1)}$ using a weight matrix $\boldsymbol{W}^{(l)} = \mathrm{vec}^{-1} \boldsymbol{\theta}^{(l)} \in \mathbb{R}^{h^{(l)} \times h^{(l-1)}}$ (bias terms can be added as an additional column and by appending a 1 to the input). Activation layers map $\boldsymbol{z}^{(l-1)} \mapsto \boldsymbol{z}^{(l)} = \sigma(\boldsymbol{z}^{(l-1)})$ element-wise for a (typically smooth) $\sigma \colon \mathbb{R} \to \mathbb{R}$.

## 2.1 Energy Natural Gradients for Physics-Informed Neural Networks

Let us consider a domain $\Omega \subseteq \mathbb{R}^{d}$ and the partial differential equation

$$\mathcal{L}u = f \quad \text{in } \Omega, \qquad u = g \quad \text{on } \partial\Omega,$$

with right-hand side $f$, boundary data $g$ and a differential operator $\mathcal{L}$, e.g. the negative Laplacian $-\mathcal{L}u = \Delta_{\boldsymbol{x}}u = \sum_{i=1}^{d} \partial_{x_i}^{2} u$. We parametrize $u$ with a neural net and train its parameters $\boldsymbol{\theta}$ to minimize the loss

$$
\begin{aligned}
L(\boldsymbol{\theta}) &= \frac{1}{2N_{\Omega}} \sum_{n=1}^{N_{\Omega}} (\mathcal{L}u_{\boldsymbol{\theta}}(\boldsymbol{x}_n) - f(\boldsymbol{x}_n))^2 + \frac{1}{2N_{\partial\Omega}} \sum_{n=1}^{N_{\partial\Omega}} (u_{\boldsymbol{\theta}}(\boldsymbol{x}_n^{\mathrm{b}}) - g(\boldsymbol{x}_n^{\mathrm{b}}))^2 \\
&=: L_{\Omega}(\boldsymbol{\theta}) + L_{\partial\Omega}(\boldsymbol{\theta})
\end{aligned}
\tag{1}
$$

with points $\{\boldsymbol{x}_n \in \Omega\}_{n=1}^{N_{\Omega}}$ from the domain's interior, and points $\{\boldsymbol{x}_n^{\mathrm{b}} \in \partial\Omega\}_{n=1}^{N_{\partial\Omega}}$ on its boundary.[2]

First-order optimizers like gradient descent and Adam struggle at producing satisfactory solutions when used to train PINNs [9]. Instead, function space-inspired second-order methods have lately shown promising results [42]. We focus on *energy natural gradient descent (ENGD [41])* which—applied to PINN objectives like (1)—corresponds to the Gauss-Newton method [6, Chapter 6.3]. ENGD mimics Newton's method *in function space* up to a projection onto the model's tangent space and a discretization error that vanishes quadratically in the step size, thus providing locally optimal residual updates. Alternatively, the Gauss-Newton method can be motivated from the standpoint of operator preconditioning, where the Gauss-Newton matrix leads to optimal conditioning of the problem [14].

Natural gradient methods perform parameter updates via a preconditioned gradient descent scheme $\boldsymbol{\theta} \leftarrow \boldsymbol{\theta} - \alpha \boldsymbol{G}(\boldsymbol{\theta})^{+} \nabla L(\boldsymbol{\theta})$, where $\boldsymbol{G}(\boldsymbol{\theta})^{+}$ denotes the pseudo-inverse of a suitable *Gramian matrix* $\boldsymbol{G}(\boldsymbol{\theta}) \in \mathbb{R}^{D \times D}$ and $\alpha$ is a step size. ENGD for the PINN loss (1) uses the Gramian

$$
\begin{aligned}
\boldsymbol{G}(\boldsymbol{\theta}) &= \frac{1}{N_{\Omega}} \sum_{n=1}^{N_{\Omega}} \left( \mathrm{J}_{\boldsymbol{\theta}} \mathcal{L}u_{\boldsymbol{\theta}}(\boldsymbol{x}_n) \right)^{\top} \mathrm{J}_{\boldsymbol{\theta}} \mathcal{L}u_{\boldsymbol{\theta}}(\boldsymbol{x}_n) + \frac{1}{N_{\partial\Omega}} \sum_{n=1}^{N_{\partial\Omega}} \left( \mathrm{J}_{\boldsymbol{\theta}} u_{\boldsymbol{\theta}}(\boldsymbol{x}_n^{\mathrm{b}}) \right)^{\top} \mathrm{J}_{\boldsymbol{\theta}} u_{\boldsymbol{\theta}}(\boldsymbol{x}_n^{\mathrm{b}}) \\
&=: \boldsymbol{G}_{\Omega}(\boldsymbol{\theta}) + \boldsymbol{G}_{\partial\Omega}(\boldsymbol{\theta}) \,.
\end{aligned}
\tag{2}
$$

(2) is the Gauss-Newton matrix of the residual $\boldsymbol{r}(\boldsymbol{\theta}) = \left( \boldsymbol{r}_{\Omega}(\boldsymbol{\theta})^{\top} / \sqrt{N_{\Omega}}, \boldsymbol{r}_{\partial\Omega}(\boldsymbol{\theta})^{\top} / \sqrt{N_{\partial\Omega}} \right)^{\top} \in \mathbb{R}^{N_{\Omega} + N_{\partial\Omega}}$ with interior and boundary residuals $r_{\Omega,n}(\boldsymbol{\theta}) = \mathcal{L}u_{\boldsymbol{\theta}}(\boldsymbol{x}_n) - f(\boldsymbol{x}_n)$ and $r_{\partial\Omega,n}(\boldsymbol{\theta}) = u_{\boldsymbol{\theta}}(\boldsymbol{x}_n^{\mathrm{b}}) - g(\boldsymbol{x}_n^{\mathrm{b}})$.

## 2.2 Kronecker-factored Approximate Curvature

We review Kronecker-factored approximate curvature (KFAC) which was introduced by Heskes [22], Martens & Grosse [38] in the context of maximum likelihood estimation to approximate the per-layer Fisher information matrix by a Kronecker product to speed up approximate natural gradient descent [1]. The Fisher associated with the loss $1/2N \sum_{n=1}^{N} \| u_{\boldsymbol{\theta}}(\boldsymbol{x}_n) - y_n \|_2^2$ with targets $y_n \in \mathbb{R}$ is

$$
\boldsymbol{F}(\boldsymbol{\theta}) = \frac{1}{N} \sum_{n=1}^{N} \left( \mathrm{J}_{\boldsymbol{\theta}} u_{\boldsymbol{\theta}}(\boldsymbol{x}_n) \right)^{\top} \mathrm{J}_{\boldsymbol{\theta}} u_{\boldsymbol{\theta}}(\boldsymbol{x}_n) = \frac{1}{N} \sum_{n=1}^{N} \left( \mathrm{J}_{\boldsymbol{\theta}} u_n \right)^{\top} \mathrm{J}_{\boldsymbol{\theta}} u_n \quad \in \mathbb{R}^{D \times D}, \tag{3}
$$

where $u_n = u_{\boldsymbol{\theta}}(\boldsymbol{x}_n)$, and it coincides with the classical Gauss-Newton matrix [37]. The established KFAC approximates (3). While the boundary Gramian $\boldsymbol{G}_{\partial\Omega}(\boldsymbol{\theta})$ has the same structure as $\boldsymbol{F}(\boldsymbol{\theta})$, the interior Gramian $\boldsymbol{G}_{\Omega}(\boldsymbol{\theta})$ does not as it involves derivative rather than function evaluations of the net.

---

[2]The second regression loss can also include other constraints like measurement data.

KFAC tackles the Fisher's per-layer block diagonal, $\boldsymbol{F}(\boldsymbol{\theta}) \approx \mathrm{diag}(\boldsymbol{F}^{(1)}(\boldsymbol{\theta}), \ldots, \boldsymbol{F}^{(L)}(\boldsymbol{\theta}))$ with $\boldsymbol{F}^{(l)}(\boldsymbol{\theta}) = {}^1\!/_N \sum_{n=1}^{N} (\mathrm{J}_{\boldsymbol{\theta}^{(l)}} u_n)^\top \mathrm{J}_{\boldsymbol{\theta}^{(l)}} u_n \in \mathbb{R}^{p^{(l)} \times p^{(l)}}$. For a fully-connected layer's block, let's examine the term $\mathrm{J}_{\boldsymbol{\theta}^{(l)}} u_{\boldsymbol{\theta}}(\boldsymbol{x})$ from Equation (3) for a fixed data point. The layer parameters $\boldsymbol{\theta}^{(l)} = \mathrm{vec}\, \boldsymbol{W}^{(l)}$ enter the computation via $\boldsymbol{z}^{(l)} = \boldsymbol{W}^{(l)} \boldsymbol{z}^{(l-1)}$ and we have $\mathrm{J}_{\boldsymbol{W}^{(l)}} \boldsymbol{z}^{(l)} = \boldsymbol{z}^{(l-1)^\top} \otimes \boldsymbol{I}$ [e.g. 10]. Further, the chain rule gives the decomposition $\mathrm{J}_{\boldsymbol{W}^{(l)}} u = (\mathrm{J}_{\boldsymbol{z}^{(l)}} u) \mathrm{J}_{\boldsymbol{W}^{(l)}} \boldsymbol{z}^{(l)} = \boldsymbol{z}^{(l-1)^\top} \otimes \mathrm{J}_{\boldsymbol{z}^{(l)}} u$. Inserting into $\boldsymbol{F}^{(l)}(\boldsymbol{\theta})$, summing over data points, and using the expectation approximation $\sum_n \boldsymbol{A}_n \otimes \boldsymbol{B}_n \approx N^{-1}(\sum_n \boldsymbol{A}_n) \otimes (\sum_n \boldsymbol{B}_n)$ from Martens & Grosse [38], we obtain the KFAC approximation for linear layers in supervised square loss regression with a network's output,

$$\boldsymbol{F}^{(l)}(\boldsymbol{\theta}) \approx \underbrace{\left( \frac{1}{N} \sum_{n=1}^{N} \boldsymbol{z}_n^{(l-1)} \boldsymbol{z}_n^{(l-1)^\top} \right)}_{=: \boldsymbol{A}^{(l)} \in \mathbb{R}^{h^{(l-1)} \times h^{(l-1)}}} \otimes \underbrace{\left( \frac{1}{N} \sum_{n=1}^{N} (\mathrm{J}_{\boldsymbol{z}^{(l)}} u_n)^\top \, \mathrm{J}_{\boldsymbol{z}^{(l)}} u_n \right)}_{=: \boldsymbol{B}^{(l)} \in \mathbb{R}^{h^{(l)} \times h^{(l)}}} . \tag{4}$$

It is cheap to store and invert by inverting the two Kronecker factors.

## 3 Kronecker-Factored Approximate Curvature for PINNs

ENGD's Gramian is a sum of PDE and boundary Gramians, $\boldsymbol{G}(\boldsymbol{\theta}) = \boldsymbol{G}_\Omega(\boldsymbol{\theta}) + \boldsymbol{G}_{\partial\Omega}(\boldsymbol{\theta})$. We will approximate each Gramian separately with a block diagonal matrix with Kronecker-factored blocks, $\boldsymbol{G}_\bullet(\boldsymbol{\theta}) \approx \mathrm{diag}(\boldsymbol{G}_\bullet^{(1)}(\boldsymbol{\theta}), \ldots, \boldsymbol{G}_\bullet^{(L)}(\boldsymbol{\theta}))$ for $\bullet \in \{\Omega, \partial\Omega\}$ with $\boldsymbol{G}_\bullet^{(l)}(\boldsymbol{\theta}) \approx \boldsymbol{A}_\bullet^{(l)} \otimes \boldsymbol{B}_\bullet^{(l)}$. For the boundary Gramian $\boldsymbol{G}_{\partial\Omega}(\boldsymbol{\theta})$, we can re-use the established KFAC from Equation (4) as its loss corresponds to regression over the network's output. The interior Gramian $\boldsymbol{G}_\Omega(\boldsymbol{\theta})$, however, involves PDE terms in the form of network derivatives and therefore *cannot* be approximated with the existing KFAC. It requires a new approximation that we develop here for the running example of the Poisson equation and more general PDEs (Equations (9) and (14)). To do so, we need to make the dependency between the weights and the differential operator $\mathcal{L}u$ explicit. We use Taylor-mode automatic differentiation to express this computation of higher-order derivatives as forward passes of a larger net with shared weights, for which we then propose a Kronecker-factored approximation, building on KFAC's recently-proposed generalization to linear layers with weight sharing [17].

### 3.1 Higher-order Forward Mode Automatic Differentiation as Weight Sharing

Here, we review higher-order forward mode, also known as *Taylor-mode*, automatic differentiation [19, 18, 3, tutorial in §C]. Many PDEs only incorporate first- and second-order partial derivatives and we focus our discussion on second-order Taylor-mode for MLPs to keep the presentation light. However, one can treat higher-order PDEs and arbitrary network architectures completely analogously.

Taylor-mode propagates directional (higher-order) derivatives. We now recap the forward propagation rules for MLPs consisting of fully-connected and element-wise activation layers. Our goal is to evaluate first-and second-order partial derivatives of the form $\partial_{x_i} u, \partial_{x_i, x_j}^2 u$ for $i, j = 1, \ldots, d$. At the first layer, set $\boldsymbol{z}^{(0)} = \boldsymbol{x} \in \mathbb{R}^d, \partial_{x_i} \boldsymbol{z}^{(0)} = \boldsymbol{e}_i \in \mathbb{R}^d$, i.e., the $i$-th basis vector and $\partial_{x_i, x_j}^2 \boldsymbol{z}^{(0)} = \boldsymbol{0} \in \mathbb{R}^d$.

For a linear layer $f_{\boldsymbol{\theta}^{(l)}}(\boldsymbol{z}^{(l-1)}) = \boldsymbol{W}^{(l)} \boldsymbol{z}^{(l-1)}$, applying the chain rule yields the propagation rule

$$\boldsymbol{z}^{(l)} = \boldsymbol{W}^{(l)} \boldsymbol{z}^{(l-1)} \quad \in \mathbb{R}^{h^{(l)}} , \tag{5a}$$

$$\partial_{x_i} \boldsymbol{z}^{(l)} = \boldsymbol{W}^{(l)} \partial_{x_i} \boldsymbol{z}^{(l-1)} \quad \in \mathbb{R}^{h^{(l)}} , \tag{5b}$$

$$\partial_{x_i, x_j}^2 \boldsymbol{z}^{(l)} = \boldsymbol{W}^{(l)} \partial_{x_i, x_j}^2 \boldsymbol{z}^{(l-1)} \quad \in \mathbb{R}^{h^{(l)}} . \tag{5c}$$

The propagation rule through a nonlinear element-wise activation layer $\boldsymbol{z}^{(l-1)} \mapsto \sigma(\boldsymbol{z}^{(l-1)})$ is

$$\boldsymbol{z}^{(l)} = \sigma(\boldsymbol{z}^{(l-1)}) \quad \in \mathbb{R}^{h^{(l)}} , \tag{6a}$$

$$\partial_{x_i} \boldsymbol{z}^{(l)} = \sigma'(\boldsymbol{z}^{(l-1)}) \odot \partial_{x_i} \boldsymbol{z}^{(l-1)} \quad \in \mathbb{R}^{h^{(l)}} , \tag{6b}$$

$$\partial_{x_i, x_j}^2 \boldsymbol{z}^{(l)} = \partial_{x_i} \boldsymbol{z}^{(l-1)} \odot \sigma''(\boldsymbol{z}^{(l-1)}) \odot \partial_{x_j} \boldsymbol{z}^{(l-1)} + \sigma'(\boldsymbol{z}^{(l-1)}) \odot \partial_{x_i, x_j}^2 \boldsymbol{z}^{(l-1)} \quad \in \mathbb{R}^{h^{(l)}} . \tag{6c}$$

**Forward Laplacian** For differential operators of special structure, we can fuse the Taylor-mode forward propagation of individual directional derivatives in Equations (5) and (6) and obtain a more efficient computation. E.g., to compute not the full Hessian but only the Laplacian, we can simplify the forward pass, which yields the *forward Laplacian* framework of Li et al. [29]. To the best of our knowledge, this connection has not been pointed out in the literature. Concretely, by summing (5c) and (6c) over $i = j$, we obtain the Laplacian forward pass for linear and activation layers

$$\Delta_{\boldsymbol{x}} \boldsymbol{z}^{(l)} = \boldsymbol{W}^{(l)} \Delta_{\boldsymbol{x}} \boldsymbol{z}^{(l-1)} \quad \in \mathbb{R}^{h^{(l)}}, \tag{7a}$$

$$\Delta_{\boldsymbol{x}} \boldsymbol{z}^{(l)} = \sigma'(\boldsymbol{z}^{(l-1)}) \odot \Delta_{\boldsymbol{x}} \boldsymbol{z}^{(l-1)} + \sum_{i=1}^{d} \sigma''(\boldsymbol{z}^{(l-1)}) \odot (\partial_{x_i} \boldsymbol{z}^{(l-1)})^{\odot 2} \quad \in \mathbb{R}^{h^{(l)}}. \tag{7b}$$

This reduces computational cost, but is restricted to PDEs that involve second-order derivatives only via the Laplacian, or a partial Laplacian over a sub-set of input coordinates (e.g. heat equation, §4). For a more general second-order linear PDE operator $\mathcal{L} = \sum_{i,j=1}^{d} c_{i,j} \partial^2_{x_i, x_j}$, the forward pass for a linear layer is $\mathcal{L}\boldsymbol{z}^{(l)} = \boldsymbol{W}^{(l)} \mathcal{L}\boldsymbol{z}^{(l-1)} \in \mathbb{R}^{h^{(l)}}$, generalizing (7a), and similarly for Equation (7b)

$$\mathcal{L}\boldsymbol{z}^{(l)} = \sigma'(\boldsymbol{z}^{(l-1)}) \odot \mathcal{L}\boldsymbol{z}^{(l-1)} + \sum_{i,j=1}^{d} c_{i,j} \sigma''(\boldsymbol{z}^{(l-1)}) \odot \partial_{x_i} \boldsymbol{z}^{(l-1)} \odot \partial_{x_j} \boldsymbol{z}^{(l-1)} \quad \in \mathbb{R}^{h^{(l)}},$$

see §C.3 for details. This is different from [30], which transforms the input space such that the coefficients are diagonal with entries $\{0, \pm 1\}$, reducing the computation to two forward Laplacians.

Importantly, the computation of higher-order derivatives for linear layers boils down to a forward pass through the layer with weight sharing over the different partial derivatives (Equation (5)), and weight sharing can potentially be reduced depending on the differential operator's structure (Equation (7a)). Therefore, we can use the concept of KFAC in the presence of weight sharing to derive a principled Kronecker approximation for Gramians containing differential operator terms.

## 3.2 KFAC for Gauss-Newton Matrices with the Laplace Operator

Let's consider the Poisson equation's interior Gramian block for a linear layer (suppressing $\Omega$ in $N_{\Omega}$)

$$\boldsymbol{G}_{\Omega}^{(l)}(\boldsymbol{\theta}) = \frac{1}{N} \sum_{n=1}^{N} (\mathrm{J}_{\boldsymbol{W}^{(l)}} \Delta_{\boldsymbol{x}} u_n)^{\top} \mathrm{J}_{\boldsymbol{W}^{(l)}} \Delta_{\boldsymbol{x}} u_n.$$

Because we made the Laplacian computation explicit through Taylor-mode autodiff (§3.1, specifically Equation (7a)), we can stack all output vectors that share the layer's weight into a matrix $\boldsymbol{Z}_n^{(l)} \in \mathbb{R}^{h^{(l)} \times S}$ with $S = d + 2$ and columns $\boldsymbol{Z}_{n,1}^{(l)} = \boldsymbol{z}_n^{(l)}, \boldsymbol{Z}_{n,2}^{(l)} = \partial_{x_1} \boldsymbol{z}_n^{(l)}, \dots, \boldsymbol{Z}_{n,1+d}^{(l)} = \partial_{x_d} \boldsymbol{z}_n^{(l)}$, and $\boldsymbol{Z}_{n,2+d}^{(l)} = \Delta_{\boldsymbol{x}} \boldsymbol{z}_n^{(l)}$ (likewise $\boldsymbol{Z}_n^{(l)} \in \mathbb{R}^{h^{(l-1)} \times S}$ for the layer inputs), then apply the chain rule

$$\mathrm{J}_{\boldsymbol{W}^{(l)}} \Delta_{\boldsymbol{x}} u_n = (\mathrm{J}_{\boldsymbol{Z}_n^{(l)}} \Delta_{\boldsymbol{x}} u_n) \mathrm{J}_{\boldsymbol{W}^{(l)}} \boldsymbol{Z}_n^{(l)} = \sum_{s=1}^{S} \underbrace{\boldsymbol{Z}_{n,s}^{(l-1)}}_{\in \mathbb{R}^{h^{(l-1)}}}^{\top} \otimes \underbrace{\mathrm{J}_{\boldsymbol{Z}_{n,s}^{(l)}} \Delta_{\boldsymbol{x}} u_n}_{=: \boldsymbol{g}_{n,s}^{(l)} \in \mathbb{R}^{h^{(l)}}},$$

which has a structure similar to the Jacobian in §2.2, but with an additional sum over the $S$ shared vectors. With that, we can now express the exact interior Gramian for a layer as

$$\boldsymbol{G}_{\Omega}^{(l)}(\boldsymbol{\theta}) = \frac{1}{N} \sum_{n=1}^{N} \sum_{s=1}^{S} \sum_{s'=1}^{S} \boldsymbol{Z}_{n,s}^{(l-1)} \boldsymbol{Z}_{n,s'}^{(l-1)\top} \otimes \boldsymbol{g}_{n,s}^{(l)} \boldsymbol{g}_{n,s'}^{(l)\top}. \tag{8}$$

Next, we want to approximate Equation (8) with a Kronecker product. To avoid introducing a new convention, we rely on the KFAC approximation for linear layers with weight sharing developed by Eschenhagen et al. [17]—specifically, the approximation called *KFAC-expand*. This drops all terms with $s \neq s'$, then applies the expectation approximation from §2.2 over the batch and shared axes:



KFAC for the Gauss-Newton matrix of a Laplace operator

$$\boldsymbol{G}_{\Omega}^{(l)}(\boldsymbol{\theta}) \approx \left( \frac{1}{NS} \sum_{n,s=1}^{N,S} \boldsymbol{Z}_{n,s}^{(l-1)} \boldsymbol{Z}_{n,s}^{(l-1)\top} \right) \otimes \left( \frac{1}{N} \sum_{n,s=1}^{N,S} \boldsymbol{g}_{n,s}^{(l)} \boldsymbol{g}_{n,s}^{(l)\top} \right) =: \boldsymbol{A}_{\Omega}^{(l)} \otimes \boldsymbol{B}_{\Omega}^{(l)} \tag{9}$$



### 3.3 KFAC for Generalized Gauss-Newton Matrices Involving General PDE Terms

To generalize the previous section, let's consider the general $M$-dimensional PDE system of order $k$,

$$\Psi(u, D_{\boldsymbol{x}}u, \ldots, D_{\boldsymbol{x}}^{k}u) = \mathbf{0} \in \mathbb{R}^{M}, \tag{10}$$

where $D_{\boldsymbol{x}}^{m}u$ collects all partial derivatives of order $m$. For $m \in \{0, \ldots, k\}$ there are $S_m = \binom{d+m-1}{d-1}$ independent partial derivatives and the total number of independent derivatives is $S := \sum_{m=0}^{k} S_m = \binom{d+k}{k}$. $\Psi$ is a smooth mapping from all partial derivatives to $\mathbb{R}^{M}$, $\Psi \colon \mathbb{R}^{S} \to \mathbb{R}^{M}$. To construct a PINN loss for Equation (10), we feed the residual $\boldsymbol{r}_{\Omega,n}(\boldsymbol{\theta}) := \Psi(u_{\boldsymbol{\theta}}(\boldsymbol{x}_n), D_{\boldsymbol{x}}u_{\boldsymbol{\theta}}(\boldsymbol{x}_n), \ldots, D_{\boldsymbol{x}}^{k}u_{\boldsymbol{\theta}}(\boldsymbol{x}_n)) \in \mathbb{R}^{M}$ where $D_{\boldsymbol{x}}^{m}u_{\boldsymbol{\theta}}(\boldsymbol{x}_n) \in \mathbb{R}^{d \times S_m}$ into a smooth convex criterion function $\ell \colon \mathbb{R}^{M} \to \mathbb{R}$,

$$L_{\Omega}(\boldsymbol{\theta}) := \frac{1}{N} \sum_{n=1}^{N} \ell(\boldsymbol{r}_{\Omega,n}(\boldsymbol{\theta})) . \tag{11}$$

The generalized Gauss-Newton (GGN) matrix [51] is the Hessian of $L_{\Omega}(\boldsymbol{\theta})$ when the residual is linearized w.r.t. $\boldsymbol{\theta}$ before differentiation. It is positive semi-definite and has the form

$$\boldsymbol{G}_{\Omega}(\boldsymbol{\theta}) := \frac{1}{N} \sum_{n=1}^{N} (\mathrm{J}_{\boldsymbol{\theta}}\boldsymbol{r}_{\Omega,n}(\boldsymbol{\theta}))^{\top} \boldsymbol{\Lambda}(\boldsymbol{r}_{\Omega,n}) (\mathrm{J}_{\boldsymbol{\theta}}\boldsymbol{r}_{\Omega,n}(\boldsymbol{\theta})) , \tag{12}$$

with $\boldsymbol{\Lambda}(\boldsymbol{r}) := \nabla_{\boldsymbol{r}}^{2}\ell(\boldsymbol{r}) \in \mathbb{R}^{M \times M} \succ 0$ the criterion's Hessian, e.g. $\ell(\boldsymbol{r}) = 1/2\|\boldsymbol{r}\|_2^2$ and $\boldsymbol{\Lambda}(\boldsymbol{r}) = \boldsymbol{I}_M$.

Generalizing the second-order Taylor-mode from §3.1 to higher orders for the linear layer, we find

$$D_{\boldsymbol{x}}^{m}\boldsymbol{z}^{(l)} = \boldsymbol{W}^{(l)} D_{\boldsymbol{x}}^{m}\boldsymbol{z}^{(l-1)} \qquad \in \mathbb{R}^{h^{(l)} \times S_m} \tag{13}$$

for any $m$. Hence, we can derive a forward propagation for the required derivatives where a linear layer processes at most $S$ vectors[3], i.e. the linear layer's weight is shared over the matrices $D_{\boldsymbol{x}}^{0}\boldsymbol{z}^{(l-1)} := \boldsymbol{z}^{(l-1)}, D_{\boldsymbol{x}}^{1}\boldsymbol{z}^{(l-1)}, \ldots, D_{\boldsymbol{x}}^{k}\boldsymbol{z}^{(l-1)}$. Stacking them into a matrix $\boldsymbol{Z}_n^{(l-1)} = (\boldsymbol{z}^{(l-1)}, D_{\boldsymbol{x}}^{1}\boldsymbol{z}^{(l-1)}, \ldots, D_{\boldsymbol{x}}^{k}\boldsymbol{z}^{(l-1)}) \in \mathbb{R}^{h^{(l-1)} \times S}$ (and $\boldsymbol{Z}_n^{(l)}$ for the outputs), the chain rule yields

$$\begin{aligned}
\boldsymbol{G}_{\Omega}^{(l)}(\boldsymbol{\theta}) &= \frac{1}{N} \sum_{n=1}^{N} \left(\mathrm{J}_{\boldsymbol{W}^{(l)}}\boldsymbol{Z}_n^{(l)}\right)^{\top} \left(\mathrm{J}_{\boldsymbol{Z}_n^{(l)}}\boldsymbol{r}_{\Omega,n}\right)^{\top} \boldsymbol{\Lambda}(\boldsymbol{r}_{\Omega,n}) \left(\mathrm{J}_{\boldsymbol{Z}_n^{(l)}}\boldsymbol{r}_{\Omega,n}\right) \left(\mathrm{J}_{\boldsymbol{W}^{(l)}}\boldsymbol{Z}_n^{(l)}\right) \\
&= \frac{1}{N} \sum_{n,s,s'=1}^{N,S,S} \left(\mathrm{J}_{\boldsymbol{W}^{(l)}}\boldsymbol{Z}_{n,s}^{(l)}\right)^{\top} \left(\mathrm{J}_{\boldsymbol{Z}_{n,s}^{(l)}}\boldsymbol{r}_{\Omega,n}\right)^{\top} \boldsymbol{\Lambda}(\boldsymbol{r}_{\Omega,n}) \left(\mathrm{J}_{\boldsymbol{Z}_{n,s'}^{(l)}}\boldsymbol{r}_{\Omega,n}\right) \left(\mathrm{J}_{\boldsymbol{W}^{(l)}}\boldsymbol{Z}_{n,s'}^{(l)}\right) \\
&= \frac{1}{N} \sum_{n,s,s'=1}^{N,S,S} \boldsymbol{Z}_{n,s}^{(l-1)} \boldsymbol{Z}_{n,s'}^{(l-1)\top} \otimes \left(\mathrm{J}_{\boldsymbol{Z}_{n,s}^{(l)}}\boldsymbol{r}_{\Omega,n}\right)^{\top} \boldsymbol{\Lambda}(\boldsymbol{r}_{\Omega,n}) \left(\mathrm{J}_{\boldsymbol{Z}_{n,s'}^{(l)}}\boldsymbol{r}_{\Omega,n}\right)
\end{aligned}$$

where $\boldsymbol{Z}_{n,s}^{(l-1)} \in \mathbb{R}^{h^{(l-1)}}$ denotes the $s$-th column of $\boldsymbol{Z}_n^{(l-1)}$. Following the same steps as in §3.2, we apply the KFAC-expand approximation from [17] to obtain the generalization of Equation (9):

---

**KFAC for the GGN matrix of a general PDE operator**

$$\boldsymbol{G}_{\Omega}^{(l)}(\boldsymbol{\theta}) \approx \left(\frac{1}{NS} \sum_{n,s=1}^{N,S} \boldsymbol{Z}_{n,s}^{(l-1)} \boldsymbol{Z}_{n,s'}^{(l-1)\top}\right) \otimes \left(\frac{1}{N} \sum_{n,s=1}^{N,S} \left(\mathrm{J}_{\boldsymbol{Z}_{n,s}^{(l)}}\boldsymbol{r}_{\Omega,n}\right)^{\top} \boldsymbol{\Lambda}(\boldsymbol{r}_{\Omega,n}) \left(\mathrm{J}_{\boldsymbol{Z}_{n,s}^{(l)}}\boldsymbol{r}_{\Omega,n}\right)\right) \tag{14}$$

$$=: \boldsymbol{A}_{\Omega}^{(l)} \otimes \boldsymbol{B}_{\Omega}^{(l)}$$

---

To bring this expression even closer to Equation (9), we can re-write the second Kronecker factor using an outer product decomposition $\boldsymbol{\Lambda}(\boldsymbol{r}_{\Omega,n}) = \sum_{m=1}^{M} \boldsymbol{l}_{n,m} \boldsymbol{l}_{n,m}$ with $\boldsymbol{l}_{n,m} \in \mathbb{R}^{M}$, then introduce $\boldsymbol{g}_{n,s,m}^{(l)} := (\mathrm{J}_{\boldsymbol{Z}_{n,s}^{(l)}}\boldsymbol{r}_{\Omega,n})^{\top} \boldsymbol{l}_{n,m} \in \mathbb{R}^{h^{(l)}}$ and write the second term as $1/N \sum_{n,s,m=1}^{N,S,M} \boldsymbol{g}_{n,s,m}^{(l)} \boldsymbol{g}_{n,s,m}^{(l)\top}$, similar to the Kronecker-factored low-rank (KFLR) approach of Botev et al. [5].

---

[3]Depending on the linear operator, one may reduce weight sharing, as demonstrated for the Laplacian in §3.1.

**KFAC for variational problems** Our proposed KFAC approximation is not limited to PINNs and can be used for variational problems of the form

$$\min_u \int_\Omega \ell(u, \partial_{\boldsymbol{x}} u, \ldots, \partial_{\boldsymbol{x}}^k u) \mathrm{d}\boldsymbol{x}, \tag{15}$$

where $\ell \colon \mathbb{R}^K \to \mathbb{R}$ is a convex function. We can perceive this as a special case of the setting above with $\Psi = \mathrm{id}$ and hence the KFAC approximation (14) remains meaningful. In particular, it can be used for the *deep Ritz method* and other variational approaches to solve PDEs [16].

## 3.4 Algorithmic Details

To design an optimizer based on our KFAC approximation, we re-use techniques from the original KFAC [38] & ENGD [41] algorithms. §B shows pseudo-code for our method on the Poisson equation.

At iteration $t$, we approximate the per-layer interior and boundary Gramians using our derived Kronecker approximation (Equations (9) and (14)), $\boldsymbol{G}_{\Omega,t}^{(l)} \approx \boldsymbol{A}_{\Omega,t}^{(l)} \otimes \boldsymbol{B}_{\Omega,t}^{(l)}$ and $\boldsymbol{G}_{\partial\Omega,t}^{(l)} \approx \boldsymbol{A}_{\partial\Omega,t}^{(l)} \otimes \boldsymbol{B}_{\partial\Omega,t}^{(l)}$.

**Exponential moving average and damping** For preconditioning, we accumulate the Kronecker factors $\boldsymbol{A}_{\bullet,t}^{(l)}, \boldsymbol{B}_{\bullet,t}^{(l)}$ over time using an exponential moving average $\hat{\boldsymbol{A}}_{\bullet,t}^{(l)} = \beta \hat{\boldsymbol{A}}_{\bullet,t-1}^{(l)} + (1-\beta) \boldsymbol{A}_{\bullet,t}^{(l)}$ of factor $\beta \in [0, 1)$ (identically for $\hat{\boldsymbol{B}}_{\bullet,t}^{(l)}$), similar to the original KFAC. Moreover, we apply the same constant damping of strength $\lambda > 0$ to all Kronecker factors, $\tilde{\boldsymbol{A}}_{\bullet,t}^{(l)} = \hat{\boldsymbol{A}}_{\bullet,t}^{(l)} + \lambda \boldsymbol{I}$ and $\tilde{\boldsymbol{B}}_{\bullet,t}^{(l)} = \hat{\boldsymbol{B}}_{\bullet,t}^{(l)} + \lambda \boldsymbol{I}$ such that the curvature approximation used for preconditioning at step $t$ is

$$\boldsymbol{G}_{\bullet,t} \approx \mathrm{diag}\left( \tilde{\boldsymbol{A}}_{\Omega,t}^{(1)} \otimes \tilde{\boldsymbol{B}}_{\Omega,t}^{(1)}, \ldots, \tilde{\boldsymbol{A}}_{\Omega,t}^{(L)} \otimes \tilde{\boldsymbol{B}}_{\Omega,t}^{(L)} \right) + \mathrm{diag}\left( \tilde{\boldsymbol{A}}_{\partial\Omega,t}^{(1)} \otimes \tilde{\boldsymbol{B}}_{\partial\Omega,t}^{(1)}, \ldots, \tilde{\boldsymbol{A}}_{\partial\Omega,t}^{(L)} \otimes \tilde{\boldsymbol{B}}_{\partial\Omega,t}^{(L)} \right).$$

**Gradient preconditioning** Given layer $l$'s mini-batch gradient $\boldsymbol{g}_t^{(l)} = \partial L(\boldsymbol{\theta}_t)/\partial\boldsymbol{\theta}^{(l)} \in \mathbb{R}^{p^{(l)}}$, we obtain an update direction $\boldsymbol{\Delta}_t^{(l)} = -(\tilde{\boldsymbol{A}}_{\Omega,t}^{(l)} \otimes \tilde{\boldsymbol{B}}_{\Omega,t}^{(l)} + \tilde{\boldsymbol{A}}_{\partial\Omega,t}^{(l)} \otimes \tilde{\boldsymbol{B}}_{\partial\Omega,t}^{(l)})^{-1} \boldsymbol{g}_t^{(l)} \in \mathbb{R}^{p^{(l)}}$ using the trick of [38, Appendix I] to invert the Kronecker sum via eigen-decomposing all Kronecker factors.

**Learning rate and momentum** From the preconditioned gradient $\boldsymbol{\Delta}_t \in \mathbb{R}^D$, we consider two different updates $\boldsymbol{\theta}_{t+1} = \boldsymbol{\theta}_t + \boldsymbol{\delta}_t$ we call *KFAC* and *KFAC\**. KFAC uses momentum over previous updates, $\hat{\boldsymbol{\delta}}_t = \mu \boldsymbol{\delta}_{t-1} + \boldsymbol{\Delta}_t$, and $\mu$ is chosen by the practitioner. Like ENGD, it uses a logarithmic grid line search, selecting $\boldsymbol{\delta}_t = \alpha_\star \hat{\boldsymbol{\delta}}_t$ with $\alpha_\star = \arg\min_\alpha L(\boldsymbol{\theta}_t + \alpha \hat{\boldsymbol{\delta}}_t)$ where $\alpha \in \{2^{-30}, \ldots, 2^0\}$. KFAC* uses the automatic learning rate and momentum heuristic of the original KFAC optimizer. It parametrizes the iteration's update as $\boldsymbol{\delta}_{t+1}(\alpha, \mu) = \alpha \boldsymbol{\Delta}_t + \mu \boldsymbol{\delta}_t$, then obtains the optimal parameters by minimizing the quadratic model $m(\boldsymbol{\delta}_{t+1}) = L(\boldsymbol{\theta}_t) + \boldsymbol{\delta}_{t+1}^\top \boldsymbol{g}_t + 1/2 \boldsymbol{\delta}_{t+1}^\top (\boldsymbol{G}(\boldsymbol{\theta}_t) + \lambda \boldsymbol{I}) \boldsymbol{\delta}_{t+1}$ with the exact damped Gramian. The optimal learning rate and momentum $\arg\min_{\alpha,\mu} m(\boldsymbol{\delta}_{t+1})$ are

$$\begin{pmatrix} \alpha_\star \\ \mu_\star \end{pmatrix} = - \begin{pmatrix} \boldsymbol{\Delta}_t^\top \boldsymbol{G}(\boldsymbol{\theta}_t) \boldsymbol{\Delta}_t + \lambda \|\boldsymbol{\Delta}_t\|^2 & \boldsymbol{\Delta}_t^\top \boldsymbol{G}(\boldsymbol{\theta}_t) \boldsymbol{\delta}_t + \lambda \boldsymbol{\Delta}_t^\top \boldsymbol{\delta}_t \\ \boldsymbol{\Delta}_t^\top \boldsymbol{G}(\boldsymbol{\theta}_t) \boldsymbol{\delta}_t + \lambda \boldsymbol{\Delta}_t^\top \boldsymbol{\delta}_t & \boldsymbol{\delta}_t^\top \boldsymbol{G}(\boldsymbol{\theta}_t) \boldsymbol{\Delta}_t + \lambda \|\boldsymbol{\delta}_t\|^2 \end{pmatrix}^{-1} \begin{pmatrix} \boldsymbol{\Delta}_t^\top \boldsymbol{g}_t \\ \boldsymbol{\delta}_t^\top \boldsymbol{g}_t \end{pmatrix}$$

(see [38, Section 7] for details). The computational cost is dominated by the two Gramian-vector products with $\boldsymbol{\Delta}_t$ and $\boldsymbol{\delta}_t$. By using the Gramian's outer product structure [12, 45], we perform them with autodiff [48, 51] using one Jacobian-vector product each, as recommended in [38].

**Computational complexity** Inverting layer $l$'s Kronecker approximation of the Gramian requires $\mathcal{O}(h^{(l)^3} + h^{(l+1)^3})$ time and $\mathcal{O}(h^{(l)^2} + h^{(l+1)^2})$ storage, where $h^{(l)}$ is the number of neurons in the $l$-th layer, whereas inverting the exact block for layer would require $\mathcal{O}(h^{(l)^3} h^{(l+1)^3})$ time and $\mathcal{O}(h^{(l)^2} h^{(l+1)^2})$ memory. In general, the improvement from the Kronecker factorization depends on how close to square the weight matrices of a layer are, and therefore on the architecture. In practise, the Kronecker factorization usually significantly reduces memory and run time. Further improvements can be achieved by using structured Kronecker factors, e.g. (block-)diagonal matrices [32].

We use the forward Laplacian framework in our implementation, which we found to be significantly faster and more memory efficient than computing batched Hessian traces, see §C.4.

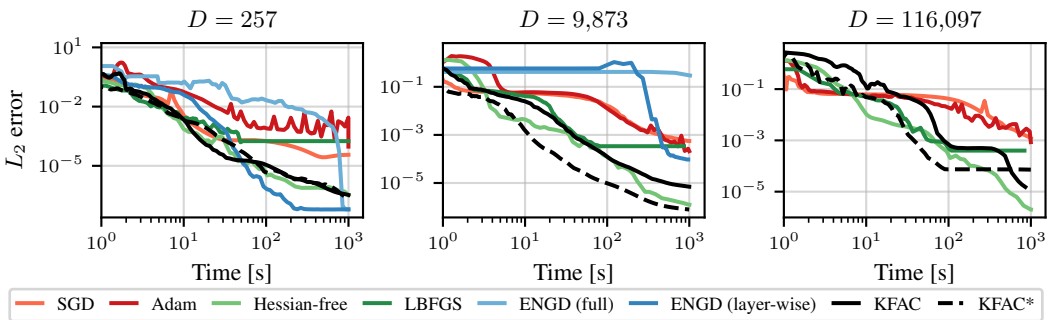

Figure 1: Performance of different optimizers on the 2d Poisson equation (16) measured in relative $L_2$ error against wall clock time for architectures with different parameter dimensions $D$.

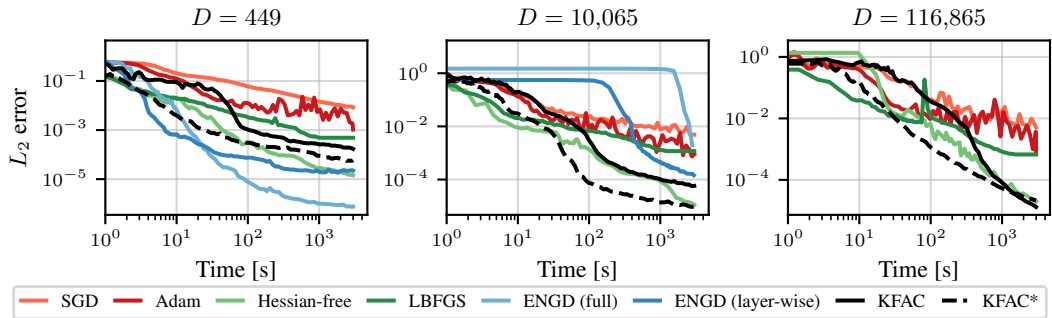

Figure 2: Performance of different optimizers on the (4+1)d heat equation (17) measured in relative $L^2$ error against wall clock time for architectures with different parameter dimensions $D$.

## 4 Experiments

We implement KFAC, KFAC*, and ENGD with either the per-layer or full Gramian in PyTorch [46]. As a matrix-free version of ENGD, we use the Hessian-free optimizer [36] which uses truncated conjugate gradients (CG) with exact Gramian-vector products to precondition the gradient. We chose this because there is a fully-featured implementation from Tatzel et al. [55] which offers many additional heuristics like adaptive damping, CG backtracking, and backtracking line search, allowing this algorithm to work well with little hyper-parameter tuning. As baselines, we use SGD with tuned learning rate and momentum, Adam with tuned learning rate, and LBFGS with tuned learning rate and history size. We tune hyper-parameters using Weights & Biases [60] (see §A.1 for the exact protocol). For random/grid search, we run an initial round of approximately 50 runs with generous search spaces, then narrow them down and re-run for another 50 runs; for Bayesian search, we assign the same total compute to each optimizer. We report runs with lowest $L_2$ error estimated on a held-out data set with the known solution to the studied PDE. To be comparable, all runs are executed on a compute cluster with RTX 6000 GPUs (24 GiB RAM) in double precision, and we use the same computation time budget for all optimizers on a fixed PINN problem. All search spaces and best run hyper-parameters, as well as training curves over iteration count rather than time, are in §A.

**Pedagogical example: 2d Poisson equation**   We start with a low-dimensional Poisson equation from Müller & Zeinhofer [41] to reproduce ENGD's performance (Figure 1). It is given by

$$-\Delta u(x,y) = 2\pi^2 \sin(\pi x)\sin(\pi y) \quad \text{for } (x,y) \in [0,1]^2$$
$$u(x,y) = 0 \quad \text{for } (x,y) \in \partial[0,1]^2. \tag{16}$$

We choose a fixed data set of same size as the original paper, then use random/grid search to evaluate the performance of all optimizers for different tanh-activated MLPs, one shallow and two with five fully-connected layers of different width (all details in §A.2). We include ENGD whenever the network's parameter space is small enough to build up the Gramian.

For the shallow net (Figure 1, left), we can reproduce the results of [41], where exact ENGD achieves high accuracy. In terms of computation time, our KFACs are competitive with full-ENGD for a long

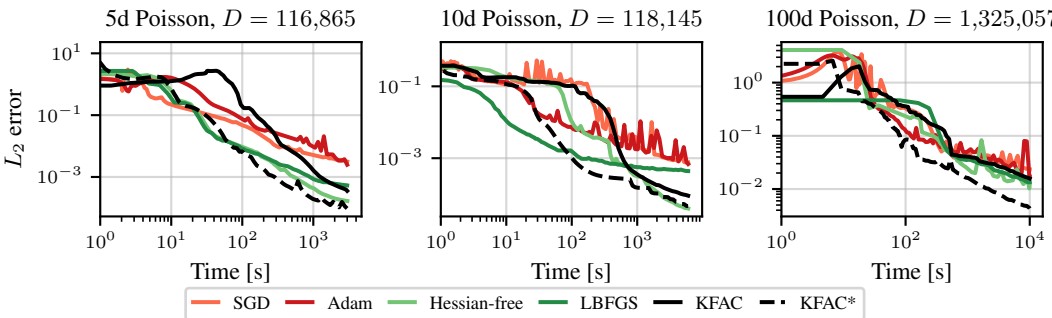

Figure 3: Optimizer performance on Poisson equations in high dimensions and different boundary conditions measured in relative $L_2$ error against wall clock time for networks with $D$ parameters.

phase, outperforming the first-order and quasi-Newton baselines. In contrast to ENGD, which runs out of memory for networks with more than $10\,000$ parameters, KFAC scales to larger networks (Figure 1, center and right) and is competitive with other second-order optimizers like Hessian-free, which uses more sophisticated heuristics. We make similar observations on a small (1+1)d heat equation with the same models, see §A.7 and fig. A10.

**An evolutionary problem: (4+1)d heat equation**   To demonstrate that our methods can also be applied to other problems than the Poisson equation, we consider a four-dimensional heat equation

$$\partial_t u(t, \boldsymbol{x}) - \kappa \Delta_{\boldsymbol{x}} u(t, \boldsymbol{x}) = 0 \quad \text{for } t \in [0, 1], \boldsymbol{x} \in [0, 1]^4 \,,$$

$$u(0, \boldsymbol{x}) = \sum_{i=1}^{4} \sin(2x_i) \quad \text{for } \boldsymbol{x} \in [0, 1]^4 \,, \tag{17}$$

$$u(t, \boldsymbol{x}) = \exp(-t) \sum_{i=1}^{4} \sin(2x_i) \quad \text{for } t \in [0, 1], \boldsymbol{x} \in \partial[0, 1]^4 \,,$$

with diffusivity constant $\kappa = 1/4$, similar to that studied in [41] (see §A.6 for the heat equation's PINN loss). We use the previous architectures with same hidden widths and evaluate optimizer performance with random/grid search (all details in §A.8), see Figure 2. To prevent over-fitting, we use mini-batches and sample a new batch each iteration. We noticed that KFAC improves significantly when batches are sampled less frequently and hypothesize that it might need more iterations to make similar progress than one iteration of Hessian-free or ENGD on a batch. Consequently we sample a new batch only every 100 iterations for KFAC. To ensure that this does not lead to an unfair advantage for KFAC, we conduct an additional experiment for the MLP with $D = 116\,864$ where we tune batch sizes, batch sampling frequencies, and all hyper-parameters with generous search spaces using Bayesian search (§A.10). We find that this does not significantly boost performance of the other methods (compare Figures 2 and A14). Again, we observe that KFAC offers competitive performance compared to other second-order methods for networks with prohibitive size for ENGD and consistently outperforms SGD, Adam, and LBFGS. We confirmed these observations with another 5d Poisson equation on the same architectures, see §A.3 and fig. A7.

**High-dimensional Poisson equations**   To demonstrate scaling to high-dimensional PDEs and even larger neural networks, we consider three Poisson equations ($d = 5, 10, 100$) with different boundary conditions used in [16, 41], which admit the solutions

$$u_\star(\boldsymbol{x}) = \sum_{i=1}^{5} \cos(\pi x_i) \quad \text{for } \boldsymbol{x} \in [0, 1]^5 \,,$$

$$u_\star(\boldsymbol{x}) = \sum_{k=1}^{5} x_{2k-1} x_{2k} \quad \text{for } \boldsymbol{x} \in [0, 1]^{10} \,, \tag{18}$$

$$u_\star(\boldsymbol{x}) = \|\boldsymbol{x}\|_2^2 \quad \text{for } \boldsymbol{x} \in [0, 1]^{100} \,.$$

We use the same architectures as before, but with larger intermediate widths and parameters up to a million (Figure 3). Due to lacking references for training such high-dimensional problems, we

select all hyper-parameters via Bayesian search, including batch sizes and batch sampling frequencies (details in §A.5). We see a similar picture as before with KFAC consistently outperforming first-order methods and LBFGS, offering competitive performance with Hessian-free. To account for the possibility that the Bayesian search did not properly identify good hyper-parameters, we conduct a random/grid search experiment for the 10d Poisson equation (Figure 3, middle), using similar batch sizes and same batch sampling frequencies as for the $(4+1)$d heat equation (details in §A.4). In this experiment, KFAC also achieved similar performance than Hessian-free and outperformed SGD, Adam, and LBFGS (Figure A8).

**(9+1)d Fokker-Planck equation** To show the applicability to nonlinear PDEs, we consider a Fokker-Planck equation in logarithmic space. PINN formulations of the Fokker-Planck equation have been considered in [23, 54]. Concretely, we are solving a nine-dimensional equation of the form

$$\partial_t q(t, \boldsymbol{x}) - \frac{d}{2} - \frac{1}{2}\nabla q(t, \boldsymbol{x}) \cdot \boldsymbol{x} - \|\nabla q(t, \boldsymbol{x})\|^2 - \Delta q(t, \boldsymbol{x}) = 0, \quad q(0) = \log(p^*(0)), \quad (19)$$

with $d = 9$, $t \in [0, 1]$ and $\boldsymbol{x} \in \mathbb{R}^9$, where in practice $\mathbb{R}^9$ is replaced by $[-5, 5]^9$. The solution is $q^* = \log(p^*)$ and $p^*$ is given by $p^*(t, \boldsymbol{x}) \sim \mathcal{N}(0, \exp(-t)\boldsymbol{I} + (1 - \exp(-t))2\boldsymbol{I})$. We model the solution with a medium sized tanh-activated MLP with $D = 118\,145$ parameters, batch sizes are $N_\Omega = 3\,000$, $N_{\partial\Omega} = 1\,000$, and we assign each run a computation time budget of $6\,000$ s. As in previous experiments, the batches are re-sampled every iteration for all optimizers except for KFAC and KFAC*, which use the same batch for ten steps (details in §A.11). Figure 4 reports the $L^2$ error over training time. Again, KFAC is among the best performing optimizers offering competitive performance to Hessian-free and clearly outperforming all first-order methods.

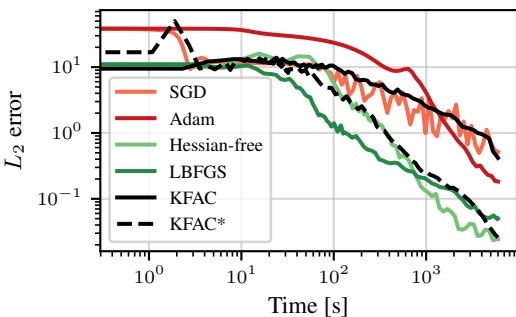

Figure 4: Performance of different optimizers on a (9+1)d logarithmic Fokker-Planck equation in relative $L_2$ error against wall clock time.

## 5 Discussion and Conclusion

We extended the concept of Kronecker-factored approximate curvature (KFAC) to Gauss-Newton matrices of Physics-informed neural network (PINN) losses that involve derivatives, rather than function evaluations, of the neural net. This greatly reduces the computational cost of approximate natural gradient methods, which are known to work well on PINNs, and allows them to scale to much larger nets. Our approach goes beyond the established KFAC for traditional supervised problems as it captures contributions from a PDE's differential operator that are crucial for optimization. To establish KFAC for such losses, we use Taylor-mode autodiff to view the differential operator's compute graph as a forward net with shared weights, then apply the recently-developed formulation of KFAC for linear layers with weight sharing. Empirically, we find that our KFAC-based optimizers are competitive with expensive second-order methods on small problems and scale to high-dimensional neural networks and PDEs while consistently outperforming first-order methods and LBFGS.

**Limitations & future directions** While our implementation currently only supports MLPs and the Poisson and heat equations, the concepts we use to derive KFAC (Taylor-mode, weight sharing) apply to arbitrary architectures and PDEs, as described in §3.3. We are excited that our current algorithms show promising performance when compared to second-order methods with sophisticated heuristics. In fact, the original KFAC optimizer itself [38] relies heavily on such heuristics that are said to be crucial for its performance [8]. Our algorithms borrow components, but we did not explore all bells and whistles, e.g. adaptive damping and heuristics to distribute damping over the Kronecker factors. We believe our current algorithm's performance can further be improved, e.g. by exploring (1) updating the KFAC matrices less frequently, as is standard for traditional KFAC, (2) merging the two Kronecker approximations for boundary and interior Gramians into a single one, (3) removing matrix inversions [31], (4) using structured Kronecker factors [32], (5) computing the Kronecker factors in parallel with the gradient [11], (6) using single or mixed precision training [40], and (7) studying cheaper KFAC flavours based on the empirical Fisher [27] or input-based curvature [2, 49].

## Acknowledgments and Disclosure of Funding

The authors thank Runa Eschenhagen for insightful discussions on KFAC for linear weight sharing layers. FD would like to thank Luca Thiede for his adamant questions about Taylor mode and forward Laplacians. Resources used in preparing this research were provided, in part, by the Province of Ontario, the Government of Canada through CIFAR, and companies sponsoring the Vector Institute. JM acknowledges funding by the Deutsche Forschungsgemeinschaft (DFG, German Research Foundation) under the project number 442047500 through the Collaborative Research Center *Sparsity and Singular Structures* (SFB 1481). MZ acknowledges support from an ETH Postdoctoral Fellowship for the project "Reliable, Efficient, and Scalable Methods for Scientific Machine Learning".

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

# Kronecker-Factored Approximate Curvature for Physics-Informed Neural Networks (Supplementary Material)

## A   Experimental Details and Additional Results

### A.1   Hyper-Parameter Tuning Protocol

In all our experiments, we tune the following optimizer hyper-parameters and otherwise use the
PyTorch default values:

- **SGD:** learning rate, momentum

- **Adam:** learning rate
- **Hessian-free:** type of curvature matrix (Hessian or GGN), damping, whether to adapt damping over time (yes or no), maximum number of CG iterations
- **LBFGS:** learning rate, history size
- **ENGD:** damping, factor of the exponential moving average applied to the Gramian, initialization of the Gramian (zero or identity matrix)
- **KFAC:** factor of the exponential moving average applied to the Kronecker factors, damping, momentum, initialization of the Kronecker factors (zero or identity matrix)
- **KFAC*:** factor of the exponential moving average applied to the Kronecker factors, damping, initialization of the Kronecker factors (zero or identity matrix)

Depending on the optimizer and experiment we use grid, random, or Bayesian search from Weights & Biases to determine the hyper-parameters. Each individual run is executed in double precision and allowed to run for a given time budget, and we rank runs by the final $L_2$ error on a fixed evaluation data set. To allow comparison, all runs are executed on RTX 6000 GPUs with 24 GiB of RAM. For grid and random searches, we use a round-based approach. First, we choose a relatively wide search space and limit to approximately 50 runs. In a second round, we narrow down the hyper-parameter space based on the first round, then re-run for another approximately 50 runs. We will release the details of all hyper-parameter search spaces, as well as the hyper-parameters for the best runs in our implementation.

### A.2    2d Poisson Equation

**Setup**    We consider a two-dimensional Poisson equation $-\Delta u(x, y) = 2\pi^2 \sin(\pi x) \sin(\pi y)$ on the unit square $(x, y) \in [0, 1]^2$ with sine product right-hand side and zero boundary conditions $u(x, y) = 0$ for $(x, y) \in \partial [0, 1]^2$. We choose a single set of training points with $N_\Omega = 900, N_{\partial\Omega} = 120$. The $L_2$ error is evaluated on a separate set of $9\,000$ data points using the known solution $u_\star(x, y) = \sin(\pi x) \sin(\pi y)$. Each run is limited to a compute time of $1\,000$ s. We compare three MLP architectures of increasing size, each of whose linear layers are Tanh-activated except for the final one: a shallow $2 \to 64 \to 1$ MLP with $D = 257$ trainable parameters, a five layer $2 \to 64 \to 64 \to 48 \to 48 \to 1$ MLP with $D = 9\,873$ trainable parameters, and a five layer $2 \to 256 \to 256 \to 128 \to 128 \to 1$ MLP with $D = 116\,097$ trainable parameters. For the biggest architecture, full and per-layer ENGD lead to out-of-memory errors and are thus not tested in the experiments. Figure A5 visualizes the results, and Figure A6 illustrates the learned solutions over training for all optimizers on the shallow MLP.

**Best run details**    The runs shown in Figure A5 correspond to the following hyper-parameters:

- $2 \to 64 \to 1$ MLP with $D = 257$
    - **SGD:** learning rate: $1.805\,015 \cdot 10^{-2}$, momentum: $9.9 \cdot 10^{-1}$
    - **Adam:** learning rate: $1.692\,339 \cdot 10^{-3}$
    - **Hessian-free:** curvature matrix: GGN, initial damping: 500, constant damping: no, maximum CG iterations: 300
    - **LBFGS:** learning rate: $5 \cdot 10^{-1}$, history size: 150
    - **ENGD (full):** damping: $1 \cdot 10^{-10}$, exponential moving average: $3 \cdot 10^{-1}$, initialize Gramian to identity: yes
    - **ENGD (layer-wise):** damping: 0, exponential moving average: $9 \cdot 10^{-1}$, initialize Gramian to identity: yes
    - **KFAC:** damping: $1.544\,099 \cdot 10^{-12}$, momentum: $5.117\,575 \cdot 10^{-1}$, exponential moving average: $4.496\,490 \cdot 10^{-1}$, initialize Kronecker factors to identity: yes
    - **KFAC*:** damping: $1.215\,640 \cdot 10^{-10}$, exponential moving average: $9.263\,314 \cdot 10^{-1}$, initialize Kronecker factors to identity: yes
- $2 \to 64 \to 64 \to 48 \to 48 \to 1$ MLP with $D = 9\,873$
    - **SGD:** learning rate: $3.758\,303 \cdot 10^{-3}$, momentum: $9 \cdot 10^{-1}$
    - **Adam:** learning rate: $2.052\,448 \cdot 10^{-4}$

(a)

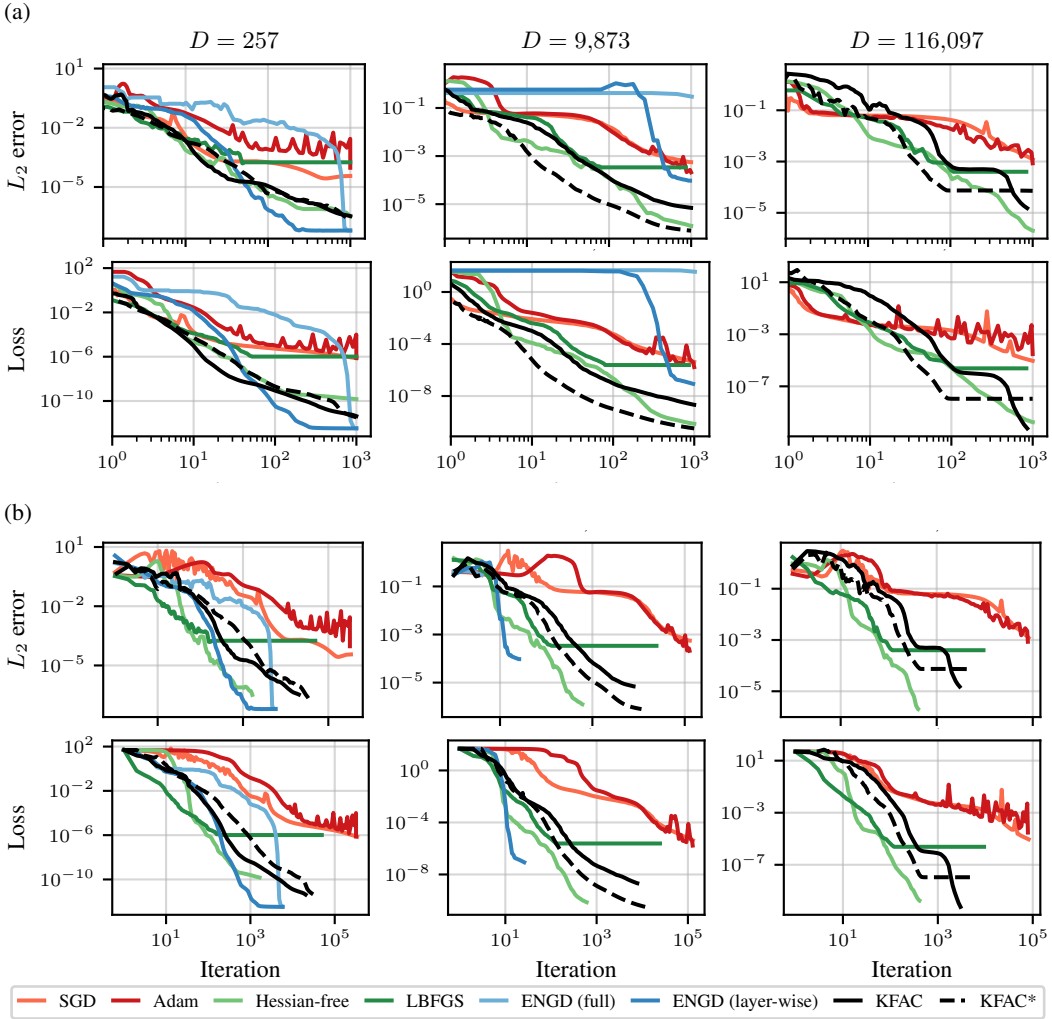

Figure A5: Training loss and evaluation $L_2$ error for learning the solution to a 2d Poisson equation over (a) time and (b) steps. Columns are different neural networks.

- **Hessian-free:** curvature matrix: GGN, initial damping: $1 \cdot 10^{-1}$, constant damping: no, maximum CG iterations: 350
- **LBFGS:** learning rate: $1 \cdot 10^{-1}$, history size: 200
- **ENGD (full):** damping: $1 \cdot 10^{-10}$, exponential moving average: $6 \cdot 10^{-1}$, initialize Gramian to identity: no
- **ENGD (layer-wise):** damping: $1 \cdot 10^{-6}$, exponential moving average: $3 \cdot 10^{-1}$, initialize Gramian to identity: no
- **KFAC:** damping: $2.640\,390 \cdot 10^{-11}$, momentum: $9.995\,595 \cdot 10^{-2}$, exponential moving average: $5.556\,664 \cdot 10^{-1}$, initialize Kronecker factors to identity: yes
- **KFAC*:** damping: $2.989\,247 \cdot 10^{-13}$, exponential moving average: $6.258\,340 \cdot 10^{-1}$, initialize Kronecker factors to identity: yes

- $2 \rightarrow 256 \rightarrow 256 \rightarrow 128 \rightarrow 128 \rightarrow 1$ MLP with $D = 116\,097$
  - **SGD:** learning rate: $2.478\,674 \cdot 10^{-3}$, momentum: $9 \cdot 10^{-1}$
  - **Adam:** learning rate: $6.406\,108 \cdot 10^{-4}$
  - **Hessian-free:** curvature matrix: GGN, initial damping: 50, constant damping: no, maximum CG iterations: 350
  - **LBFGS:** learning rate: $1 \cdot 10^{-1}$, history size: 225

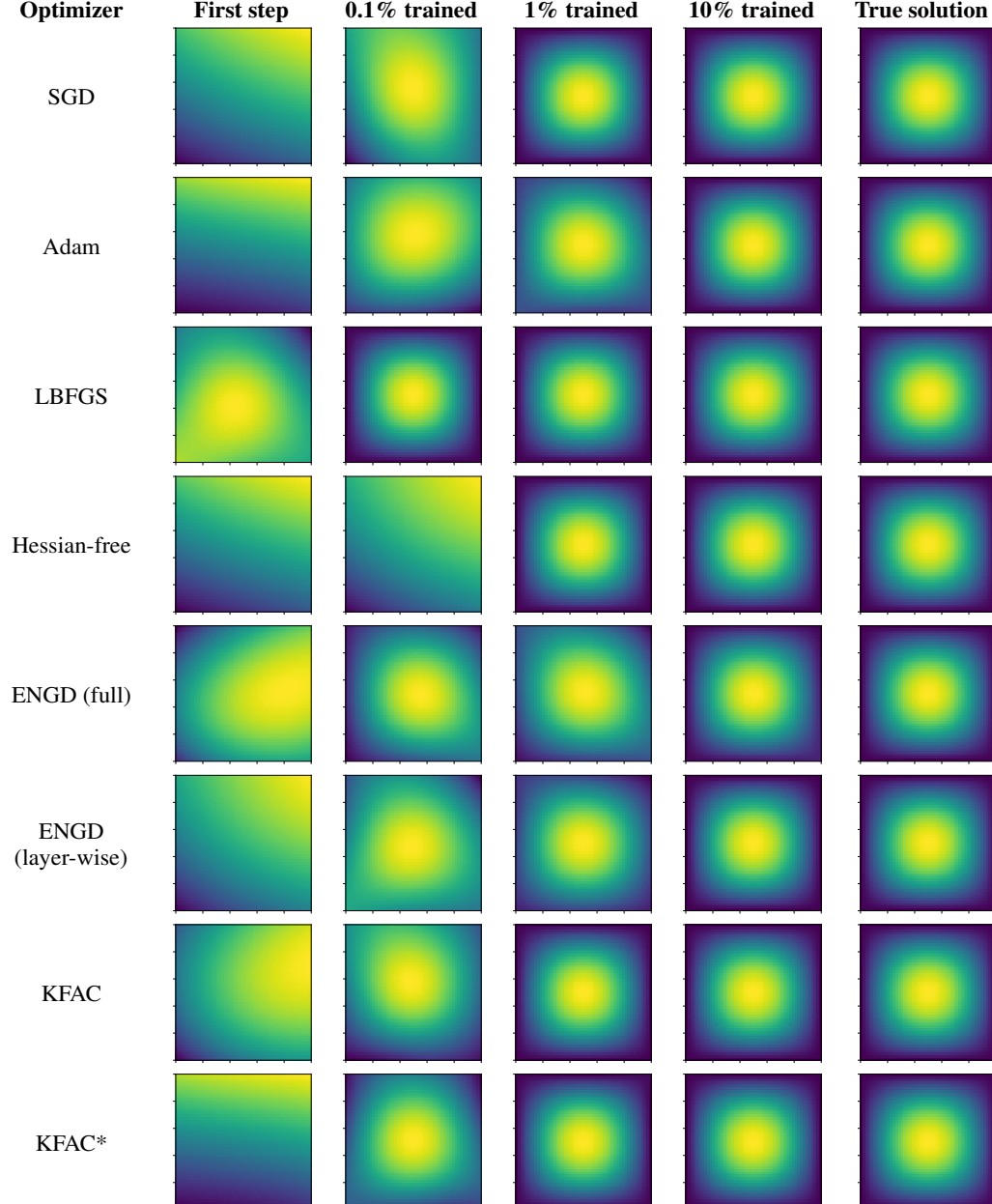

Figure A6: Visual comparison learned and true solutions while training with different optimizers for the 2d Poisson equation using a two-layer MLP (corresponding to the curves in Figure 1 left). All functions are shown on the unit square $(x, y) \in \Omega = [0; 1]^2$ and normalized to the unit interval.

- **KFAC:** damping: $1.710\,269 \cdot 10^{-13}$, momentum: $8.484\,996 \cdot 10^{-1}$, exponential moving average: $9.636\,460 \cdot 10^{-1}$, initialize Kronecker factors to identity: yes
- **KFAC*:** damping: $1.232\,407 \cdot 10^{-13}$, exponential moving average: $9.488\,207 \cdot 10^{-1}$, initialize Kronecker factors to identity: yes

**Search space details** The runs shown in Figure A5 were determined to be the best via a search with approximately 50 runs on the following search spaces which were obtained by refining an initially wider search ($\mathcal{U}$ denotes a uniform, and $\mathcal{LU}$ a log-uniform distribution):

- $2 \to 64 \to 1$ MLP with $D = 257$

- **SGD:** learning rate: $\mathcal{LU}([1 \cdot 10^{-3}; 1 \cdot 10^{-1}])$, momentum: $\mathcal{U}(\{0, 3 \cdot 10^{-1}, 6 \cdot 10^{-1}, 9 \cdot 10^{-1}, 9.9 \cdot 10^{-1}\})$
  - **Adam:** learning rate: $\mathcal{LU}([5 \cdot 10^{-4}; 1 \cdot 10^{-1}])$
  - **Hessian-free:** curvature matrix: $\mathcal{U}(\{\text{GGN}, \text{Hessian}\})$, initial damping: $\mathcal{U}(\{100, 1, 1 \cdot 10^{-2}, 1 \cdot 10^{-4}, 1 \cdot 10^{-6}\})$, constant damping: $\mathcal{U}(\{\text{no}, \text{yes}\})$, maximum CG iterations: $\mathcal{U}(\{50, 250\})$
  - **LBFGS:** learning rate: $\mathcal{U}(\{5 \cdot 10^{-1}, 2 \cdot 10^{-1}, 1 \cdot 10^{-1}, 5 \cdot 10^{-2}, 2 \cdot 10^{-2}, 1 \cdot 10^{-2}\})$, history size: $\mathcal{U}(\{75, 100, 125, 150, 175, 200, 225, 250\})$
  - **ENGD (full):** damping: $\mathcal{U}(\{1 \cdot 10^{-6}, 1 \cdot 10^{-8}, 1 \cdot 10^{-10}, 1 \cdot 10^{-12}, 0\})$, exponential moving average: $\mathcal{U}(\{0, 3 \cdot 10^{-1}, 6 \cdot 10^{-1}, 9 \cdot 10^{-1}, 9.9 \cdot 10^{-1}\})$, initialize Gramian to identity: $\mathcal{U}(\{\text{no}, \text{yes}\})$
  - **ENGD (layer-wise):** damping: $\mathcal{U}(\{1 \cdot 10^{-4}, 1 \cdot 10^{-6}, 1 \cdot 10^{-8}, 1 \cdot 10^{-10}, 0\})$, exponential moving average: $\mathcal{U}(\{0, 3 \cdot 10^{-1}, 6 \cdot 10^{-1}, 9 \cdot 10^{-1}, 9.9 \cdot 10^{-1}\})$, initialize Gramian to identity: $\mathcal{U}(\{\text{no}, \text{yes}\})$
  - **KFAC:** damping: $\mathcal{LU}([1 \cdot 10^{-13}; 1 \cdot 10^{-7}])$, momentum: $\mathcal{U}([0; 9.9 \cdot 10^{-1}])$, exponential moving average: $\mathcal{U}([0; 9.9 \cdot 10^{-1}])$, initialize Kronecker factors to identity: yes
  - **KFAC*:** damping: $\mathcal{LU}([1 \cdot 10^{-13}; 1 \cdot 10^{-7}])$, exponential moving average: $\mathcal{U}([0; 9.9 \cdot 10^{-1}])$, initialize Kronecker factors to identity: yes

- $2 \rightarrow 64 \rightarrow 64 \rightarrow 48 \rightarrow 48 \rightarrow 1$ MLP with $D = 9\,873$

  - **SGD:** learning rate: $\mathcal{LU}([1 \cdot 10^{-3}; 1 \cdot 10^{-2}])$, momentum: $\mathcal{U}(\{0, 3 \cdot 10^{-1}, 6 \cdot 10^{-1}, 9 \cdot 10^{-1}\})$
  - **Adam:** learning rate: $\mathcal{LU}([1 \cdot 10^{-4}; 5 \cdot 10^{-1}])$
  - **Hessian-free:** curvature matrix: $\mathcal{U}(\{\text{GGN}, \text{Hessian}\})$, initial damping: $\mathcal{U}(\{1, 1 \cdot 10^{-1}, 1 \cdot 10^{-2}, 1 \cdot 10^{-3}, 1 \cdot 10^{-4}\})$, constant damping: $\mathcal{U}(\{\text{no}, \text{yes}\})$, maximum CG iterations: $\mathcal{U}(\{50, 250\})$
  - **LBFGS:** learning rate: $\mathcal{U}(\{5 \cdot 10^{-1}, 2 \cdot 10^{-1}, 1 \cdot 10^{-1}, 5 \cdot 10^{-2}, 2 \cdot 10^{-2}, 1 \cdot 10^{-2}\})$, history size: $\mathcal{U}(\{50, 75, 100, 125, 150, 175, 200, 225\})$
  - **ENGD (full):** damping: $\mathcal{U}(\{1 \cdot 10^{-8}, 1 \cdot 10^{-9}, 1 \cdot 10^{-10}, 1 \cdot 10^{-11}, 1 \cdot 10^{-12}, 0\})$, exponential moving average: $\mathcal{U}(\{0, 3 \cdot 10^{-1}, 6 \cdot 10^{-1}, 9 \cdot 10^{-1}\})$, initialize Gramian to identity: $\mathcal{U}(\{\text{no}, \text{yes}\})$
  - **ENGD (layer-wise):** damping: $\mathcal{U}(\{1 \cdot 10^{-2}, 1 \cdot 10^{-3}, 1 \cdot 10^{-4}, 1 \cdot 10^{-5}, 1 \cdot 10^{-6}\})$, exponential moving average: $\mathcal{U}(\{0, 3 \cdot 10^{-1}, 6 \cdot 10^{-1}, 9 \cdot 10^{-1}, 9.9 \cdot 10^{-1}\})$, initialize Gramian to identity: $\mathcal{U}(\{\text{no}, \text{yes}\})$
  - **KFAC:** damping: $\mathcal{LU}([1 \cdot 10^{-13}; 1 \cdot 10^{-7}])$, momentum: $\mathcal{U}([0; 9.9 \cdot 10^{-1}])$, exponential moving average: $\mathcal{U}([0; 9.9 \cdot 10^{-1}])$, initialize Kronecker factors to identity: $\mathcal{U}(\{\text{no}, \text{yes}\})$
  - **KFAC*:** damping: $\mathcal{LU}([1 \cdot 10^{-13}; 1 \cdot 10^{-7}])$, exponential moving average: $\mathcal{U}([0; 9.9 \cdot 10^{-1}])$, initialize Kronecker factors to identity: $\mathcal{U}(\{\text{no}, \text{yes}\})$

- $2 \rightarrow 256 \rightarrow 256 \rightarrow 128 \rightarrow 128 \rightarrow 1$ MLP with $D = 116\,097$

  - **SGD:** learning rate: $\mathcal{LU}([1 \cdot 10^{-3}; 1 \cdot 10^{-2}])$, momentum: $\mathcal{U}(\{0, 3 \cdot 10^{-1}, 6 \cdot 10^{-1}, 9 \cdot 10^{-1}\})$
  - **Adam:** learning rate: $\mathcal{LU}([1 \cdot 10^{-4}; 5 \cdot 10^{-1}])$
  - **Hessian-free:** curvature matrix: $\mathcal{U}(\{\text{GGN}, \text{Hessian}\})$, initial damping: $\mathcal{U}(\{1, 1 \cdot 10^{-1}, 1 \cdot 10^{-2}, 1 \cdot 10^{-3}, 1 \cdot 10^{-4}\})$, constant damping: $\mathcal{U}(\{\text{no}, \text{yes}\})$, maximum CG iterations: $\mathcal{U}(\{50, 250\})$
  - **LBFGS:** learning rate: $\mathcal{U}(\{5 \cdot 10^{-1}, 2 \cdot 10^{-1}, 1 \cdot 10^{-1}, 5 \cdot 10^{-2}, 2 \cdot 10^{-2}, 1 \cdot 10^{-2}\})$, history size: $\mathcal{U}(\{50, 75, 100, 125, 150, 175, 200, 225\})$
  - **KFAC:** damping: $\mathcal{LU}([1 \cdot 10^{-13}; 1 \cdot 10^{-7}])$, momentum: $\mathcal{U}([0; 9.9 \cdot 10^{-1}])$, exponential moving average: $\mathcal{U}([0; 9.9 \cdot 10^{-1}])$, initialize Kronecker factors to identity: $\mathcal{U}(\{\text{no}, \text{yes}\})$
  - **KFAC*:** damping: $\mathcal{LU}([1 \cdot 10^{-13}; 1 \cdot 10^{-7}])$, exponential moving average: $\mathcal{U}([0; 9.9 \cdot 10^{-1}])$, initialize Kronecker factors to identity: $\mathcal{U}(\{\text{no}, \text{yes}\})$

## A.3  5d Poisson Equation

**Setup**   We consider a five-dimensional Poisson equation $-\Delta u(\boldsymbol{x}) = \pi^2 \sum_{i=1}^{5} \cos(\pi x_i)$ on the five-dimensional unit square $\boldsymbol{x} \in [0,1]^5$ with cosine sum right-hand side and boundary conditions $u(\boldsymbol{x}) = \sum_{i=1}^{5} \cos(\pi x_i)$ for $\boldsymbol{x} \in \partial[0,1]^5$. We sample training batches of size $N_\Omega = 3\,000, N_{\partial\Omega} = 500$ and evaluate the $L_2$ error on a separate set of $30\,000$ data points using the known solution $u_\star(\boldsymbol{x}) = \sum_{i=1}^{5} \cos(\pi x_i)$. All optimizers except for KFAC sample a new training batch each iteration. KFAC only re-samples every 100 iterations because we noticed significant improvement with multiple iterations on a fixed batch. To make sure that this does not lead to an unfair advantage of KFAC, we conduct an additional experiment where we also tune the batch sampling frequency, as well as other hyper-parameters; see §A.5. The results presented in this section are consistent with this additional experiment (compare the rightmost column of Figure A7 and the leftmost column of Figure A9). Each run is limited to 3000 s. We compare three MLP architectures of increasing size, each of whose linear layers are Tanh-activated except for the final one: a shallow $5 \to 64 \to 1$ MLP with $D = 449$ trainable parameters, a five layer $5 \to 64 \to 64 \to 48 \to 48 \to 1$ MLP with $D = 10\,065$ trainable parameters, and a five layer $5 \to 256 \to 256 \to 128 \to 128 \to 1$ MLP with $D = 116\,864$ trainable parameters. For the biggest architecture, full and layer-wise ENGD lead to out-of-memory errors and are thus not tested in the experiments. Figure A7 visualizes the results.

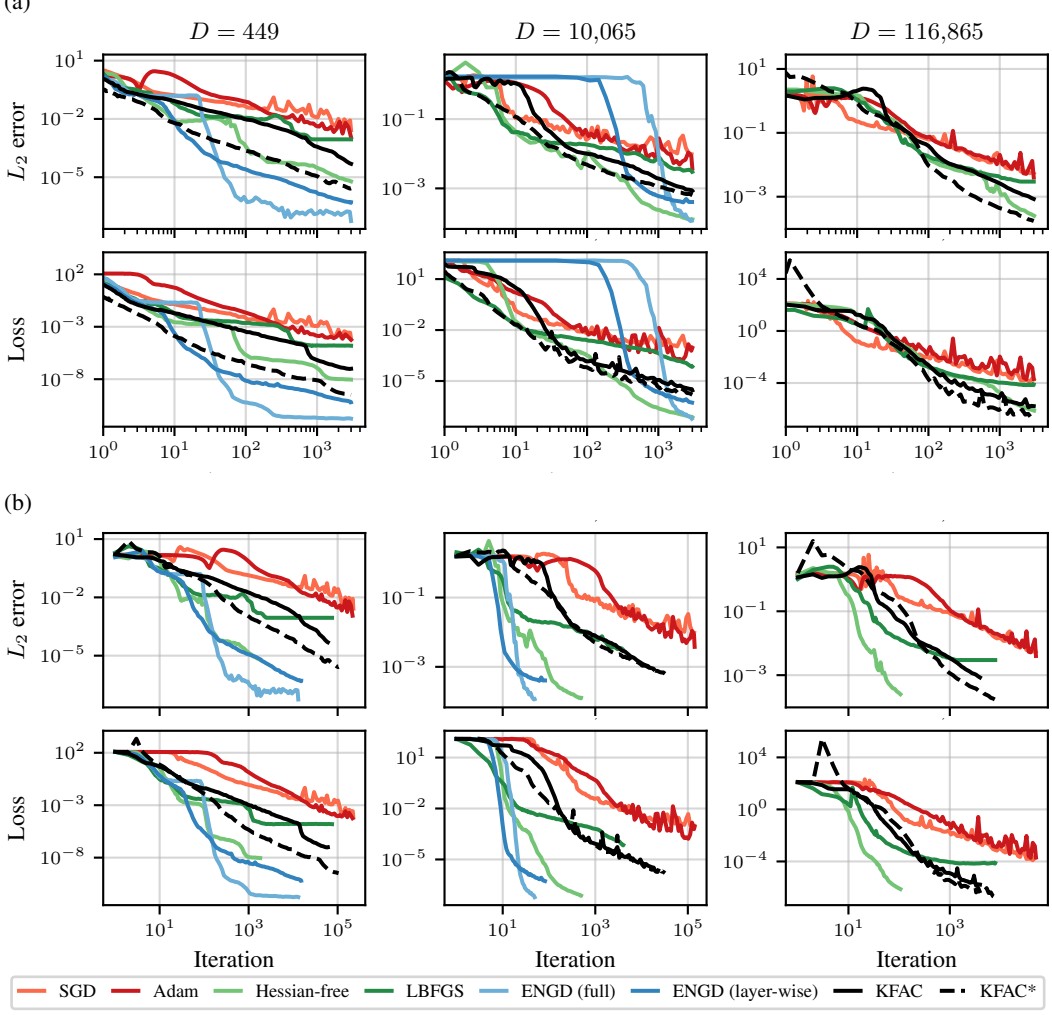

Figure A7: Training loss and evaluation $L_2$ error for learning the solution to a 5d Poisson equation over (a) time and (b) steps. Columns are different neural networks.

**Best run details** The runs shown in Figure A7 correspond to the following hyper-parameters:

- $5 \to 64 \to 1$ MLP with $D = 449$
  - **SGD:** learning rate: $4.829\,757 \cdot 10^{-3}$, momentum: $9 \cdot 10^{-1}$
  - **Adam:** learning rate: $1.527\,101 \cdot 10^{-3}$
  - **Hessian-free:** curvature matrix: GGN, initial damping: 5, constant damping: no, maximum CG iterations: 350
  - **LBFGS:** learning rate: $5 \cdot 10^{-2}$, history size: 125
  - **ENGD (full):** damping: $1 \cdot 10^{-11}$, exponential moving average: $6 \cdot 10^{-1}$, initialize Gramian to identity: yes
  - **ENGD (layer-wise):** damping: $1 \cdot 10^{-6}$, exponential moving average: $6 \cdot 10^{-1}$, initialize Gramian to identity: yes
  - **KFAC:** damping: $3.030\,734 \cdot 10^{-13}$, momentum: $4.410\,155 \cdot 10^{-1}$, exponential moving average: $3.260\,663 \cdot 10^{-1}$, initialize Kronecker factors to identity: yes
  - **KFAC*:** damping: $9.835\,853 \cdot 10^{-13}$, exponential moving average: $7.714\,287 \cdot 10^{-1}$, initialize Kronecker factors to identity: yes

- $5 \to 64 \to 64 \to 48 \to 48 \to 1$ MLP with $D = 10\,065$
  - **SGD:** learning rate: $1.007\,555 \cdot 10^{-3}$, momentum: $9 \cdot 10^{-1}$
  - **Adam:** learning rate: $6.999\,994 \cdot 10^{-4}$
  - **Hessian-free:** curvature matrix: GGN, initial damping: $1 \cdot 10^{-1}$, constant damping: no, maximum CG iterations: 350
  - **LBFGS:** learning rate: $1 \cdot 10^{-1}$, history size: 225
  - **ENGD (full):** damping: $1 \cdot 10^{-8}$, exponential moving average: $3 \cdot 10^{-1}$, initialize Gramian to identity: no
  - **ENGD (layer-wise):** damping: $1 \cdot 10^{-6}$, exponential moving average: $3 \cdot 10^{-1}$, initialize Gramian to identity: no
  - **KFAC:** damping: $1.183\,063 \cdot 10^{-12}$, momentum: $9.058\,900 \cdot 10^{-1}$, exponential moving average: $9.588\,846 \cdot 10^{-1}$, initialize Kronecker factors to identity: yes
  - **KFAC*:** damping: $2.829\,461 \cdot 10^{-10}$, exponential moving average: $9.001\,393 \cdot 10^{-1}$, initialize Kronecker factors to identity: yes

- $5 \to 256 \to 256 \to 128 \to 128 \to 1$ MLP with $D = 116\,865$
  - **SGD:** learning rate: $1.924\,173 \cdot 10^{-3}$, momentum: $9 \cdot 10^{-1}$
  - **Adam:** learning rate: $5.416\,376 \cdot 10^{-4}$
  - **Hessian-free:** curvature matrix: GGN, initial damping: 5, constant damping: no, maximum CG iterations: 350
  - **LBFGS:** learning rate: $2 \cdot 10^{-2}$, history size: 225
  - **KFAC:** damping: $1.844\,213 \cdot 10^{-11}$, momentum: $7.528\,559 \cdot 10^{-1}$, exponential moving average: $9.307\,849 \cdot 10^{-1}$, initialize Kronecker factors to identity: yes
  - **KFAC*:** damping: $2.183\,605 \cdot 10^{-12}$, exponential moving average: $9.563\,992 \cdot 10^{-1}$, initialize Kronecker factors to identity: yes

**Search space details** The runs shown in Figure A7 were determined to be the best via a search with approximately 50 runs on the following search spaces which were obtained by refining an initially wider search ($\mathcal{U}$ denotes a uniform, and $\mathcal{LU}$ a log-uniform distribution):

- $5 \to 64 \to 1$ MLP with $D = 449$
  - **SGD:** learning rate: $\mathcal{LU}([1 \cdot 10^{-3}; 1 \cdot 10^{-2}])$, momentum: $\mathcal{U}(\{0, 3 \cdot 10^{-1}, 6 \cdot 10^{-1}, 9 \cdot 10^{-1}\})$
  - **Adam:** learning rate: $\mathcal{LU}([1 \cdot 10^{-4}; 5 \cdot 10^{-1}])$
  - **Hessian-free:** curvature matrix: $\mathcal{U}(\{\text{GGN}, \text{Hessian}\})$, initial damping: $\mathcal{U}(\{1, 1 \cdot 10^{-1}, 1 \cdot 10^{-2}, 1 \cdot 10^{-3}, 1 \cdot 10^{-4}\})$, constant damping: $\mathcal{U}(\{\text{no}, \text{yes}\})$, maximum CG iterations: $\mathcal{U}(\{50, 250\})$
  - **LBFGS:** learning rate: $\mathcal{U}(\{5 \cdot 10^{-1}, 2 \cdot 10^{-1}, 1 \cdot 10^{-1}, 5 \cdot 10^{-2}, 2 \cdot 10^{-2}, 1 \cdot 10^{-2}\})$, history size: $\mathcal{U}(\{50, 75, 100, 125, 150, 175, 200, 225\})$

- **ENGD (full):** damping: $\mathcal{U}(\{1 \cdot 10^{-8}, 1 \cdot 10^{-9}, 1 \cdot 10^{-10}, 1 \cdot 10^{-11}, 1 \cdot 10^{-12}, 0\})$, exponential moving average: $\mathcal{U}(\{0, 3 \cdot 10^{-1}, 6 \cdot 10^{-1}, 9 \cdot 10^{-1}\})$, initialize Gramian to identity: $\mathcal{U}(\{\text{no}, \text{yes}\})$
- **ENGD (layer-wise):** damping: $\mathcal{U}(\{1 \cdot 10^{-2}, 1 \cdot 10^{-3}, 1 \cdot 10^{-4}, 1 \cdot 10^{-5}, 1 \cdot 10^{-6}\})$, exponential moving average: $\mathcal{U}(\{0, 3 \cdot 10^{-1}, 6 \cdot 10^{-1}, 9 \cdot 10^{-1}, 9.9 \cdot 10^{-1}\})$, initialize Gramian to identity: $\mathcal{U}(\{\text{no}, \text{yes}\})$
- **KFAC:** damping: $\mathcal{LU}([1 \cdot 10^{-13}; 1 \cdot 10^{-7}])$, momentum: $\mathcal{U}([0; 9.9 \cdot 10^{-1}])$, exponential moving average: $\mathcal{U}([0; 9.9 \cdot 10^{-1}])$, initialize Kronecker factors to identity: yes
- **KFAC*:** damping: $\mathcal{LU}([1 \cdot 10^{-13}; 1 \cdot 10^{-7}])$, exponential moving average: $\mathcal{U}([0; 9.9 \cdot 10^{-1}])$, initialize Kronecker factors to identity: yes

- $5 \rightarrow 64 \rightarrow 64 \rightarrow 48 \rightarrow 48 \rightarrow 1$ MLP with $D = 10\,065$
  - **SGD:** learning rate: $\mathcal{LU}([1 \cdot 10^{-3}; 1 \cdot 10^{-2}])$, momentum: $\mathcal{U}(\{0, 3 \cdot 10^{-1}, 6 \cdot 10^{-1}, 9 \cdot 10^{-1}\})$
  - **Adam:** learning rate: $\mathcal{LU}([1 \cdot 10^{-4}; 5 \cdot 10^{-1}])$
  - **Hessian-free:** curvature matrix: $\mathcal{U}(\{\text{GGN}, \text{Hessian}\})$, initial damping: $\mathcal{U}(\{1, 1 \cdot 10^{-1}, 1 \cdot 10^{-2}, 1 \cdot 10^{-3}, 1 \cdot 10^{-4}\})$, constant damping: $\mathcal{U}(\{\text{no}, \text{yes}\})$, maximum CG iterations: $\mathcal{U}(\{50, 250\})$
  - **LBFGS:** learning rate: $\mathcal{U}(\{5 \cdot 10^{-1}, 2 \cdot 10^{-1}, 1 \cdot 10^{-1}, 5 \cdot 10^{-2}, 2 \cdot 10^{-2}, 1 \cdot 10^{-2}\})$, history size: $\mathcal{U}(\{50, 75, 100, 125, 150, 175, 200, 225\})$
  - **ENGD (full):** damping: $\mathcal{U}(\{1 \cdot 10^{-8}, 1 \cdot 10^{-9}, 1 \cdot 10^{-10}, 1 \cdot 10^{-11}, 1 \cdot 10^{-12}, 0\})$, exponential moving average: $\mathcal{U}(\{0, 3 \cdot 10^{-1}, 6 \cdot 10^{-1}, 9 \cdot 10^{-1}\})$, initialize Gramian to identity: $\mathcal{U}(\{\text{no}, \text{yes}\})$
  - **ENGD (layer-wise):** damping: $\mathcal{U}(\{1 \cdot 10^{-2}, 1 \cdot 10^{-3}, 1 \cdot 10^{-4}, 1 \cdot 10^{-5}, 1 \cdot 10^{-6}\})$, exponential moving average: $\mathcal{U}(\{0, 3 \cdot 10^{-1}, 6 \cdot 10^{-1}, 9 \cdot 10^{-1}, 9.9 \cdot 10^{-1}\})$, initialize Gramian to identity: $\mathcal{U}(\{\text{no}, \text{yes}\})$
  - **KFAC:** damping: $\mathcal{LU}([1 \cdot 10^{-13}; 1 \cdot 10^{-7}])$, momentum: $\mathcal{U}([0; 9.9 \cdot 10^{-1}])$, exponential moving average: $\mathcal{U}([0; 9.9 \cdot 10^{-1}])$, initialize Kronecker factors to identity: yes
  - **KFAC*:** damping: $\mathcal{LU}([1 \cdot 10^{-12}; 1 \cdot 10^{-6}])$, exponential moving average: $\mathcal{U}([0; 9.9 \cdot 10^{-1}])$, initialize Kronecker factors to identity: yes

- $5 \rightarrow 256 \rightarrow 256 \rightarrow 128 \rightarrow 128 \rightarrow 1$ MLP with $D = 116\,865$
  - **SGD:** learning rate: $\mathcal{LU}([1 \cdot 10^{-3}; 1 \cdot 10^{-2}])$, momentum: $\mathcal{U}(\{0, 3 \cdot 10^{-1}, 6 \cdot 10^{-1}, 9 \cdot 10^{-1}\})$
  - **Adam:** learning rate: $\mathcal{LU}([1 \cdot 10^{-4}; 5 \cdot 10^{-1}])$
  - **Hessian-free:** curvature matrix: $\mathcal{U}(\{\text{GGN}, \text{Hessian}\})$, initial damping: $\mathcal{U}(\{1, 1 \cdot 10^{-1}, 1 \cdot 10^{-2}, 1 \cdot 10^{-3}, 1 \cdot 10^{-4}\})$, constant damping: $\mathcal{U}(\{\text{no}, \text{yes}\})$, maximum CG iterations: $\mathcal{U}(\{50, 250\})$
  - **LBFGS:** learning rate: $\mathcal{U}(\{5 \cdot 10^{-1}, 2 \cdot 10^{-1}, 1 \cdot 10^{-1}, 5 \cdot 10^{-2}, 2 \cdot 10^{-2}, 1 \cdot 10^{-2}\})$, history size: $\mathcal{U}(\{50, 75, 100, 125, 150, 175, 200, 225\})$
  - **KFAC:** damping: $\mathcal{LU}([1 \cdot 10^{-13}; 1 \cdot 10^{-7}])$, momentum: $\mathcal{U}([0; 9.9 \cdot 10^{-1}])$, exponential moving average: $\mathcal{U}([0; 9.9 \cdot 10^{-1}])$, initialize Kronecker factors to identity: yes
  - **KFAC*:** damping: $\mathcal{LU}([1 \cdot 10^{-12}; 1 \cdot 10^{-6}])$, exponential moving average: $\mathcal{U}([0; 9.9 \cdot 10^{-1}])$, initialize Kronecker factors to identity: yes

### A.4 10d Poisson Equation

**Setup** We consider a 10-dimensional Poisson equation $-\Delta u(\boldsymbol{x}) = 0$ on the 10-dimensional unit square $\boldsymbol{x} \in [0, 1]^5$ with zero right-hand side and harmonic mixed second order polynomial boundary conditions $u(\boldsymbol{x}) = \sum_{i=1}^{d/2} x_{2i-1} x_{2i}$ for $\boldsymbol{x} \in \partial[0, 1]^d$. We sample training batches of size $N_\Omega = 3\,000, N_{\partial\Omega} = 1000$ and evaluate the $L_2$ error on a separate set of $30\,000$ data points using the known solution $u_\star(\boldsymbol{x}) = \sum_{i=1}^{d/2} x_{2i-1} x_{2i}$. All optimizers except for KFAC sample a new training batch each iteration. KFAC only re-samples every 100 iterations because we noticed significant improvement with multiple iterations on a fixed batch. Each run is limited to $6\,000$ s. We use a $10 \rightarrow 256 \rightarrow 256 \rightarrow 128 \rightarrow 128 \rightarrow 1$ MLP with $D = 118\,145$ MLP whose linear layers are Tanh-activated except for the final one. Figure A8 visualizes the results.

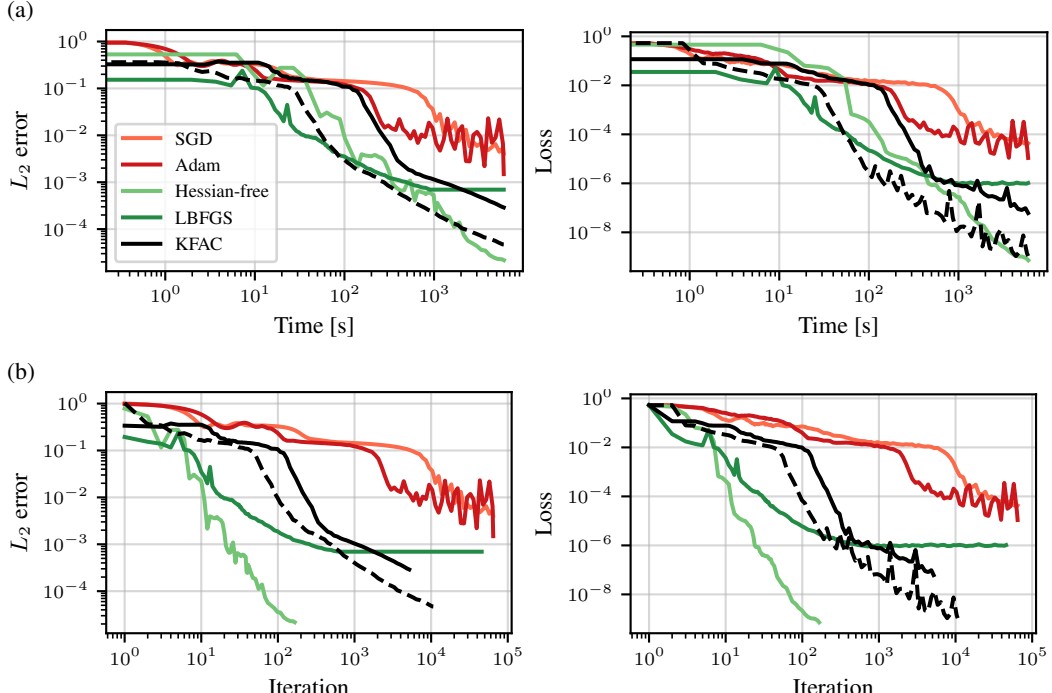

(a)

(b)

Figure A8: Training loss and evaluation $L_2$ error for learning the solution to a 10d Poisson equation over (a) time and (b) steps.

**Best run details**   The runs shown in Figure A8 correspond to the following hyper-parameters:

- **SGD:** learning rate: $6.550\,109 \cdot 10^{-3}$, momentum: $9 \cdot 10^{-1}$
- **Adam:** learning rate: $1.359\,480 \cdot 10^{-4}$
- **Hessian-free:** curvature matrix: GGN, initial damping: $1 \cdot 10^{-3}$, constant damping: no, maximum CG iterations: 250
- **LBFGS:** learning rate: $2 \cdot 10^{-1}$, history size: 200
- **KFAC:** damping: $1.056\,857 \cdot 10^{-13}$, momentum: $7.160\,131 \cdot 10^{-1}$, exponential moving average: $9.622\,372 \cdot 10^{-1}$, initialize Kronecker factors to identity: yes
- **KFAC*:** damping: $7.978\,934 \cdot 10^{-11}$, exponential moving average: $8.950\,193 \cdot 10^{-1}$, initialize Kronecker factors to identity: yes

**Search space details**   The runs shown in Figure A8 were determined to be the best via a Bayesian search on the following search spaces which each optimizer given approximately the same total computational time ($\mathcal{U}$ denotes a uniform, and $\mathcal{LU}$ a log-uniform distribution):

- **SGD:** learning rate: $\mathcal{LU}([1 \cdot 10^{-3}; 1 \cdot 10^{-2}])$, momentum: $\mathcal{U}(\{0, 3 \cdot 10^{-1}, 6 \cdot 10^{-1}, 9 \cdot 10^{-1}\})$
- **Adam:** learning rate: $\mathcal{LU}([5\text{e-}05; 5 \cdot 10^{-3}])$
- **Hessian-free:** curvature matrix: $\mathcal{U}(\{\text{GGN}, \text{Hessian}\})$, initial damping: $\mathcal{U}(\{1, 1 \cdot 10^{-1}, 1 \cdot 10^{-2}, 1 \cdot 10^{-3}, 1 \cdot 10^{-4}\})$, constant damping: $\mathcal{U}(\{\text{no}, \text{yes}\})$, maximum CG iterations: $\mathcal{U}(\{50, 250\})$
- **LBFGS:** learning rate: $\mathcal{U}(\{5 \cdot 10^{-1}, 2 \cdot 10^{-1}, 1 \cdot 10^{-1}, 5 \cdot 10^{-2}, 2 \cdot 10^{-2}, 1 \cdot 10^{-2}\})$, history size: $\mathcal{U}(\{50, 75, 100, 125, 150, 175, 200, 225\})$
- **KFAC:** damping: $\mathcal{LU}([1 \cdot 10^{-14}; 1 \cdot 10^{-8}])$, momentum: $\mathcal{U}([0; 9.9 \cdot 10^{-1}])$, exponential moving average: $\mathcal{U}([5 \cdot 10^{-1}; 9.9 \cdot 10^{-1}])$, initialize Kronecker factors to identity: yes
- **KFAC*:** damping: $\mathcal{LU}([1 \cdot 10^{-12}; 1 \cdot 10^{-6}])$, exponential moving average: $\mathcal{U}([0; 9.9 \cdot 10^{-1}])$, initialize Kronecker factors to identity: yes

## A.5  5/10/100-d Poisson Equations with Bayesian Search

**Setup**   Here, we consider three Poisson equations $-\Delta u(\boldsymbol{x}) = f(\boldsymbol{x})$ with different right-hand sides and boundary conditions on the unit square $\boldsymbol{x} \in [0, 1]^d$:

- $d = 5$ with cosine sum right-hand side $f(\boldsymbol{x}) = \pi^2 \sum_{i=1}^{d} \cos(\pi x_i)$, boundary conditions $u(\boldsymbol{x}) = \sum_{i=1}^{d} \cos(\pi x_i)$ for $\boldsymbol{x} \in \partial[0, 1]^d$, and known solution $u_\star(\boldsymbol{x}) = \sum_{i=1}^{d} \cos(\pi x_i)$. We assign each run a budget of $3\,000$ s.
- $d = 10$ with zero right-hand side $f(\boldsymbol{x}) = 0$, harmonic mixed second order polynomial boundary conditions $u(\boldsymbol{x}) = \sum_{i=1}^{d/2} x_{2i-1} x_{2i}$ for $\boldsymbol{x} \in \partial[0, 1]^d$, and known solution $u_\star(\boldsymbol{x}) = \sum_{i=1}^{d/2} x_{2i-1} x_{2i}$. We assign each run a budget of $6\,000$ s.
- $d = 100$ with constant non-zero right-hand side $f(\boldsymbol{x}) = -2d$, square norm boundary conditions $u(\boldsymbol{x}) = \|\boldsymbol{x}\|_2^2$ for $\boldsymbol{x} \in \partial[0, 1]^d$, and known solution $u_\star(\boldsymbol{x}) = \|\boldsymbol{x}\|_2^2$. We assign each run a budget of $10\,000$ s.

We tune the optimizer-hyperparameters described in §A.1, as well as the batch sizes $N_\Omega$, $N_{\partial\Omega}$, and their associated re-sampling frequencies using Bayesian search. We use five layer MLP architectures with varying widths whose layers are Tanh-activated except for the final layer. These architectures are too large to be optimized by ENGD. Figure A9 visualizes the results.

**Best run details**   The runs shown in Figure A9 correspond to the following hyper-parameters:

- 5d Poisson equation, $5 \to 256 \to 256 \to 128 \to 128 \to 1$ MLP with $D = 116\,865$
  - **SGD:** learning rate: $2.686\,653 \cdot 10^{-4}$, momentum: $9.878\,243 \cdot 10^{-1}$, $N_\Omega$: 606, $N_{\partial\Omega}$: $2\,001$, batch sampling frequency: $1\,570$
  - **Adam:** learning rate: $6.111\,767 \cdot 10^{-5}$, $N_\Omega$: 534, $N_{\partial\Omega}$: $1\,021$, batch sampling frequency: $8\,220$
  - **Hessian-free:** curvature matrix: GGN, initial damping: $1.541\,954 \cdot 10^{1}$, constant damping: no, maximum CG iterations: 358, $N_\Omega$: $1\,084$, $N_{\partial\Omega}$: $3\,837$, batch sampling frequency: 230
  - **LBFGS:** learning rate: $1.749\,124 \cdot 10^{-1}$, history size: 339, $N_\Omega$: $5\,391$, $N_{\partial\Omega}$: $4\,768$, batch sampling frequency: 155
  - **KFAC:** damping: $4.251\,462 \cdot 10^{-10}$, momentum: $9.198\,986 \cdot 10^{-1}$, exponential moving average: $9.737\,093 \cdot 10^{-1}$, initialize Kronecker factors to identity: yes, $N_\Omega$: $4\,690$, $N_{\partial\Omega}$: $2\,708$, batch sampling frequency: $2\,369$
  - **KFAC*:** damping: $2.240\,865 \cdot 10^{-12}$, exponential moving average: $8.522\,194 \cdot 10^{-1}$, initialize Kronecker factors to identity: yes, $N_\Omega$: $3\,149$, $N_{\partial\Omega}$: $3\,801$, batch sampling frequency: $1\,393$
- 10d Poisson equation, $10 \to 256 \to 256 \to 128 \to 128 \to 1$ MLP with $D = 118\,145$
  - **SGD:** learning rate: $5.805\,516 \cdot 10^{-2}$, momentum: $9.715\,522 \cdot 10^{-1}$, $N_\Omega$: 537, $N_{\partial\Omega}$: $1\,173$, batch sampling frequency: $1\,083$
  - **Adam:** learning rate: $1.337\,679 \cdot 10^{-4}$, $N_\Omega$: 115, $N_{\partial\Omega}$: $1\,960$, batch sampling frequency: $4\,975$
  - **Hessian-free:** curvature matrix: GGN, initial damping: $8.963\,629 \cdot 10^{-1}$, constant damping: no, maximum CG iterations: 143, $N_\Omega$: $3\,736$, $N_{\partial\Omega}$: 961, batch sampling frequency: 3
  - **LBFGS:** learning rate: $1.695\,334 \cdot 10^{-1}$, history size: 338, $N_\Omega$: 342, $N_{\partial\Omega}$: 765, batch sampling frequency: 845
  - **KFAC:** damping: $6.575\,415 \cdot 10^{-4}$, momentum: $9.772\,500 \cdot 10^{-1}$, exponential moving average: $2.745\,481 \cdot 10^{-1}$, initialize Kronecker factors to identity: yes, $N_\Omega$: $1\,284$, $N_{\partial\Omega}$: $2\,258$, batch sampling frequency: 455
  - **KFAC*:** damping: $7.530\,350 \cdot 10^{-12}$, exponential moving average: $9.648\,138 \cdot 10^{-1}$, initialize Kronecker factors to identity: yes, $N_\Omega$: $1\,090$, $N_{\partial\Omega}$: $1\,930$, batch sampling frequency: $2\,454$
- 100d Poisson equation, $100 \to 768 \to 768 \to 512 \to 512 \to 1$ MLP with $D = 1\,325\,057$

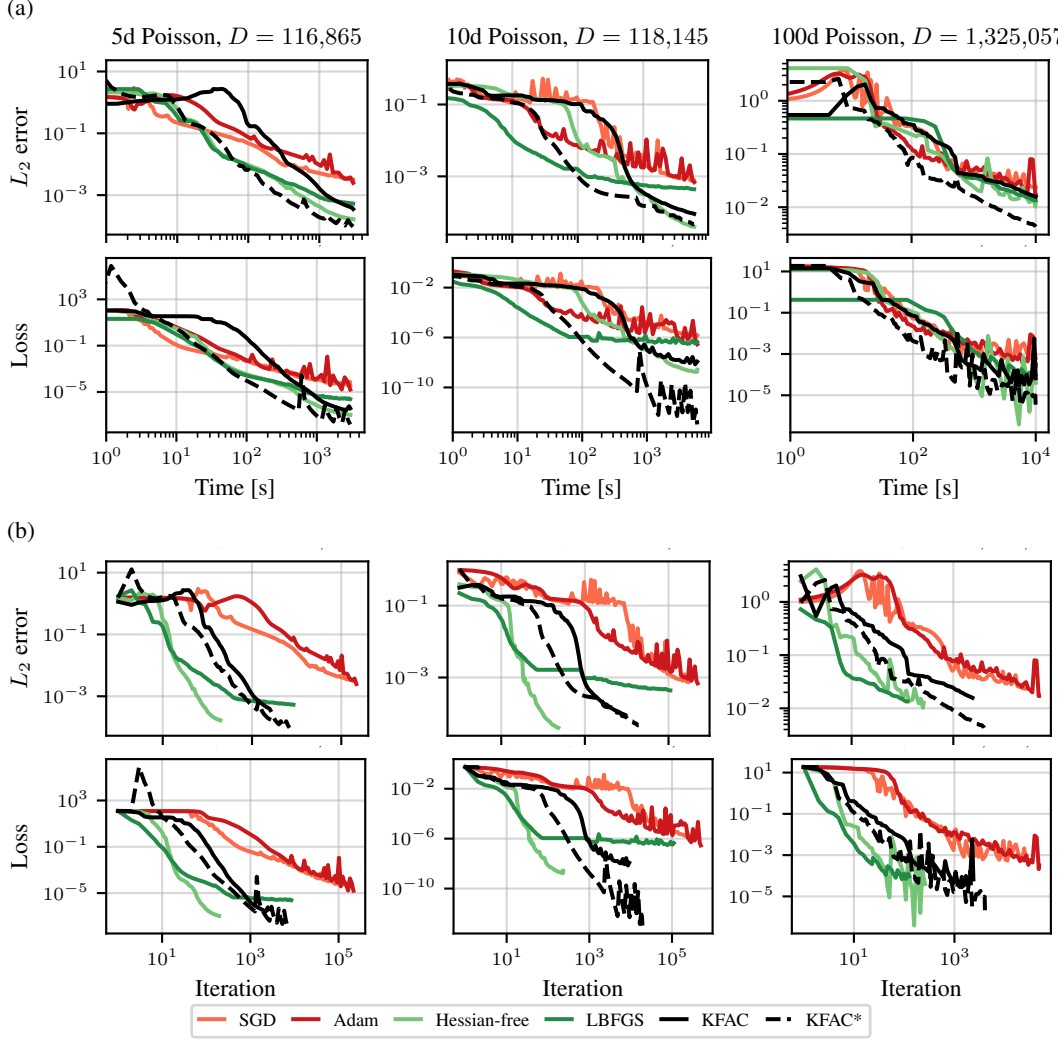

Figure A9: Training loss and evaluation $L_2$ error for learning the solution to high-dimensional Poisson equations over (a) time and (b) steps using Bayesian search.

- **SGD:** learning rate: $1.450\,764 \cdot 10^{-3}$, momentum: $9.747\,671 \cdot 10^{-1}$, $N_\Omega$: 177, $N_{\partial\Omega}$: $2\,422$, batch sampling frequency: 519

- **Adam:** learning rate: $7.894\,685 \cdot 10^{-5}$, $N_\Omega$: 100, $N_{\partial\Omega}$: 601, batch sampling frequency: 19

- **Hessian-free:** curvature matrix: GGN, initial damping: $7.705\,318 \cdot 10^{-4}$, constant damping: no, maximum CG iterations: 263, $N_\Omega$: 108, $N_{\partial\Omega}$: $2\,372$, batch sampling frequency: 55

- **LBFGS:** learning rate: $1.797\,096 \cdot 10^{-1}$, history size: 112, $N_\Omega$: $2\,115$, $N_{\partial\Omega}$: $1\,852$, batch sampling frequency: 23

- **KFAC:** damping: $9.724\,117 \cdot 10^{-3}$, momentum: $5.015\,715 \cdot 10^{-1}$, exponential moving average: $9.200\,952 \cdot 10^{-1}$, initialize Kronecker factors to identity: yes, $N_\Omega$: 124, $N_{\partial\Omega}$: $2\,332$, batch sampling frequency: 322

- **KFAC*:** damping: $1.236\,763 \cdot 10^{-7}$, exponential moving average: $8.302\,663 \cdot 10^{-1}$, initialize Kronecker factors to identity: yes, $N_\Omega$: 175, $N_{\partial\Omega}$: $2\,086$, batch sampling frequency: 16

**Search space details** The runs shown in Figure A9 were determined to be the best via a Bayesian search on the following search spaces which each optimizer given approximately the same total computational time ($\mathcal{U}$ denotes a uniform, and $\mathcal{LU}$ a log-uniform distribution):

- 5d Poisson equation, $5 \rightarrow 256 \rightarrow 256 \rightarrow 128 \rightarrow 128 \rightarrow 1$ MLP with $D = 116\,865$
  - **SGD:** learning rate: $\mathcal{LU}([1 \cdot 10^{-6}; 1])$, momentum: $\mathcal{U}([0; 9.9 \cdot 10^{-1}])$, $N_\Omega$: $\mathcal{U}(\{100, 101, \ldots, 10\,000\})$, $N_{\partial\Omega}$: $\mathcal{U}(\{50, 51, \ldots, 5\,000\})$, batch sampling frequency: $\mathcal{U}(\{0, 1, \ldots, 10\,000\})$
  - **Adam:** learning rate: $\mathcal{LU}([1 \cdot 10^{-6}; 1])$, $N_\Omega$: $\mathcal{U}(\{100, 101, \ldots, 10\,000\})$, $N_{\partial\Omega}$: $\mathcal{U}(\{50, 51, \ldots, 5\,000\})$, batch sampling frequency: $\mathcal{U}(\{0, 1, \ldots, 10\,000\})$
  - **Hessian-free:** curvature matrix: $\mathcal{U}(\{\text{GGN}, \text{Hessian}\})$, initial damping: $\mathcal{LU}([1 \cdot 10^{-15}; 1])$, constant damping: $\mathcal{U}(\{\text{no}, \text{yes}\})$, maximum CG iterations: $\mathcal{U}(\{1, 2, \ldots, 500\})$, $N_\Omega$: $\mathcal{U}(\{100, 101, \ldots, 10\,000\})$, $N_{\partial\Omega}$: $\mathcal{U}(\{50, 51, \ldots, 5\,000\})$, batch sampling frequency: $\mathcal{U}(\{0, 1, \ldots, 10\,000\})$
  - **LBFGS:** learning rate: $\mathcal{LU}([1 \cdot 10^{-6}; 1])$, history size: $\mathcal{U}(\{5, 6, \ldots, 500\})$, $N_\Omega$: $\mathcal{U}(\{100, 101, \ldots, 10\,000\})$, $N_{\partial\Omega}$: $\mathcal{U}(\{50, 51, \ldots, 5\,000\})$, batch sampling frequency: $\mathcal{U}(\{0, 1, \ldots, 10\,000\})$
  - **KFAC:** damping: $\mathcal{LU}([1 \cdot 10^{-15}; 1 \cdot 10^{-2}])$, momentum: $\mathcal{U}([0; 9.9 \cdot 10^{-1}])$, exponential moving average: $\mathcal{U}([0; 9.9 \cdot 10^{-1}])$, initialize Kronecker factors to identity: $\mathcal{U}(\{\text{no}, \text{yes}\})$, $N_\Omega$: $\mathcal{U}(\{100, 101, \ldots, 10\,000\})$, $N_{\partial\Omega}$: $\mathcal{U}(\{50, 51, \ldots, 5\,000\})$, batch sampling frequency: $\mathcal{U}(\{0, 1, \ldots, 10\,000\})$
  - **KFAC*:** damping: $\mathcal{LU}([1 \cdot 10^{-15}; 1 \cdot 10^{-2}])$, exponential moving average: $\mathcal{U}([0; 9.9 \cdot 10^{-1}])$, initialize Kronecker factors to identity: $\mathcal{U}(\{\text{no}, \text{yes}\})$, $N_\Omega$: $\mathcal{U}(\{100, 101, \ldots, 10\,000\})$, $N_{\partial\Omega}$: $\mathcal{U}(\{50, 51, \ldots, 5\,000\})$, batch sampling frequency: $\mathcal{U}(\{0, 1, \ldots, 10\,000\})$

- 10d Poisson equation, $10 \rightarrow 256 \rightarrow 256 \rightarrow 128 \rightarrow 128 \rightarrow 1$ MLP with $D = 118\,145$
  - **SGD:** learning rate: $\mathcal{LU}([1 \cdot 10^{-6}; 1])$, momentum: $\mathcal{U}([0; 9.9 \cdot 10^{-1}])$, $N_\Omega$: $\mathcal{U}(\{100, 101, \ldots, 5\,000\})$, $N_{\partial\Omega}$: $\mathcal{U}(\{50, 51, \ldots, 2\,500\})$, batch sampling frequency: $\mathcal{U}(\{0, 1, \ldots, 5\,000\})$
  - **Adam:** learning rate: $\mathcal{LU}([1 \cdot 10^{-6}; 1])$, $N_\Omega$: $\mathcal{U}(\{100, 101, \ldots, 5\,000\})$, $N_{\partial\Omega}$: $\mathcal{U}(\{50, 51, \ldots, 2\,500\})$, batch sampling frequency: $\mathcal{U}(\{0, 1, \ldots, 5\,000\})$
  - **Hessian-free:** curvature matrix: $\mathcal{U}(\{\text{GGN}, \text{Hessian}\})$, initial damping: $\mathcal{LU}([1 \cdot 10^{-15}; 1])$, constant damping: $\mathcal{U}(\{\text{no}, \text{yes}\})$, maximum CG iterations: $\mathcal{U}(\{1, 2, \ldots, 500\})$, $N_\Omega$: $\mathcal{U}(\{100, 101, \ldots, 5\,000\})$, $N_{\partial\Omega}$: $\mathcal{U}(\{50, 51, \ldots, 2\,500\})$, batch sampling frequency: $\mathcal{U}(\{0, 1, \ldots, 5\,000\})$
  - **LBFGS:** learning rate: $\mathcal{LU}([1 \cdot 10^{-6}; 1])$, history size: $\mathcal{U}(\{5, 6, \ldots, 500\})$, $N_\Omega$: $\mathcal{U}(\{100, 101, \ldots, 5\,000\})$, $N_{\partial\Omega}$: $\mathcal{U}(\{50, 51, \ldots, 2\,500\})$, batch sampling frequency: $\mathcal{U}(\{0, 1, \ldots, 5\,000\})$
  - **KFAC:** damping: $\mathcal{LU}([1 \cdot 10^{-15}; 1 \cdot 10^{-2}])$, momentum: $\mathcal{U}([0; 9.9 \cdot 10^{-1}])$, exponential moving average: $\mathcal{U}([0; 9.9 \cdot 10^{-1}])$, initialize Kronecker factors to identity: $\mathcal{U}(\{\text{no}, \text{yes}\})$, $N_\Omega$: $\mathcal{U}(\{100, 101, \ldots, 5\,000\})$, $N_{\partial\Omega}$: $\mathcal{U}(\{50, 51, \ldots, 2\,500\})$, batch sampling frequency: $\mathcal{U}(\{0, 1, \ldots, 5\,000\})$
  - **KFAC*:** damping: $\mathcal{LU}([1 \cdot 10^{-15}; 1 \cdot 10^{-2}])$, exponential moving average: $\mathcal{U}([0; 9.9 \cdot 10^{-1}])$, initialize Kronecker factors to identity: $\mathcal{U}(\{\text{no}, \text{yes}\})$, $N_\Omega$: $\mathcal{U}(\{100, 101, \ldots, 5\,000\})$, $N_{\partial\Omega}$: $\mathcal{U}(\{50, 51, \ldots, 2\,500\})$, batch sampling frequency: $\mathcal{U}(\{0, 1, \ldots, 5\,000\})$

- 100d Poisson equation, $100 \rightarrow 768 \rightarrow 768 \rightarrow 512 \rightarrow 512 \rightarrow 1$ MLP with $D = 1\,325\,057$
  - **SGD:** learning rate: $\mathcal{LU}([1 \cdot 10^{-6}; 1])$, momentum: $\mathcal{U}([0; 9.9 \cdot 10^{-1}])$, $N_\Omega$: $\mathcal{U}(\{100, 101, \ldots, 5\,000\})$, $N_{\partial\Omega}$: $\mathcal{U}(\{50, 51, \ldots, 2\,500\})$, batch sampling frequency: $\mathcal{U}(\{0, 1, \ldots, 1\,000\})$
  - **Adam:** learning rate: $\mathcal{LU}([1 \cdot 10^{-6}; 1])$, $N_\Omega$: $\mathcal{U}(\{100, 101, \ldots, 5\,000\})$, $N_{\partial\Omega}$: $\mathcal{U}(\{50, 51, \ldots, 2\,500\})$, batch sampling frequency: $\mathcal{U}(\{0, 1, \ldots, 1\,000\})$
  - **Hessian-free:** curvature matrix: $\mathcal{U}(\{\text{GGN}, \text{Hessian}\})$, initial damping: $\mathcal{LU}([1 \cdot 10^{-15}; 1])$, constant damping: $\mathcal{U}(\{\text{no}, \text{yes}\})$, maximum CG iterations: $\mathcal{U}(\{1, 2, \ldots, 500\})$, $N_\Omega$: $\mathcal{U}(\{100, 101, \ldots, 5\,000\})$, $N_{\partial\Omega}$: $\mathcal{U}(\{50, 51, \ldots, 2\,500\})$, batch sampling frequency: $\mathcal{U}(\{0, 1, \ldots, 1\,000\})$

- **LBFGS:** learning rate: $\mathcal{LU}([1 \cdot 10^{-6}; 1])$, history size: $\mathcal{U}(\{5, 6, \ldots, 500\})$, $N_\Omega$: $\mathcal{U}(\{100, 101, \ldots, 5\,000\})$, $N_{\partial\Omega}$: $\mathcal{U}(\{50, 51, \ldots, 2\,500\})$, batch sampling frequency: $\mathcal{U}(\{0, 1, \ldots, 1\,000\})$
- **KFAC:** damping: $\mathcal{LU}([1 \cdot 10^{-15}; 1 \cdot 10^{-2}])$, momentum: $\mathcal{U}([0; 9.9 \cdot 10^{-1}])$, exponential moving average: $\mathcal{U}([0; 9.9 \cdot 10^{-1}])$, initialize Kronecker factors to identity: $\mathcal{U}(\{\text{no}, \text{yes}\})$, $N_\Omega$: $\mathcal{U}(\{100, 101, \ldots, 5\,000\})$, $N_{\partial\Omega}$: $\mathcal{U}(\{50, 51, \ldots, 2\,500\})$, batch sampling frequency: $\mathcal{U}(\{0, 1, \ldots, 1\,000\})$
- **KFAC\*:** damping: $\mathcal{LU}([1 \cdot 10^{-15}; 1 \cdot 10^{-2}])$, exponential moving average: $\mathcal{U}([0; 9.9 \cdot 10^{-1}])$, initialize Kronecker factors to identity: $\mathcal{U}(\{\text{no}, \text{yes}\})$, $N_\Omega$: $\mathcal{U}(\{100, 101, \ldots, 5\,000\})$, $N_{\partial\Omega}$: $\mathcal{U}(\{50, 51, \ldots, 2\,500\})$, batch sampling frequency: $\mathcal{U}(\{0, 1, \ldots, 1\,000\})$

### A.6 PINN Loss for the Heat Equation

Consider the $(\tilde{d} + 1)$-dimensional homogeneous heat equation

$$\partial_t u(t, \tilde{\boldsymbol{x}}) - \kappa \Delta_{\tilde{\boldsymbol{x}}} u(t, \tilde{\boldsymbol{x}}) = 0$$

with spatial coordinates $\tilde{\boldsymbol{x}} \in \Omega \subseteq \mathbb{R}^{\tilde{d}}$ and time coordinate $t \in \mathrm{T} \subseteq \mathbb{R}$ where $\mathrm{T}$ is a time interval and $\kappa > 0$ denotes the heat conductivity. In this case, our neural network processes a $(d = \tilde{d} + 1)$-dimensional vector $\boldsymbol{x} = (t, \tilde{\boldsymbol{x}}^\top)^\top \in \mathbb{R}^d$ and we can re-write the heat equation as

$$\partial_{x_1} u(\boldsymbol{x}) - \kappa \sum_{d=2}^{d} \Delta_{x_d} u(\boldsymbol{x}) = 0 \,.$$

In the following, we consider the unit time interval $\mathrm{T} = [0; 1]$, the unit square $\Omega = [0; 1]^{\tilde{d}}$ and set $\kappa = \nicefrac{1}{4}$. There are two types of constraints we need to enforce on the heat equation in order to obtain unique solutions: initial conditions and boundary conditions. As our framework for the KFAC approximation assumes only two terms in the loss function, we combine the contributions from the boundary and initial values into one term.

To make this more precise, consider the following example solution of the heat equation, which will be used later on as well. As initial conditions, we use $u_0(\tilde{\boldsymbol{x}}) = u(0, \tilde{\boldsymbol{x}}) = \prod_{i=1}^{\tilde{d}} \sin(\pi \tilde{x}_i)$ for $\tilde{\boldsymbol{x}} \in \Omega$. For boundary conditions, we use $g(t, \tilde{\boldsymbol{x}}) = 0$ for $(t, \tilde{\boldsymbol{x}}) \in \mathrm{T} \times \partial\Omega$. The manufactured solution is

$$u_\star(t, \tilde{\boldsymbol{x}}) = \exp\left(-\frac{\pi^2 \tilde{d} t}{4}\right) \prod_{i=1}^{\tilde{d}} \sin(\pi[\tilde{x}_i]) \,.$$

The PINN loss for this problem consists of three terms: a PDE term, an initial value condition term, and a spatial boundary condition term,

$$L(\boldsymbol{\theta}) = \frac{1}{N_\Omega} \sum_{n=1}^{N_\Omega} \left(\partial_t u_{\boldsymbol{\theta}}(\boldsymbol{x}_n^\Omega) - \frac{1}{4} \Delta_{\tilde{\boldsymbol{x}}_n} u_{\boldsymbol{\theta}}(\boldsymbol{x}_n^\Omega)\right)^2$$

$$+ \frac{1}{N_{\partial\Omega}} \sum_{n=1}^{N_{\partial\Omega}} \left(u_{\boldsymbol{\theta}}(\boldsymbol{x}_n^{\partial\Omega}) - g(\boldsymbol{x}_n^{\partial\Omega})\right)^2$$

$$+ \frac{1}{N_0} \sum_{n=1}^{N_0} \left(u_{\boldsymbol{\theta}}(0, \boldsymbol{x}_n^0) - u_0(\boldsymbol{x}_n^0)\right)^2$$

with $\boldsymbol{x}_n^\Omega \sim \mathrm{T} \times \Omega$, and $\boldsymbol{x}_n^{\partial\Omega} \sim \mathrm{T} \times \partial\Omega$, and $\boldsymbol{x}_n^0 \sim \{0\} \times \Omega$. To fit this loss into our framework which assumes two loss terms, each of whose curvature is approximated with a Kronecker factor, we combine the initial value and boundary value conditions into a single term. Assuming $N_{\partial\Omega} = N_0 = \nicefrac{N_{\text{cond}}}{2}$ without loss of generality, we write

$$L(\boldsymbol{\theta}) = \underbrace{\frac{1}{N_\Omega} \sum_{n=1}^{N_\Omega} \left\|\partial_t u_{\boldsymbol{\theta}}(\boldsymbol{x}_n^\Omega) - \frac{1}{4} \Delta_{\tilde{\boldsymbol{x}}_n} u_{\boldsymbol{\theta}}(\boldsymbol{x}_n^\Omega) - y_n^\Omega\right\|_2^2}_{L_\Omega(\boldsymbol{\theta})} + \underbrace{\frac{1}{N_{\text{cond}}} \sum_{n=1}^{N_{\text{cond}}} \left\|u_{\boldsymbol{\theta}}(\boldsymbol{x}_n^{\text{cond}}) - y_n^{\text{cond}}\right\|_2^2}_{L_{\text{cond}}(\boldsymbol{\theta})}$$

with domain inputs $\boldsymbol{x}_n^\Omega \sim \mathrm{T} \times \Omega$ and targets $y_n^\Omega = 0$, boundary and initial condition targets $y_n^{\mathrm{cond}} = u_\star(\boldsymbol{x}_n^{\mathrm{cond}})$ with initial inputs $\boldsymbol{x}_n^{\mathrm{cond}} \sim \{0\} \times \Omega$ for $n = 1, \ldots, {}^{N_{\mathrm{cond}}}/2$ and boundary inputs $\boldsymbol{x}_n^{\mathrm{cond}} \sim \mathrm{T} \times \partial\Omega$ for $n = {}^{N_{\mathrm{cond}}}/2 + 1, \ldots, N_{\mathrm{cond}}$. This loss has the same structure as the PINN loss in Equation (1).

## A.7 1+1d Heat Equation

**Setup** We consider a 1+1-dimensional heat equation $\partial_t u(t, x) - \kappa \Delta_x u(t, x) = 0$ with $\kappa = {}^1/4$ on the unit square and unit time interval, $x, t \in [0, 1] \times [0, 1]$. The equation has zero spatial boundary conditions and the initial values are given by $u(0, x) = \sin(\pi x)$ for $\boldsymbol{x} \in [0, 1]$. We sample a single training batch of size $N_\Omega = 900$, $N_{\partial\Omega} = 120$ (${}^{N_{\partial\Omega}}/2$ points for the initial value and spatial boundary conditions each) and evaluate the $L_2$ error on a separate set of $9\,000$ data points using the known solution $u_\star(t, x) = \exp(-\pi^2 t/4) \sin(\pi x)$. Each run is limited to $1\,000\,\mathrm{s}$. We compare three MLP architectures of increasing size, each of whose linear layers are Tanh-activated except for the final one: a shallow $2 \to 64 \to 1$ MLP with $D = 257$ trainable parameters, a five layer $2 \to 64 \to 64 \to 48 \to 48 \to 1$ MLP with $D = 9\,873$ trainable parameters, and a five layer $2 \to 256 \to 256 \to 128 \to 128 \to 1$ MLP with $D = 116\,097$ trainable parameters. For the biggest architecture, full and layer-wise ENGD lead to out-of-memory errors and are thus not part of the experiments. Figure Figure A10 summarizes the results, and Figure A11 illustrates the learned solutions over training for all optimizers on the shallow MLP

**Best run details** The runs shown in Figure A10 correspond to the following hyper-parameters:

- $2 \to 64 \to 1$ MLP with $D = 257$
    - **SGD:** learning rate: $1.752\,752 \cdot 10^{-2}$, momentum: $9.9 \cdot 10^{-1}$
    - **Adam:** learning rate: $8.629\,006 \cdot 10^{-4}$
    - **Hessian-free:** curvature matrix: GGN, initial damping: $1 \cdot 10^{-4}$, constant damping: no, maximum CG iterations: 350
    - **LBFGS:** learning rate: $1 \cdot 10^{-1}$, history size: 125
    - **ENGD (full):** damping: $1 \cdot 10^{-12}$, exponential moving average: $9 \cdot 10^{-1}$, initialize Gramian to identity: no
    - **ENGD (layer-wise):** damping: $1 \cdot 10^{-10}$, exponential moving average: $3 \cdot 10^{-1}$, initialize Gramian to identity: no
    - **KFAC:** damping: $1.273\,754 \cdot 10^{-8}$, momentum: $7.562\,617 \cdot 10^{-1}$, exponential moving average: $3.611\,724 \cdot 10^{-1}$, initialize Kronecker factors to identity: yes
    - **KFAC*:** damping: $1.968\,427 \cdot 10^{-9}$, exponential moving average: $9.703\,638 \cdot 10^{-1}$, initialize Kronecker factors to identity: yes
- $2 \to 64 \to 64 \to 48 \to 48 \to 1$ MLP with $D = 9\,873$
    - **SGD:** learning rate: $9.276\,977 \cdot 10^{-2}$, momentum: $9.9 \cdot 10^{-1}$
    - **Adam:** learning rate: $2.551\,515 \cdot 10^{-3}$
    - **Hessian-free:** curvature matrix: GGN, initial damping: $1 \cdot 10^{-3}$, constant damping: no, maximum CG iterations: 200
    - **LBFGS:** learning rate: $2 \cdot 10^{-1}$, history size: 125
    - **ENGD (full):** damping: $1 \cdot 10^{-6}$, exponential moving average: $9 \cdot 10^{-1}$, initialize Gramian to identity: no
    - **ENGD (layer-wise):** damping: $1 \cdot 10^{-8}$, exponential moving average: $6 \cdot 10^{-1}$, initialize Gramian to identity: no
    - **KFAC:** damping: $3.169\,186 \cdot 10^{-13}$, momentum: $7.075\,879 \cdot 10^{-1}$, exponential moving average: $8.860\,410 \cdot 10^{-1}$, initialize Kronecker factors to identity: yes
    - **KFAC*:** damping: $5.035\,695 \cdot 10^{-14}$, exponential moving average: $9.815\,164 \cdot 10^{-1}$, initialize Kronecker factors to identity: yes
- $2 \to 256 \to 256 \to 128 \to 128 \to 1$ MLP with $D = 116\,097$
    - **SGD:** learning rate: $5.709\,474 \cdot 10^{-2}$, momentum: $9.9 \cdot 10^{-1}$
    - **Adam:** learning rate: $6.716\,485 \cdot 10^{-4}$

(a)

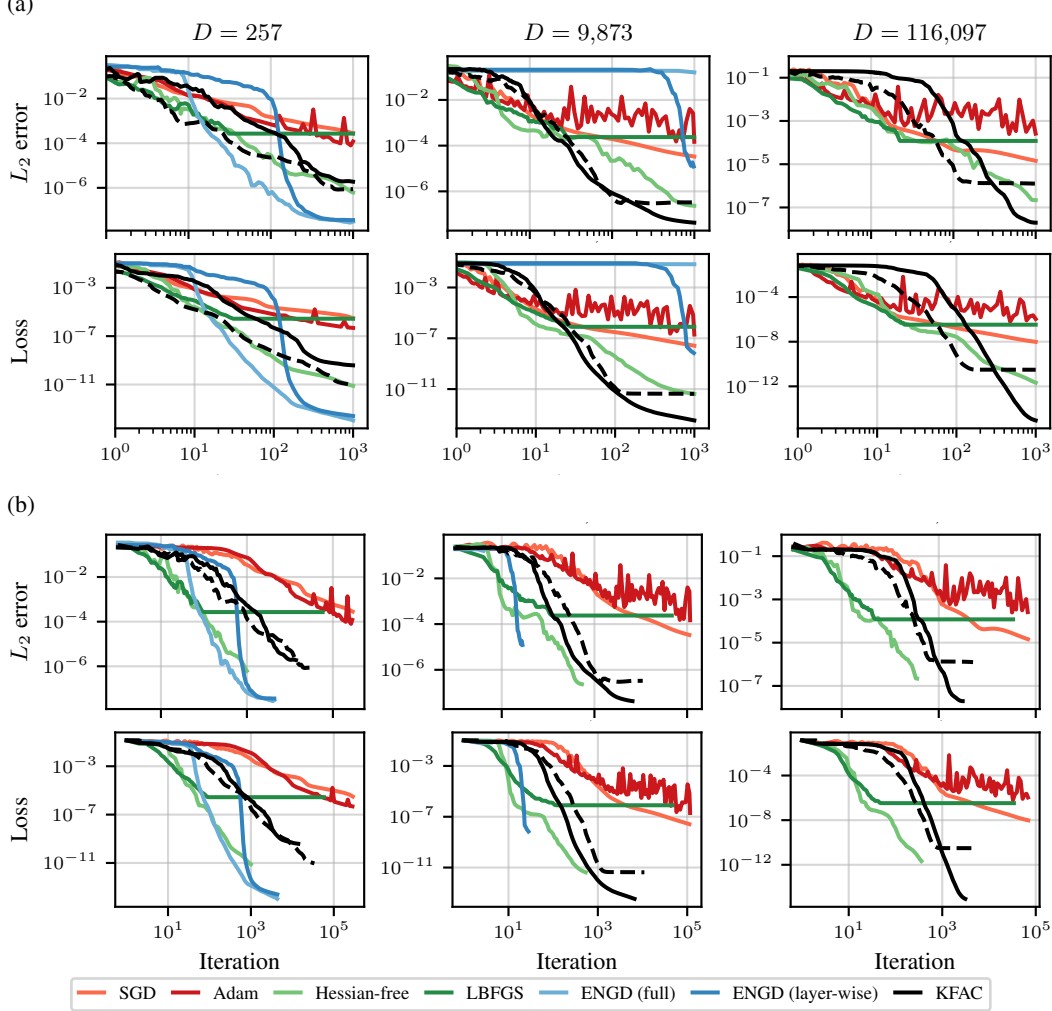

Figure A10: training loss and evaluation $L_2$ error for learning the solution to a 1+1-dimensional heat equation over (a) time and (b). each column corresponds to a different neural network.

- **Hessian-free:** curvature matrix: GGN, initial damping: $1 \cdot 10^{-2}$, constant damping: no, maximum CG iterations: 300
- **LBFGS:** learning rate: $2 \cdot 10^{-1}$, history size: 125
- **KFAC:** damping: $2.576\,488 \cdot 10^{-13}$, momentum: $2.043\,395 \cdot 10^{-2}$, exponential moving average: $9.727\,829 \cdot 10^{-1}$, initialize Kronecker factors to identity: yes
- **KFAC*:** damping: $7.343\,493 \cdot 10^{-11}$, exponential moving average: $9.765\,844 \cdot 10^{-1}$, initialize Kronecker factors to identity: yes

**Search space details**   The runs shown in Figure A10 were determined to be the best via a search with approximately 50 runs on the following search spaces which were obtained by refining an initially wider search ($\mathcal{U}$ denotes a uniform, and $\mathcal{LU}$ a log-uniform distribution):

- $2 \to 64 \to 1$ MLP with $D = 257$

  - **SGD:** learning rate: $\mathcal{LU}([1 \cdot 10^{-3}; 1 \cdot 10^{-1}])$, momentum: $\mathcal{U}(\{0, 3 \cdot 10^{-1}, 6 \cdot 10^{-1}, 9 \cdot 10^{-1}, 9.9 \cdot 10^{-1}\})$
  - **Adam:** learning rate: $\mathcal{LU}([5 \cdot 10^{-4}; 1 \cdot 10^{-1}])$

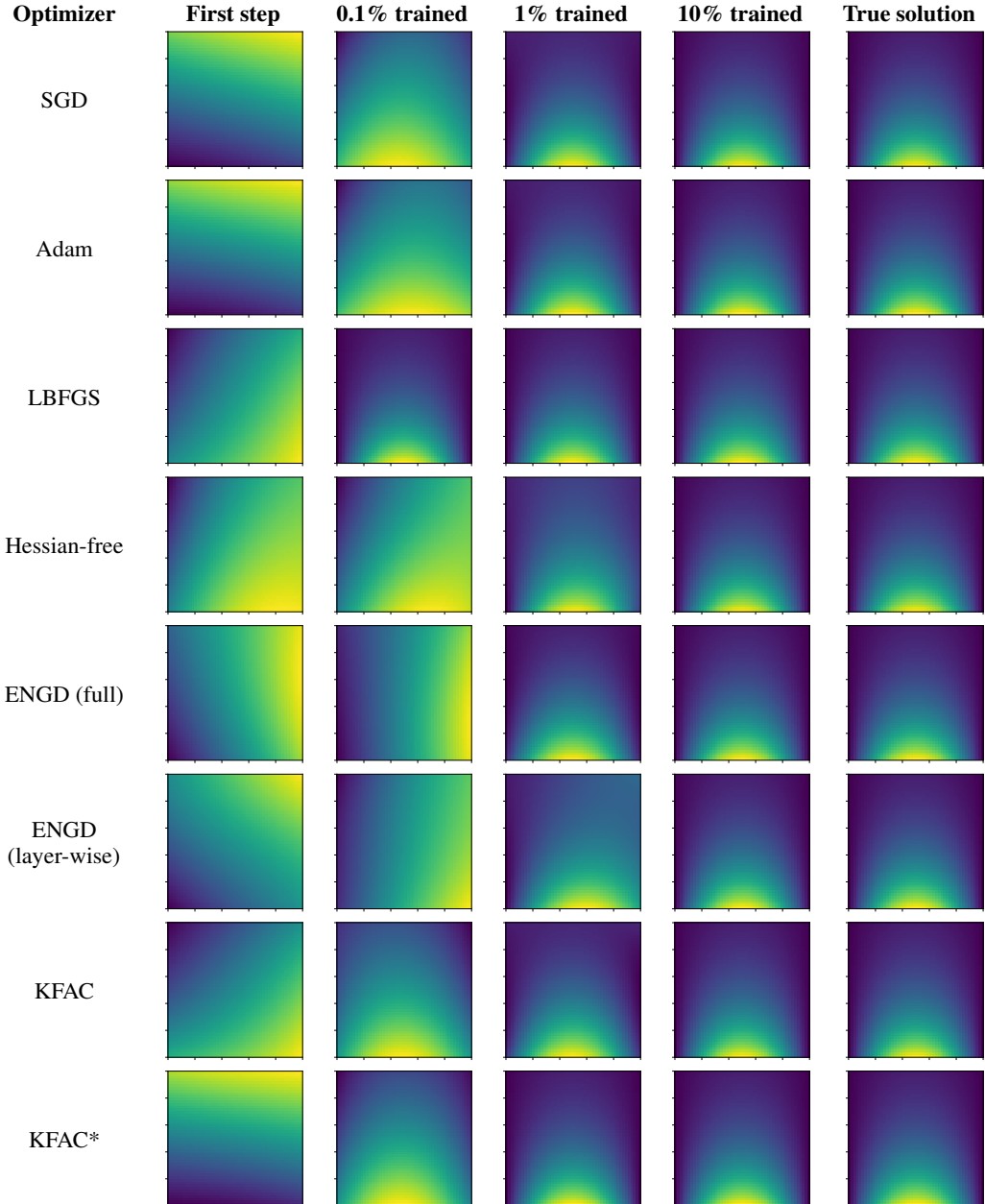

Figure A11: Visual comparison learned and true solutions while training with different optimizers for the 1+1d heat equation using a two-layer MLP (corresponding to the curves in Figure A10 left). All functions are shown on the unit square $(x, t) \in \Omega = [0; 1]^2$ and normalized to the unit interval.

- **Hessian-free:** curvature matrix: $\mathcal{U}(\{\text{GGN}, \text{Hessian}\})$, initial damping: $\mathcal{U}(\{100, 1, 1 \cdot 10^{-2}, 1 \cdot 10^{-4}, 1 \cdot 10^{-6}\})$, constant damping: $\mathcal{U}(\{\text{no}, \text{yes}\})$, maximum CG iterations: $\mathcal{U}(\{50, 250\})$

- **LBFGS:** learning rate: $\mathcal{U}(\{5 \cdot 10^{-1}, 2 \cdot 10^{-1}, 1 \cdot 10^{-1}, 5 \cdot 10^{-2}, 2 \cdot 10^{-2}, 1 \cdot 10^{-2}\})$, history size: $\mathcal{U}(\{75, 100, 125, 150, 175, 200, 225, 250\})$

- **ENGD (full):** damping: $\mathcal{U}(\{1 \cdot 10^{-6}, 1 \cdot 10^{-8}, 1 \cdot 10^{-10}, 1 \cdot 10^{-12}, 0\})$, exponential moving average: $\mathcal{U}(\{0, 3 \cdot 10^{-1}, 6 \cdot 10^{-1}, 9 \cdot 10^{-1}, 9.9 \cdot 10^{-1}\})$, initialize Gramian to identity: $\mathcal{U}(\{\text{no}, \text{yes}\})$

- **ENGD (layer-wise):** damping: $\mathcal{U}(\{1\cdot10^{-4}, 1\cdot10^{-6}, 1\cdot10^{-8}, 1\cdot10^{-10}, 0\})$, exponential moving average: $\mathcal{U}(\{0, 3\cdot10^{-1}, 6\cdot10^{-1}, 9\cdot10^{-1}, 9.9\cdot10^{-1}\})$, initialize Gramian to identity: $\mathcal{U}(\{\text{no, yes}\})$
- **KFAC:** damping: $\mathcal{LU}([1\cdot10^{-13}; 1\cdot10^{-7}])$, momentum: $\mathcal{U}([5\cdot10^{-1}; 9.9\cdot10^{-1}])$, exponential moving average: $\mathcal{U}([0; 9.9\cdot10^{-1}])$, initialize Kronecker factors to identity: yes
- **KFAC*:** damping: $\mathcal{LU}([1\cdot10^{-13}; 1\cdot10^{-7}])$, exponential moving average: $\mathcal{U}([5\cdot10^{-1}; 9.9\cdot10^{-1}])$, initialize Kronecker factors to identity: yes

- $2 \to 64 \to 64 \to 48 \to 48 \to 1$ MLP with $D = 9\,873$
  - **SGD:** learning rate: $\mathcal{LU}([1\cdot10^{-3}; 1\cdot10^{-1}])$, momentum: $\mathcal{U}(\{0, 3\cdot10^{-1}, 6\cdot10^{-1}, 9\cdot10^{-1}, 9.9\cdot10^{-1}\})$
  - **Adam:** learning rate: $\mathcal{LU}([5\cdot10^{-4}; 1\cdot10^{-1}])$
  - **Hessian-free:** curvature matrix: $\mathcal{U}(\{\text{GGN, Hessian}\})$, initial damping: $\mathcal{U}(\{100, 1, 1\cdot10^{-2}, 1\cdot10^{-4}, 1\cdot10^{-6}\})$, constant damping: $\mathcal{U}(\{\text{no, yes}\})$, maximum CG iterations: $\mathcal{U}(\{50, 250\})$
  - **LBFGS:** learning rate: $\mathcal{U}(\{5\cdot10^{-1}, 2\cdot10^{-1}, 1\cdot10^{-1}, 5\cdot10^{-2}, 2\cdot10^{-2}, 1\cdot10^{-2}\})$, history size: $\mathcal{U}(\{75, 100, 125, 150, 175, 200, 225, 250\})$
  - **ENGD (full):** damping: $\mathcal{U}(\{1\cdot10^{-6}, 1\cdot10^{-8}, 1\cdot10^{-10}, 1\cdot10^{-12}, 0\})$, exponential moving average: $\mathcal{U}(\{0, 3\cdot10^{-1}, 6\cdot10^{-1}, 9\cdot10^{-1}, 9.9\cdot10^{-1}\})$, initialize Gramian to identity: $\mathcal{U}(\{\text{no, yes}\})$
  - **ENGD (layer-wise):** damping: $\mathcal{U}(\{1\cdot10^{-4}, 1\cdot10^{-6}, 1\cdot10^{-8}, 1\cdot10^{-10}, 0\})$, exponential moving average: $\mathcal{U}(\{0, 3\cdot10^{-1}, 6\cdot10^{-1}, 9\cdot10^{-1}, 9.9\cdot10^{-1}\})$, initialize Gramian to identity: $\mathcal{U}(\{\text{no, yes}\})$
  - **KFAC:** damping: $\mathcal{LU}([1\cdot10^{-13}; 1\cdot10^{-7}])$, momentum: $\mathcal{U}([0; 9.9\cdot10^{-1}])$, exponential moving average: $\mathcal{U}([5\cdot10^{-1}; 9.9\cdot10^{-1}])$, initialize Kronecker factors to identity: yes
  - **KFAC*:** damping: $\mathcal{LU}([1\cdot10^{-15}; 1\cdot10^{-9}])$, exponential moving average: $\mathcal{U}([5\cdot10^{-1}; 9.9\cdot10^{-1}])$, initialize Kronecker factors to identity: yes

- $2 \to 256 \to 256 \to 128 \to 128 \to 1$ MLP with $D = 116\,097$
  - **SGD:** learning rate: $\mathcal{LU}([1\cdot10^{-3}; 1\cdot10^{-1}])$, momentum: $\mathcal{U}(\{0, 3\cdot10^{-1}, 6\cdot10^{-1}, 9\cdot10^{-1}, 9.9\cdot10^{-1}\})$
  - **Adam:** learning rate: $\mathcal{LU}([5\cdot10^{-4}; 1\cdot10^{-1}])$
  - **Hessian-free:** curvature matrix: $\mathcal{U}(\{\text{GGN, Hessian}\})$, initial damping: $\mathcal{U}(\{100, 1, 1\cdot10^{-2}, 1\cdot10^{-4}, 1\cdot10^{-6}\})$, constant damping: $\mathcal{U}(\{\text{no, yes}\})$, maximum CG iterations: $\mathcal{U}(\{50, 250\})$
  - **LBFGS:** learning rate: $\mathcal{U}(\{5\cdot10^{-1}, 2\cdot10^{-1}, 1\cdot10^{-1}, 5\cdot10^{-2}, 2\cdot10^{-2}, 1\cdot10^{-2}\})$, history size: $\mathcal{U}(\{75, 100, 125, 150, 175, 200, 225, 250\})$
  - **KFAC:** damping: $\mathcal{LU}([1\cdot10^{-14}; 1\cdot10^{-7}])$, momentum: $\mathcal{U}([0; 9.9\cdot10^{-1}])$, exponential moving average: $\mathcal{U}([5\cdot10^{-1}; 9.9\cdot10^{-1}])$, initialize Kronecker factors to identity: yes
  - **KFAC*:** damping: $\mathcal{LU}([1\cdot10^{-14}; 1\cdot10^{-6}])$, exponential moving average: $\mathcal{U}([5\cdot10^{-1}; 9.9\cdot10^{-1}])$, initialize Kronecker factors to identity: yes

## A.8  4+1d Heat Equation

**Setup**  We consider a 4+1-dimensional heat equation $\partial_t u(t, \boldsymbol{x}) - \kappa\Delta_{\boldsymbol{x}} u(t, \boldsymbol{x}) = 0$ with $\kappa = \frac{1}{4}$ on the four-dimensional unit square and unit time interval, $\boldsymbol{x}, t \in [0, 1]^4 \times [0, 1]$. The equation has spatial boundary conditions $u(t, x) = \exp(-t)\sum_{i=1}^4 \sin(2x_i)$ for $t, \boldsymbol{x} \in [0, 1] \times \partial[0, 1]^4$ throughout time, and initial value conditions $u(0, \boldsymbol{x}) = \sum_{i=1}^4 \sin(2x_i)$ for $\boldsymbol{x} \in [0, 1]^4$. We sample training batches of size $N_\Omega = 3\,000, N_{\partial\Omega} = 500$ ($\frac{N_{\partial\Omega}}{2}$ points for the initial value and spatial boundary conditions each) and evaluate the $L_2$ error on a separate set of $30\,000$ data points using the known solution $u_\star(t, \boldsymbol{x}) = \exp(-t)\sum_{i=1}^4 \sin(2x_i)$. All optimizers except for KFAC sample a new training batch each iteration. KFAC only re-samples every 100 iterations because we noticed significant improvement with multiple iterations on a fixed batch. To make sure that this does not lead to an unfair advantage of KFAC, we conduct an additional experiment where we also tune the batch sampling frequency, as well as other hyper-parameters; see §A.10. The results presented in this section are consistent with this additional experiment (compare the rightmost column of Figure A12

and Figure A14). Each run is limited to 3000 s. We compare three MLP architectures of increasing size, each of whose linear layers are Tanh-activated except for the final one: a shallow $5 \to 64 \to 1$ MLP with $D = 449$ trainable weights, a five layer $5 \to 64 \to 64 \to 48 \to 48 \to 1$ MLP with $D = 10\,065$ trainable weights, and a five layer $5 \to 256 \to 256 \to 128 \to 128 \to 1$ MLP with $D = 116\,864$ trainable weights. For the biggest architecture, full and layer-wise ENGD lead to out-of-memory errors and are thus not tested. Figure A12 visualizes the results.

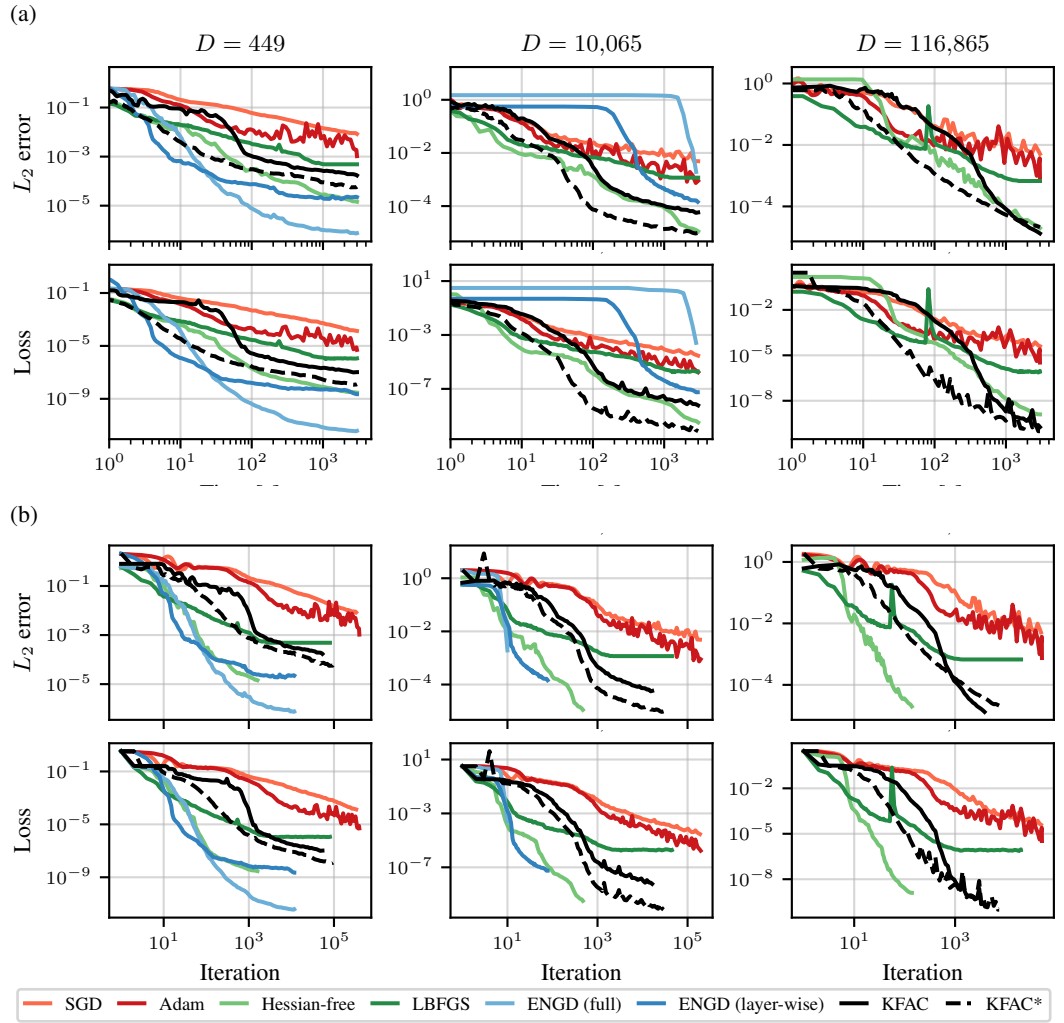

Figure A12: Training loss and evaluation $L_2$ error for learning the solution to a 4+1-d heat equation over (a) time and (b) steps. Columns are different neural networks.

**Search space details**   The runs shown in Figure A12 were determined to be the best via a search with approximately 50 runs on the following search spaces which were obtained by refining an initially wider search ($\mathcal{U}$ denotes a uniform, and $\mathcal{LU}$ a log-uniform distribution):

- $5 \to 64 \to 1$ MLP with $D = 449$
    - **SGD:** learning rate: $7.737\,742 \cdot 10^{-3}$, momentum: $9 \cdot 10^{-1}$
    - **Adam:** learning rate: $3.708\,460 \cdot 10^{-3}$
    - **Hessian-free:** curvature matrix: GGN, initial damping: $2 \cdot 10^{-1}$, constant damping: no, maximum CG iterations: $300$
    - **LBFGS:** learning rate: $2 \cdot 10^{-1}$, history size: $175$
    - **ENGD (full):** damping: $1 \cdot 10^{-10}$, exponential moving average: $6 \cdot 10^{-1}$, initialize Gramian to identity: yes

- **ENGD (layer-wise):** damping: $1 \cdot 10^{-6}$, exponential moving average: 0, initialize Gramian to identity: yes
- **KFAC:** damping: $1.000\,288 \cdot 10^{-9}$, momentum: $9.474\,108 \cdot 10^{-1}$, exponential moving average: $7.783\,519 \cdot 10^{-1}$, initialize Kronecker factors to identity: yes
- **KFAC*:** damping: $2.965\,060 \cdot 10^{-8}$, exponential moving average: $9.574\,717 \cdot 10^{-1}$, initialize Kronecker factors to identity: yes

- $5 \to 64 \to 64 \to 48 \to 48 \to 1$ MLP with $D = 10\,065$
  - **SGD:** learning rate: $9.357\,973 \cdot 10^{-3}$, momentum: $9 \cdot 10^{-1}$
  - **Adam:** learning rate: $7.801\,748 \cdot 10^{-4}$
  - **Hessian-free:** curvature matrix: GGN, initial damping: $5 \cdot 10^{-3}$, constant damping: no, maximum CG iterations: 400
  - **LBFGS:** learning rate: $1 \cdot 10^{-1}$, history size: 225
  - **ENGD (full):** damping: $1 \cdot 10^{-8}$, exponential moving average: 0, initialize Gramian to identity: yes
  - **ENGD (layer-wise):** damping: $1 \cdot 10^{-6}$, exponential moving average: $3 \cdot 10^{-1}$, initialize Gramian to identity: no
  - **KFAC:** damping: $4.143\,385 \cdot 10^{-14}$, momentum: $7.660\,303 \cdot 10^{-1}$, exponential moving average: $9.821\,414 \cdot 10^{-1}$, initialize Kronecker factors to identity: yes
  - **KFAC*:** damping: $1.955\,740 \cdot 10^{-10}$, exponential moving average: $9.821\,778 \cdot 10^{-1}$, initialize Kronecker factors to identity: yes

- $5 \to 256 \to 256 \to 128 \to 128 \to 1$ MLP with $D = 116\,865$
  - **SGD:** learning rate: $7.192\,473 \cdot 10^{-3}$, momentum: $9 \cdot 10^{-1}$
  - **Adam:** learning rate: $5.266\,284 \cdot 10^{-4}$
  - **Hessian-free:** curvature matrix: GGN, initial damping: $2 \cdot 10^{-3}$, constant damping: no, maximum CG iterations: 250
  - **LBFGS:** learning rate: $2 \cdot 10^{-1}$, history size: 200
  - **KFAC:** damping: $8.581\,322 \cdot 10^{-13}$, momentum: $8.501\,747 \cdot 10^{-1}$, exponential moving average: $9.803\,115 \cdot 10^{-1}$, initialize Kronecker factors to identity: yes
  - **KFAC*:** damping: $3.405\,440 \cdot 10^{-14}$, exponential moving average: $8.445\,471 \cdot 10^{-1}$, initialize Kronecker factors to identity: yes

**Search space details** The runs shown in Figure A12 were determined to be the best via a search with approximately 50 runs on the following search spaces which were obtained by refining an initially wider search ($\mathcal{U}$ denotes a uniform, and $\mathcal{LU}$ a log-uniform distribution):

- $5 \to 64 \to 1$ MLP with $D = 449$
  - **SGD:** learning rate: $\mathcal{LU}([1 \cdot 10^{-3}; 1 \cdot 10^{-2}])$, momentum: $\mathcal{U}(\{0, 3 \cdot 10^{-1}, 6 \cdot 10^{-1}, 9 \cdot 10^{-1}\})$
  - **Adam:** learning rate: $\mathcal{LU}([5 \cdot 10^{-4}; 1 \cdot 10^{-1}])$
  - **Hessian-free:** curvature matrix: $\mathcal{U}(\{\text{GGN}, \text{Hessian}\})$, initial damping: $\mathcal{U}(\{1, 1 \cdot 10^{-1}, 1 \cdot 10^{-2}, 1 \cdot 10^{-3}, 1 \cdot 10^{-4}\})$, constant damping: $\mathcal{U}(\{\text{no}, \text{yes}\})$, maximum CG iterations: $\mathcal{U}(\{50, 250\})$
  - **LBFGS:** learning rate: $\mathcal{U}(\{5 \cdot 10^{-1}, 2 \cdot 10^{-1}, 1 \cdot 10^{-1}, 5 \cdot 10^{-2}, 2 \cdot 10^{-2}, 1 \cdot 10^{-2}\})$, history size: $\mathcal{U}(\{50, 75, 100, 125, 150, 175, 200, 225\})$
  - **ENGD (full):** damping: $\mathcal{U}(\{1 \cdot 10^{-8}, 1 \cdot 10^{-9}, 1 \cdot 10^{-10}, 1 \cdot 10^{-11}, 1 \cdot 10^{-12}, 0\})$, exponential moving average: $\mathcal{U}(\{0, 3 \cdot 10^{-1}, 6 \cdot 10^{-1}, 9 \cdot 10^{-1}\})$, initialize Gramian to identity: $\mathcal{U}(\{\text{no}, \text{yes}\})$
  - **ENGD (layer-wise):** damping: $\mathcal{U}(\{1 \cdot 10^{-2}, 1 \cdot 10^{-3}, 1 \cdot 10^{-4}, 1 \cdot 10^{-5}, 1 \cdot 10^{-6}\})$, exponential moving average: $\mathcal{U}(\{0, 3 \cdot 10^{-1}, 6 \cdot 10^{-1}, 9 \cdot 10^{-1}, 9.9 \cdot 10^{-1}\})$, initialize Gramian to identity: $\mathcal{U}(\{\text{no}, \text{yes}\})$
  - **KFAC:** damping: $\mathcal{LU}([1 \cdot 10^{-12}; 1 \cdot 10^{-6}])$, momentum: $\mathcal{U}([0; 9.9 \cdot 10^{-1}])$, exponential moving average: $\mathcal{U}([5 \cdot 10^{-1}; 9.9 \cdot 10^{-1}])$, initialize Kronecker factors to identity: yes
  - **KFAC*:** damping: $\mathcal{LU}([1 \cdot 10^{-13}; 1 \cdot 10^{-7}])$, exponential moving average: $\mathcal{U}([5 \cdot 10^{-1}; 9.9 \cdot 10^{-1}])$, initialize Kronecker factors to identity: yes

- $5 \rightarrow 64 \rightarrow 64 \rightarrow 48 \rightarrow 48 \rightarrow 1$ MLP with $D = 10\,065$
  - **SGD:** learning rate: $\mathcal{LU}([1 \cdot 10^{-3}; 1 \cdot 10^{-2}])$, momentum: $\mathcal{U}(\{0, 3 \cdot 10^{-1}, 6 \cdot 10^{-1}, 9 \cdot 10^{-1}\})$
  - **Adam:** learning rate: $\mathcal{LU}([5 \cdot 10^{-4}; 1 \cdot 10^{-1}])$
  - **Hessian-free:** curvature matrix: $\mathcal{U}(\{\text{GGN}, \text{Hessian}\})$, initial damping: $\mathcal{U}(\{1, 1 \cdot 10^{-1}, 1 \cdot 10^{-2}, 1 \cdot 10^{-3}, 1 \cdot 10^{-4}\})$, constant damping: $\mathcal{U}(\{\text{no}, \text{yes}\})$, maximum CG iterations: $\mathcal{U}(\{50, 250\})$
  - **LBFGS:** learning rate: $\mathcal{U}(\{5 \cdot 10^{-1}, 2 \cdot 10^{-1}, 1 \cdot 10^{-1}, 5 \cdot 10^{-2}, 2 \cdot 10^{-2}, 1 \cdot 10^{-2}\})$, history size: $\mathcal{U}(\{50, 75, 100, 125, 150, 175, 200, 225\})$
  - **ENGD (full):** damping: $\mathcal{U}(\{1 \cdot 10^{-8}, 1 \cdot 10^{-9}, 1 \cdot 10^{-10}, 1 \cdot 10^{-11}, 1 \cdot 10^{-12}, 0\})$, exponential moving average: $\mathcal{U}(\{0, 3 \cdot 10^{-1}, 6 \cdot 10^{-1}, 9 \cdot 10^{-1}\})$, initialize Gramian to identity: $\mathcal{U}(\{\text{no}, \text{yes}\})$
  - **ENGD (layer-wise):** damping: $\mathcal{U}(\{1 \cdot 10^{-2}, 1 \cdot 10^{-3}, 1 \cdot 10^{-4}, 1 \cdot 10^{-5}, 1 \cdot 10^{-6}\})$, exponential moving average: $\mathcal{U}(\{0, 3 \cdot 10^{-1}, 6 \cdot 10^{-1}, 9 \cdot 10^{-1}, 9.9 \cdot 10^{-1}\})$, initialize Gramian to identity: $\mathcal{U}(\{\text{no}, \text{yes}\})$
  - **KFAC:** damping: $\mathcal{LU}([1 \cdot 10^{-14}; 1 \cdot 10^{-8}])$, momentum: $\mathcal{U}([0; 9.9 \cdot 10^{-1}])$, exponential moving average: $\mathcal{U}([5 \cdot 10^{-1}; 9.9 \cdot 10^{-1}])$, initialize Kronecker factors to identity: yes
  - **KFAC*:** damping: $\mathcal{LU}([1 \cdot 10^{-14}; 1 \cdot 10^{-8}])$, exponential moving average: $\mathcal{U}([5 \cdot 10^{-1}; 9.9 \cdot 10^{-1}])$, initialize Kronecker factors to identity: yes
- $5 \rightarrow 256 \rightarrow 256 \rightarrow 128 \rightarrow 128 \rightarrow 1$ MLP with $D = 116\,865$
  - **SGD:** learning rate: $\mathcal{LU}([1 \cdot 10^{-3}; 1 \cdot 10^{-2}])$, momentum: $\mathcal{U}(\{0, 3 \cdot 10^{-1}, 6 \cdot 10^{-1}, 9 \cdot 10^{-1}\})$
  - **Adam:** learning rate: $\mathcal{LU}([5 \cdot 10^{-4}; 1 \cdot 10^{-1}])$
  - **Hessian-free:** curvature matrix: $\mathcal{U}(\{\text{GGN}, \text{Hessian}\})$, initial damping: $\mathcal{U}(\{1, 1 \cdot 10^{-1}, 1 \cdot 10^{-2}, 1 \cdot 10^{-3}, 1 \cdot 10^{-4}\})$, constant damping: $\mathcal{U}(\{\text{no}, \text{yes}\})$, maximum CG iterations: $\mathcal{U}(\{50, 250\})$
  - **LBFGS:** learning rate: $\mathcal{U}(\{5 \cdot 10^{-1}, 2 \cdot 10^{-1}, 1 \cdot 10^{-1}, 5 \cdot 10^{-2}, 2 \cdot 10^{-2}, 1 \cdot 10^{-2}\})$, history size: $\mathcal{U}(\{50, 75, 100, 125, 150, 175, 200, 225\})$
  - **KFAC:** damping: $\mathcal{LU}([1 \cdot 10^{-14}; 1 \cdot 10^{-8}])$, momentum: $\mathcal{U}([0; 9.9 \cdot 10^{-1}])$, exponential moving average: $\mathcal{U}([5 \cdot 10^{-1}; 9.9 \cdot 10^{-1}])$, initialize Kronecker factors to identity: yes
  - **KFAC*:** damping: $\mathcal{LU}([1 \cdot 10^{-14}; 1 \cdot 10^{-8}])$, exponential moving average: $\mathcal{U}([5 \cdot 10^{-1}; 9.9 \cdot 10^{-1}])$, initialize Kronecker factors to identity: yes

### A.9 Robustness Under Model Initialization for 4+1d Heat Equation

Here we study the robustness of our results from §A.8 for the 4+1d heat equation when initializing the neural network differently. We choose the MLP with $D = 10\,065$ parameters from Figure 2's middle panel which is bigger than the two-layer toy model, while still allowing to run ENGD. Using the same hyper-parameters, we re-run all optimizers with 10 different model initializations. The results are shown in Figure A13. We observe that all optimizers perform similar to Figure 2, except for LBFGS which diverges for some runs.

### A.10 4+1d Heat Equation with Bayesian Search

**Setup** We consider the same heat equation as in §A.8 and use the $5 \rightarrow 256 \rightarrow 256 \rightarrow 128 \rightarrow 128 \rightarrow 1$ MLP with $D = 116\,865$. We tune all optimizer hyper-parameters as described in §A.1 and also tune the batch sizes $N_\Omega, N_{\partial\Omega}$, as well as their re-sampling frequencies. Figure A14 summarizes the results.

**Best run details** The runs shown in Figure A14 correspond to the following hyper-parameters:

- **SGD:** learning rate: $1.614\,965 \cdot 10^{-2}$, momentum: $9.899\,167 \cdot 10^{-1}$, $N_\Omega$: 527, $N_{\partial\Omega}$: 2\,157, batch sampling frequency: 543
- **Adam:** learning rate: $2.583\,569 \cdot 10^{-4}$, $N_\Omega$: 472, $N_{\partial\Omega}$: 3\,018, batch sampling frequency: 177

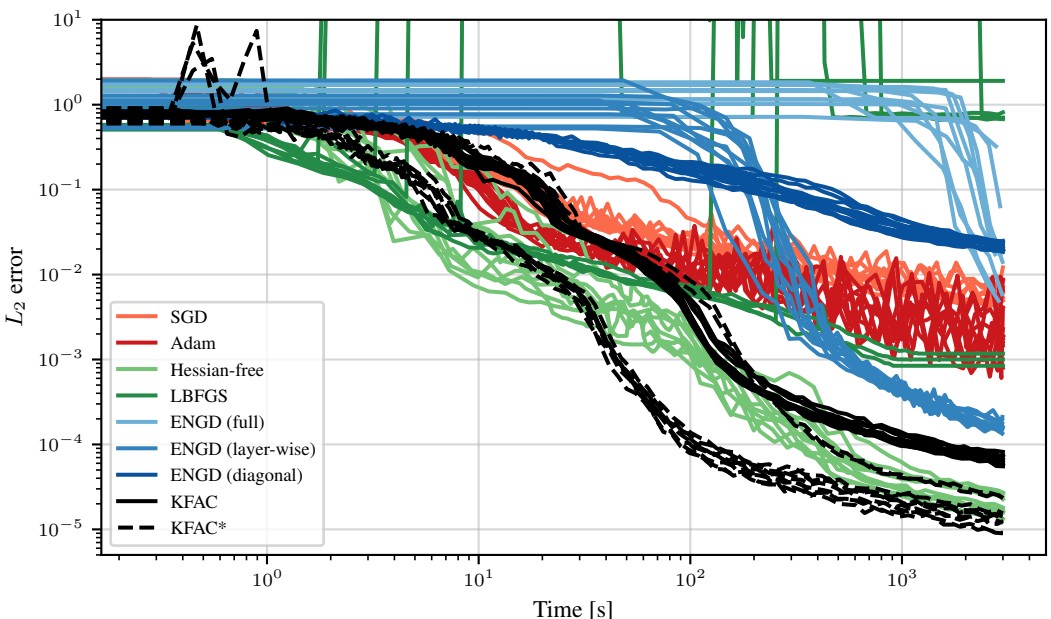

Figure A13: Best runs from the MLP with 10 065 parameters on the 4+1d heat equation from Figure 2 middle repeated over 10 different model initializations. All optimizers perform similarly, except for LBFGS which diverges for some runs.

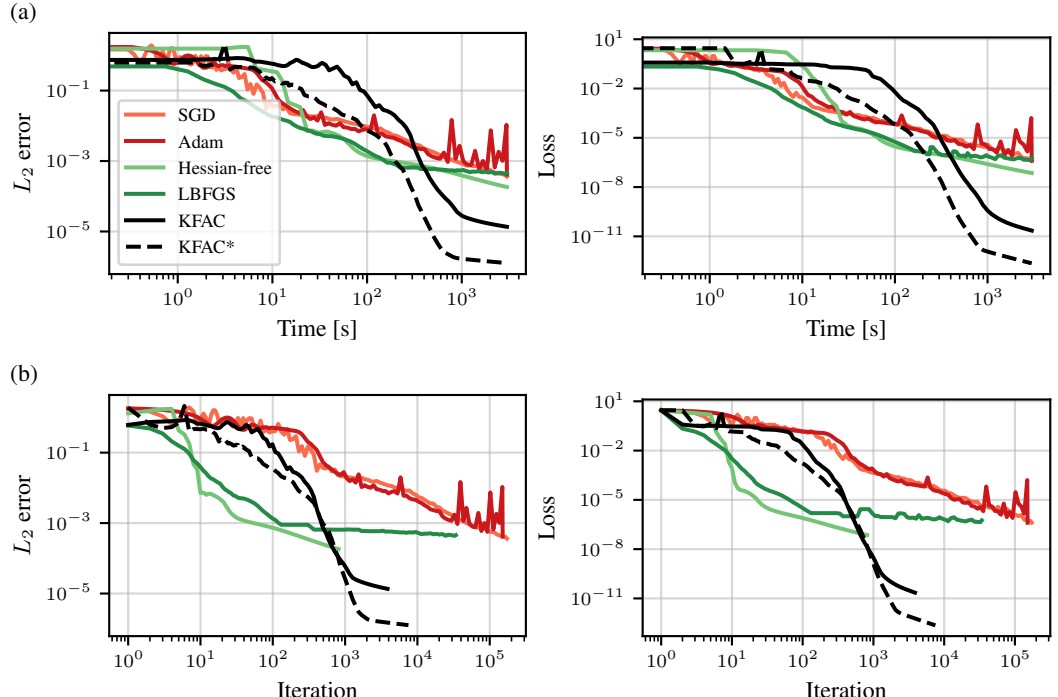

Figure A14: Training loss and evaluation $L_2$ error for learning the solution to a 4+1-dimensional heat equation over (a) time and (b) using Bayesian search.

- **Hessian-free:** curvature matrix: GGN, initial damping: $5.077\,634 \cdot 10^{-4}$, constant damping: yes, maximum CG iterations: 163, $N_\Omega$: 1 172, $N_{\partial\Omega}$: 1 637, batch sampling frequency: 5 440

- **LBFGS:** learning rate: $1.029\,194 \cdot 10^{-1}$, history size: $488$, $N_\Omega$: $582$, $N_{\partial\Omega}$: $2\,038$, batch sampling frequency: $315$
- **KFAC:** damping: $8.435\,180 \cdot 10^{-14}$, momentum: $9.718\,645 \cdot 10^{-1}$, exponential moving average: $9.800\,744 \cdot 10^{-1}$, initialize Kronecker factors to identity: yes, $N_\Omega$: $2\,525$, $N_{\partial\Omega}$: $2\,663$, batch sampling frequency: $7\,916$
- **KFAC\*:** damping: $8.837\,871 \cdot 10^{-15}$, exponential moving average: $9.887\,596 \cdot 10^{-1}$, initialize Kronecker factors to identity: yes, $N_\Omega$: $2\,563$, $N_{\partial\Omega}$: $2\,873$, batch sampling frequency: $9\,647$

**Search space details** The runs shown in Figure A14 were determined to be the best via a Bayesian search on the following search spaces which each optimizer given approximately the same total computational time ($\mathcal{U}$ denotes a uniform, and $\mathcal{LU}$ a log-uniform distribution):

- **SGD:** learning rate: $\mathcal{LU}([1 \cdot 10^{-6}; 1])$, momentum: $\mathcal{U}([0; 9.9 \cdot 10^{-1}])$, $N_\Omega$: $\mathcal{U}(\{100, 101, \ldots, 10\,000\})$, $N_{\partial\Omega}$: $\mathcal{U}(\{50, 51, \ldots, 5\,000\})$, batch sampling frequency: $\mathcal{U}(\{0, 1, \ldots, 10\,000\})$
- **Adam:** learning rate: $\mathcal{LU}([1 \cdot 10^{-6}; 1])$, $N_\Omega$: $\mathcal{U}(\{100, 101, \ldots, 10\,000\})$, $N_{\partial\Omega}$: $\mathcal{U}(\{50, 51, \ldots, 5\,000\})$, batch sampling frequency: $\mathcal{U}(\{0, 1, \ldots, 10\,000\})$
- **Hessian-free:** curvature matrix: $\mathcal{U}(\{\text{GGN}, \text{Hessian}\})$, initial damping: $\mathcal{LU}([1 \cdot 10^{-15}; 1])$, constant damping: $\mathcal{U}(\{\text{no}, \text{yes}\})$, maximum CG iterations: $\mathcal{U}(\{1, 2, \ldots, 500\})$, $N_\Omega$: $\mathcal{U}(\{100, 101, \ldots, 10\,000\})$, $N_{\partial\Omega}$: $\mathcal{U}(\{50, 51, \ldots, 5\,000\})$, batch sampling frequency: $\mathcal{U}(\{0, 1, \ldots, 10\,000\})$
- **LBFGS:** learning rate: $\mathcal{LU}([1 \cdot 10^{-6}; 1])$, history size: $\mathcal{U}(\{5, 6, \ldots, 500\})$, $N_\Omega$: $\mathcal{U}(\{100, 101, \ldots, 10\,000\})$, $N_{\partial\Omega}$: $\mathcal{U}(\{50, 51, \ldots, 5\,000\})$, batch sampling frequency: $\mathcal{U}(\{0, 1, \ldots, 10\,000\})$
- **KFAC:** damping: $\mathcal{LU}([1 \cdot 10^{-15}; 1 \cdot 10^{-2}])$, momentum: $\mathcal{U}([0; 9.9 \cdot 10^{-1}])$, exponential moving average: $\mathcal{U}([0; 9.9 \cdot 10^{-1}])$, initialize Kronecker factors to identity: $\mathcal{U}(\{\text{no}, \text{yes}\})$, $N_\Omega$: $\mathcal{U}(\{100, 101, \ldots, 10\,000\})$, $N_{\partial\Omega}$: $\mathcal{U}(\{50, 51, \ldots, 5\,000\})$, batch sampling frequency: $\mathcal{U}(\{0, 1, \ldots, 10\,000\})$
- **KFAC\*:** damping: $\mathcal{LU}([1 \cdot 10^{-15}; 1 \cdot 10^{-2}])$, exponential moving average: $\mathcal{U}([0; 9.9 \cdot 10^{-1}])$, initialize Kronecker factors to identity: $\mathcal{U}(\{\text{no}, \text{yes}\})$, $N_\Omega$: $\mathcal{U}(\{100, 101, \ldots, 10\,000\})$, $N_{\partial\Omega}$: $\mathcal{U}(\{50, 51, \ldots, 5\,000\})$, batch sampling frequency: $\mathcal{U}(\{0, 1, \ldots, 10\,000\})$

### A.11  9+1-d Logarithmic Fokker-Planck Equation with Random Search

For a given drift $\boldsymbol{\mu} : [0, 1] \times \mathbb{R}^d \to \mathbb{R}^d$ and diffusivity $\sigma : [0, 1] \to \mathbb{R}^{d \times d}$ the Fokker-Planck equation with initial probability density $p_0$ is given by

$$\partial_t p + \text{div}(\boldsymbol{\mu} p) - \frac{1}{2} \text{Tr}(\sigma\sigma^\top \nabla^2 p) = 0, \quad p(0) = p_0,$$

which is posed on $[0, 1] \times \mathbb{R}^d$. Note that $p(t, \cdot)$ is a probability density on $\mathbb{R}^d$ for all $t \in [0, 1]$. We transform the above equation into logarithmic space via $q = \log(p)$. Then $q$ solves

$$\partial_t q + \text{div}(\boldsymbol{\mu}) + \nabla q \cdot \boldsymbol{\mu} - \frac{1}{2}\|\sigma^\top \nabla q\|^2 - \frac{1}{2} \text{tr}(\sigma\sigma^\top \nabla^2 q) = 0, \quad q(0) = \log p_0.$$

For the concrete example of the main text, we set $\boldsymbol{\mu}(t, \boldsymbol{x}) = -\frac{1}{2}x$ and $\sigma = \sqrt{2}\boldsymbol{I} \in \mathbb{R}^{d \times d}$. We consider a 9+1 dimensional Fokker-Planck equation in logarithmic space and replace the unbounded domain by $[0, 1] \times [-5, 5]^d$. Precisely, we aim to solve the equation

$$\partial_t q(t, \boldsymbol{x}) - \frac{d}{2} - \frac{1}{2}\nabla q(t, \boldsymbol{x}) \cdot \boldsymbol{x} - \|\nabla q(t, \boldsymbol{x})\|^2 - \Delta q(t, \boldsymbol{x}) = 0, \quad q(0) = \log(p^*(0)),$$

where $d = 9$, $t \in [0, 1]$ and $\boldsymbol{x} \in [-5, 5]$. The solution $q^* = \log(p^*)$ is given as $p^*(t, \boldsymbol{x}) \sim \mathcal{N}(0, \exp(-t)\boldsymbol{I} + (1 - \exp(-t))2\boldsymbol{I})$. The PINN loss includes the PDE residual and the initial conditions. We model the solution with a medium sized tanh-activated MLP with $D = 118\,145$ and the layer structure $10 \to 256 \to 256 \to 128 \to 128 \to 1$ and use batch sizes of $N_\Omega = 3\,000$, $N_{\partial\Omega} = 1\,000$. Each run is assigned a budget of $6\,000$ s. Figure A15 visualizes the results.

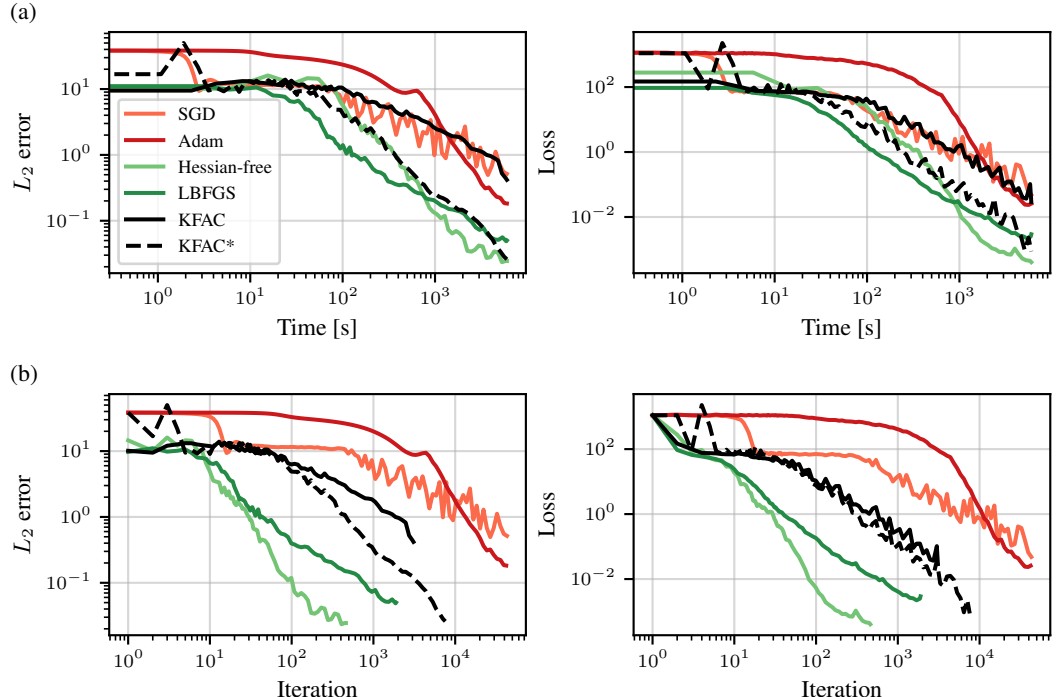

Figure A15: Training loss and evaluation $L_2$ error for learning the solution to a (9+1)d log Fokker-Planck equation over (a) time and (b) steps.

**Search space details** The runs shown in Figure A15 were determined to be the best via a random search on the following search spaces which each optimizer given approximately the same total computational time ($\mathcal{U}$ denotes a uniform, and $\mathcal{LU}$ a log-uniform distribution):

- **SGD:** learning rate: $\mathcal{LU}([1 \cdot 10^{-3}; 1 \cdot 10^{-2}])$, momentum: $\mathcal{U}(\{0, 3 \cdot 10^{-1}, 6 \cdot 10^{-1}, 9 \cdot 10^{-1}\})$
- **Adam:** learning rate: $\mathcal{LU}([5e\text{-}05; 5 \cdot 10^{-3}])$
- **Hessian-free:** curvature matrix: $\mathcal{U}(\{GGN, Hessian\})$, initial damping: $\mathcal{U}(\{1, 1 \cdot 10^{-1}, 1 \cdot 10^{-2}, 1 \cdot 10^{-3}, 1 \cdot 10^{-4}\})$, constant damping: $\mathcal{U}(\{no, yes\})$, maximum CG iterations: $\mathcal{U}(\{50, 250\})$
- **LBFGS:** learning rate: $\mathcal{U}(\{5 \cdot 10^{-1}, 2 \cdot 10^{-1}, 1 \cdot 10^{-1}, 5 \cdot 10^{-2}, 2 \cdot 10^{-2}, 1 \cdot 10^{-2}\})$, history size: $\mathcal{U}(\{50, 75, 100, 125, 150, 175, 200, 225\})$
- **KFAC:** damping: $\mathcal{LU}([1 \cdot 10^{-11}; 1 \cdot 10^{-5}])$, momentum: $\mathcal{U}([0; 9.9 \cdot 10^{-1}])$, exponential moving average: $\mathcal{U}([9.99 \cdot 10^{-1}; 9.999 \cdot 10^{-1}])$, initialize Kronecker factors to identity: yes
- **KFAC*:** damping: $\mathcal{LU}([1 \cdot 10^{-11}; 1 \cdot 10^{-5}])$, exponential moving average: $\mathcal{U}([9.99 \cdot 10^{-1}; 9.999 \cdot 10^{-1}])$, initialize Kronecker factors to identity: yes

We found that KFAC* requires very large exponential moving averages to work well.

# B Pseudo-Code: KFAC for the Poisson Equation

---

**Algorithm 1** KFAC for the Poisson equation.

---

**Require:**

MLP $u_{\boldsymbol{\theta}}$ with parameters $\boldsymbol{\theta}_0 = (\boldsymbol{\theta}_0^{(1)}, \ldots, \boldsymbol{\theta}_0^{(L)}) = (\text{vec}\, \boldsymbol{W}_0^{(1)}, \ldots, \text{vec}\, \boldsymbol{W}_0^{(L)})$,

interior data $\{(\boldsymbol{x}_n, y_n)\}_{n=1}^{N_\Omega}$,

boundary data $\{(\boldsymbol{x}_n^{\mathrm{b}}, y_n^{\mathrm{b}})\}_{n=1}^{N_{\partial\Omega}}$

exponential moving average $\beta$, momentum $\mu$, Damping $\lambda$, number of steps $T$

**0) Initialization**

**for** $l = 1, \ldots, L$ **do**

     $\boldsymbol{A}_\Omega^{(l)}, \boldsymbol{B}_\Omega^{(l)}, \boldsymbol{A}_{\partial\Omega}^{(l)}, \boldsymbol{B}_{\partial\Omega}^{(l)} \leftarrow \boldsymbol{0}$ or $\boldsymbol{I}$                $\triangleright$ Initialize Kronecker factors

**end for**

**for** $t = 0, \ldots, T - 1$ **do**

   **1) Compute the interior loss and update its approximate curvature**

   $(\boldsymbol{Z}_n^{(0)} \ldots, \boldsymbol{Z}_n^{(L)}, \Delta u_n) \leftarrow \Delta u_{\boldsymbol{\theta}_t}(\boldsymbol{x}_n) \quad n = 1, \ldots, N_\Omega$      $\triangleright$ Forward Laplacian wit intermediates

   Compute layer output gradients $\boldsymbol{g}_{n,s}^{(l)} := \partial\Delta u_n / \partial\boldsymbol{z}_{n,s}^{(l)}$ with autodiff in one backward pass

   $(\boldsymbol{g}_{n,s}^{(1)}, \ldots, \boldsymbol{g}_{n,s}^{(L)}) \leftarrow \texttt{grad}(\Delta u_n, (\boldsymbol{Z}_{n,s}^{(1)}, \ldots, \boldsymbol{Z}_{n,s}^{(L)})) \quad n = 1, \ldots, N_\Omega, \quad s = 1, \ldots, S := d + 2$

   **for all** $l = 1, \ldots, L$ **do**          $\triangleright$ Update Kronecker factors of the interior loss

     $\hat{\boldsymbol{A}}_\Omega^{(l)} \leftarrow \beta\hat{\boldsymbol{A}}_\Omega^{(l)} + (1 - \beta)\frac{1}{N_\Omega S}\sum_{n=1}^{N_\Omega} \boldsymbol{Z}_{n,s}^{(l-1)}\boldsymbol{Z}_{n,s}^{(l-1)\top}$

     $\hat{\boldsymbol{B}}_\Omega^{(l)} \leftarrow \beta\hat{\boldsymbol{B}}_\Omega^{(l)} + (1 - \beta)\frac{1}{N_\Omega}\sum_{n=1}^{N_\Omega} \boldsymbol{g}_{n,s}^{(l)}\boldsymbol{g}_{n,s}^{(l)\top}$

   **end for**

   $L_\Omega(\boldsymbol{\theta}_t) \leftarrow \frac{1}{2N_\Omega}\sum_{n=1}^{N_\Omega}(\Delta u_n - y_n)^2$             $\triangleright$ Compute interior loss

   **2) Compute the boundary loss and update its approximate curvature**

   $(\boldsymbol{z}_n^{(0)} \ldots, \boldsymbol{z}_n^{(L)}, u_n) \leftarrow u_{\boldsymbol{\theta}_t}(\boldsymbol{x}_n^{\mathrm{b}}) \quad n = 1, \ldots, N_{\partial\Omega}$      $\triangleright$ Forward pass with intermediates

   Compute layer output gradients $\boldsymbol{g}_n^{(l)} := \partial u_n / \boldsymbol{z}_n^{(l)}$ with autodiff in one backward pass

   $(\boldsymbol{g}_n^{(1)} \ldots, \partial\boldsymbol{g}_n^{(L)}) \leftarrow \texttt{grad}(u_n, (\boldsymbol{z}_n^{(0)} \ldots, \boldsymbol{z}_n^{(L)})) \quad n = 1, \ldots, N_{\partial\Omega}$

   **for all** $l = 1, \ldots, L$ **do**          $\triangleright$ Update Kronecker factors of the boundary loss

     $\hat{\boldsymbol{A}}_{\partial\Omega}^{(l)} \leftarrow \beta\hat{\boldsymbol{A}}_{\partial\Omega}^{(l)} + (1 - \beta)\frac{1}{N_{\partial\Omega}}\sum_{n=1}^{N_{\partial\Omega}} \boldsymbol{z}_n^{(l-1)}\boldsymbol{z}_n^{(l-1)\top}$

     $\hat{\boldsymbol{B}}_{\partial\Omega}^{(l)} \leftarrow \beta\hat{\boldsymbol{B}}_{\partial\Omega}^{(l)} + (1 - \beta)\frac{1}{N_{\partial\Omega}}\sum_{n=1}^{N_{\partial\Omega}} \boldsymbol{g}_n^{(l)}\boldsymbol{g}_n^{(l)\top}$

   **end for**

   $L_{\partial\Omega}(\boldsymbol{\theta}_t) \leftarrow \frac{1}{2N_{\partial\Omega}}\sum_{n=1}^{N_{\partial\Omega}}(u_n - y_n^{\mathrm{b}})^2$          $\triangleright$ Compute boundary loss

   **3) Update the preconditioner (use inverse of Kronecker sum trick)**

   **for all** $l = 1, \ldots, L$ **do**

     $\boldsymbol{C}^{(l)} \leftarrow \left[(\hat{\boldsymbol{A}}_\Omega^{(l)} + \lambda\boldsymbol{I}) \otimes (\hat{\boldsymbol{B}}_\Omega^{(l)} + \lambda\boldsymbol{I}) + (\hat{\boldsymbol{A}}_{\partial\Omega}^{(l)} + \lambda\boldsymbol{I}) \otimes (\hat{\boldsymbol{B}}_{\partial\Omega}^{(l)} + \lambda\boldsymbol{I})\right]^{-1}$

   **end for**

   **4) Compute the gradient using autodiff, precondition the gradient**

   $(\boldsymbol{g}^{(1)}, \ldots, \boldsymbol{g}^{(L)}) \leftarrow \texttt{grad}(L_\Omega(\boldsymbol{\theta}_t) + L_{\partial\Omega}(\boldsymbol{\theta}_t), (\boldsymbol{\theta}_t^{(1)}, \ldots, \boldsymbol{\theta}_t^{(L)}))$      $\triangleright$ Gradient with autodiff

   **for all** $l = 1, \ldots, L$ **do**              $\triangleright$ Precondition gradient

     $\boldsymbol{\Delta}_t \leftarrow -\boldsymbol{C}^{(l)}\boldsymbol{g}^{(l)}$           $\triangleright$ Proposed update direction

     $\hat{\boldsymbol{\delta}}_t^{(l)} \leftarrow \mu\boldsymbol{\delta}_{t-1}^{(l)} + \boldsymbol{\Delta}_t^{(l)}$ if $t > 0$ else $\boldsymbol{\Delta}_t^{(l)}$      $\triangleright$ Add momentum from previous update

   **end for**

   **5) Given the direction** $\hat{\boldsymbol{\delta}}_t^{(1)}, \ldots, \hat{\boldsymbol{\delta}}_t^{(L)}$**, choose learning rate** $\alpha$ **by line search & update**

   **for** $l = 1, \ldots, L$ **do**               $\triangleright$ Parameter update

     $\boldsymbol{\delta}_t^{(l)} \leftarrow \alpha\hat{\boldsymbol{\delta}}_t^{(l)}$

     $\boldsymbol{\theta}_{t+1}^{(l)} \leftarrow \boldsymbol{\theta}_t^{(l)} + \alpha\boldsymbol{\delta}_t^{(l)}$

   **end for**

**end for**

**return** Trained parameters $\boldsymbol{\theta}_T$

---

# C Taylor-Mode Automatic Differentiation & Forward Laplacian

PINN losses involve differential operators of the neural network, for instance the Laplacian. Recently, Li et al. [29] proposed a new computational framework called *forward Laplacian* to evaluate the Laplacian and the neural network's prediction in one forward traversal. To establish a Kronecker-factorized approximation of the Gramian, which consists of the Laplacian's gradient, we need to know how a weight matrix enters its computation. Here, we describe how the weight matrix of a linear layer inside a feed-forward net enters the Laplacian's computation when using the forward Laplacian framework. We start by connecting the forward Laplacian framework to Taylor-mode automatic differentiation [18, 3], both to make the presentation self-contained and to explicitly point out this connection which we believe has not been done previously.

## C.1 Taylor-Mode Automatic Differentiation

The idea of Taylor-mode is to forward-propagate Taylor coefficients, i.e. directional derivatives, through the computation graph. We provide a brief summary based on its description in [3].

**Taylor series and directional derivatives**   Consider a function $f : \mathbb{R}^d \to \mathbb{R}$ and its $K$-th order Taylor expansion at a point $\boldsymbol{x} \in \mathbb{R}^d$ along a direction $\alpha \boldsymbol{v} \in \mathbb{R}^d$ with $\alpha \in \mathbb{R}$,

$$\hat{f}(\alpha) = f(\boldsymbol{x} + \alpha \boldsymbol{v}) = f(\boldsymbol{x}) + \alpha \left( \frac{\partial f(\boldsymbol{x})}{\partial \boldsymbol{x}} \right)^\top \boldsymbol{v} + \frac{\alpha^2}{2!} \boldsymbol{v}^\top \left( \frac{\partial^2 f(\boldsymbol{x})}{\partial \boldsymbol{x}^2} \right) \boldsymbol{v}$$

$$+ \frac{\alpha^3}{3!} \sum_{i_1, i_2 i_3} \left( \frac{\partial^3 f(\boldsymbol{x})}{\partial \boldsymbol{x}^3} \right)_{i_1, i_2, i_3} v_{i_1} v_{i_2} v_{i_3}$$

$$+ \dots$$

$$+ \frac{\alpha^K}{K!} \sum_{i_1, \dots, i_K} \left( \frac{\partial^K f(\boldsymbol{x})}{\partial \boldsymbol{x}^K} \right)_{i_1, \dots, i_K} v_{i_1} \cdots v_{i_K} .$$

We can unify this expression by introducing the $K$-th order directional derivative of $f$ at $\boldsymbol{x}$ along $\boldsymbol{v}$,

$$\partial^K f(\boldsymbol{x}) \underbrace{[\boldsymbol{v}, \dots, \boldsymbol{v}]}_{K \text{ times}} := \sum_{i_1, \dots, i_K} \left( \frac{\partial^K f(\boldsymbol{x})}{\partial \boldsymbol{x}^K} \right)_{i_1, \dots, i_K} v_{i_1} \dots v_{i_K} .$$

This simplifies the uni-directional Taylor expansion to

$$\hat{f}(\alpha) = f(\boldsymbol{x} + \alpha \boldsymbol{v}) = f(\boldsymbol{x}) + \alpha \partial f(\boldsymbol{x})[\boldsymbol{v}] + \frac{\alpha^2}{2!} \partial^2 f(\boldsymbol{x})[\boldsymbol{v}, \boldsymbol{v}] + \frac{\alpha^3}{3!} \partial^3 f(\boldsymbol{x})[\boldsymbol{v}, \boldsymbol{v}, \boldsymbol{v}]$$

$$+ \dots + \frac{\alpha^K}{K!} \partial^K f(\boldsymbol{x})[\boldsymbol{v}, \dots, \boldsymbol{v}]$$

$$=: \sum_{k=1}^K \frac{\alpha^k}{k!} \partial^k f(\boldsymbol{x}) \left[ \otimes^k \boldsymbol{v} \right] =: \sum_{k=1}^K w_k^f \alpha^k$$

where we have used the notation $\otimes^k \boldsymbol{v}$ to indicate $k$ copies of $\boldsymbol{v}$, and introduced the $k$-th order Taylor coefficient $w_k^f \in \mathbb{R}$ of $f$. This generalizes to vector-valued functions: If $f$'s output was vector-valued, say $f(\boldsymbol{x}) \in \mathbb{R}^c$, we would have Taylor-expanded each component individually and grouped coefficients of same order into vectors $\boldsymbol{w}_k^f \in \mathbb{R}^c$ such that $[\boldsymbol{w}_k^f]_i$ is the $k$-th order Taylor coefficient of the $i$th component of $f$.

**A note on generality:**   In this introduction to Taylor-mode, we limit the discussion to the computation of higher-order derivatives along a single direction $\boldsymbol{v}$, i.e. $\partial^K f(\boldsymbol{x})[\boldsymbol{v}, \dots, \boldsymbol{v}]$. This is limited though, e.g. if we set $K = 2$ then we can compute $\partial^2 f(\boldsymbol{x})[\boldsymbol{v}, \boldsymbol{v}] = \boldsymbol{v}^\top (\partial^2 f(\boldsymbol{x})/\partial \boldsymbol{x}^2) \boldsymbol{v}$. We can set $\boldsymbol{v} = \boldsymbol{e}_i$ to the $i$-th standard basis vector to compute the $i$-th diagonal element of the Hessian. But we cannot evaluate off-diagonal elements, as this would require multi-directional derivatives, like $\partial^2 f(\boldsymbol{x})[\boldsymbol{e}_i, \boldsymbol{e}_{j \neq i}]$. A more general description of Taylor-mode for multi-directional Taylor series along $M$ directions, $\hat{f}(\alpha_1, \dots, \alpha_M) = f(\boldsymbol{x} + \alpha_1 \boldsymbol{v}_1 + \dots + \alpha_M \boldsymbol{v}_M)$, which require more general directional derivatives $\partial^K f(\boldsymbol{x})[\boldsymbol{v}_1, \dots, \boldsymbol{v}_K]$ (each vector can be different) are discussed in [25]. We will use this formulation later to generalize the forward Laplacian scheme to more general weighted sums of second-order derivatives in §C.3.

**Composition rule** Next, we consider the case where $f = g \circ h$ is a composition of two functions. Starting from the Taylor coefficients $\boldsymbol{w}_0^h, \ldots \boldsymbol{w}_K^h$ of $\hat{h}(\alpha) = h(\boldsymbol{x} + \alpha \boldsymbol{v})$, the Taylor coefficients $\boldsymbol{w}_0^f, \ldots, \boldsymbol{w}_K^f$ of $\hat{f}(\alpha) = f(\boldsymbol{x} + \alpha \boldsymbol{v})$ follow from Faà di Bruno's formula [18, 3]:

$$\boldsymbol{w}_k^f = \sum_{\sigma \in \mathrm{part}(k)} \frac{1}{n_1! \ldots n_K!} \partial^{|\sigma|} g(\boldsymbol{w}_0^h) \left[ \otimes_{s \in \sigma} \boldsymbol{w}_s^h \right] \tag{C20}$$

In the above, $\mathrm{part}(k)$ is the set of all integer partitionings of $k$; a set of sets. $|\sigma|$ denotes the length of a set $\sigma \in \mathrm{part}(k)$, $n_i$ is the count of integer $i$ in $\sigma$, and $\boldsymbol{w}_0^h = h(\boldsymbol{x})$.

**Second-order Taylor-mode** Our goal is the computation of second-order derivatives of $f$ w.r.t. $\boldsymbol{x}$. So let's work out Equation (C20) up to order $K = 2$. The zeroth and first order are simply the forward pass and the forward-mode gradient chain rule. For the second-order term, we need the integer partitioning of 2, given by $\mathrm{part}(2) = \{\{1,1\}, \{2\}\}$. This results in

$$\boldsymbol{w}_0^f = g(\boldsymbol{w}_0^h) \,, \tag{C21a}$$

$$\boldsymbol{w}_1^f = \partial g(\boldsymbol{w}_0^h)[\boldsymbol{w}_1^h] \,, \tag{C21b}$$

$$\boldsymbol{w}_2^f = \frac{1}{2} \partial^2 g(\boldsymbol{w}_0^h)[\boldsymbol{w}_1^h, \boldsymbol{w}_1^h] + \partial g(\boldsymbol{w}_0^h)[\boldsymbol{w}_2^h] \,. \tag{C21c}$$

We can also express $\boldsymbol{w}_1^f, \boldsymbol{w}_2^f$ in terms of Jacobian- and Hessian-vector products of $g$,

$$\boldsymbol{w}_1^f = \left( \mathrm{J}_{\boldsymbol{w}_0^h} g(\boldsymbol{w}_0^h) \right) \boldsymbol{w}_1^h \,, \tag{C22a}$$

$$\boldsymbol{w}_2^f = \frac{1}{2} \begin{pmatrix} \boldsymbol{w}_1^{h\top} \frac{\partial^2 [g(\boldsymbol{w}_0^h)]_1}{\partial \boldsymbol{w}_0^{h2}} \boldsymbol{w}_1^h \\ \vdots \\ \boldsymbol{w}_1^{h\top} \frac{\partial^2 [g(\boldsymbol{w}_0^h)]_D}{\partial \boldsymbol{w}_0^{h2}} \boldsymbol{w}_1^h \end{pmatrix} + \left( \mathrm{J}_{\boldsymbol{w}_0^h} g(\boldsymbol{w}_0^h) \right) \boldsymbol{w}_2^h \,. \tag{C22b}$$

Note that first-order Taylor-mode (Equation (C22a)) corresponds to the standard forward-mode autodiff which pushes forward error signals through Jacobian-vector products.

## C.2 Forward Laplacian

Our goal is to compute the Laplacian of $f : \mathbb{R}^d \to \mathbb{R}^c$ (in practise, $c = 1$),

$$\Delta_{\boldsymbol{x}} f(\boldsymbol{x}) = \sum_{i=1}^d \begin{pmatrix} \partial^2 [f(\boldsymbol{x})]_1 [\boldsymbol{e}_i, \boldsymbol{e}_i] \\ \vdots \\ \partial^2 [f(\boldsymbol{x})]_c [\boldsymbol{e}_i, \boldsymbol{e}_i] \end{pmatrix} := 2 \sum_{i=1}^d \boldsymbol{w}_{2,i}^f \in \mathbb{R}^c \,, \tag{C23}$$

where $\boldsymbol{e}_i$ is the $i$-th standard basis vector, $[f(\boldsymbol{x})]_j$ is the $j$-th component of $f(\boldsymbol{x})$, and we have introduced the second-order Taylor coefficients $\boldsymbol{w}_{2,i}^f$ of $f$ along $\boldsymbol{e}_i$. The Laplacian requires computing, then summing, the second-order Taylor coefficients of $d$ Taylor approximations $\{f(\boldsymbol{x} + \boldsymbol{e}_i)\}_{i=1,\ldots,d}$.

**Naive approach** We can use Taylor-mode differentiation to compute all these components in one forward traversal. Adding the extra loop over the Taylor expansions we want to compute in parallel, we obtain the following scheme from Equation (C21),

$$\boldsymbol{w}_0^f = g(\boldsymbol{w}_0^h) \,, \tag{C24a}$$

$$\left\{ \boldsymbol{w}_{1,i}^f \right\}_{i=1,\ldots,d} = \left\{ \partial g(\boldsymbol{w}_0^h)[\boldsymbol{w}_{1,i}^h] \right\}_{i=1,\ldots,d} \,, \tag{C24b}$$

$$\left\{ \boldsymbol{w}_{2,i}^f \right\}_{i=1,\ldots,d} = \left\{ \frac{1}{2} \partial^2 g(\boldsymbol{w}_0^h)[\boldsymbol{w}_{1,i}^h, \boldsymbol{w}_{1,i}^h] + \partial g(\boldsymbol{w}_0^h)[\boldsymbol{w}_{2,i}^h] \right\}_{i=1,\ldots,d} \,. \tag{C24c}$$

**Forward Laplacian framework** Computing the Laplacian via Equation (C24) first computes, then sums, the diagonal second-order derivatives $\{\boldsymbol{w}_{2,i}^f\}_{i=1,\ldots,d}$. Note that we can pull the sum

inside the forward propagation scheme, specifically Equation (C24c), and push-forward the summed second-order coefficients. This simplifies Equation (C24) to

$$\boldsymbol{w}_0^f = g(\boldsymbol{w}_0^h)\,, \tag{C25a}$$

$$\left\{\boldsymbol{w}_{1,i}^f\right\}_{i=1,\ldots,d} = \left\{\partial g(\boldsymbol{w}_0^h)[\boldsymbol{w}_{1,i}^h]\right\}_{i=1,\ldots,d}\,, \tag{C25b}$$

$$\underbrace{\sum_{i=1}^d \boldsymbol{w}_{2,i}^f}_{^{1\!/2}\Delta_{\boldsymbol{x}} f(\boldsymbol{x})} = \left(\frac{1}{2}\sum_{i=1}^d \partial^2 g(\boldsymbol{w}_0^h)[\boldsymbol{w}_{1,i}^h, \boldsymbol{w}_{1,i}^h]\right) + \partial g(\boldsymbol{w}_0^h)\underbrace{\left[\sum_{i=1}^d \boldsymbol{w}_{2,i}^h\right]}_{^{1\!/2}\Delta_{\boldsymbol{x}} g(\boldsymbol{x})}\,. \tag{C25c}$$

Equation (C25) is the forward Laplacian framework from Li et al. [29] for computing the Laplacian of a neural network. Here, we have derived it from Taylor-mode automatic differentiation. Note that Equation (C25) requires less computations and memory than Equation (C24) because we can pull the summation from the Laplacian into the forward propagation scheme.

### C.2.1 Forward Laplacian for Elementwise Activation Layers

We now describe Equation (C25) for the case where $g : \mathbb{R}^c \to \mathbb{R}^c$ acts element-wise via $\sigma : \mathbb{R} \to \mathbb{R}$. We will write $\sigma(\bullet), \sigma'(\bullet), \sigma''(\bullet)$ to indicate element-wise application of $\sigma$, its first derivative $\sigma'$, and second derivative $\sigma''$ to all elements of $\bullet$. Further, let $\odot$ denote element-wise multiplication, and $(\bullet)^{\odot 2}$ element-wise squaring. With that, we can write the Jacobian as $\mathrm{J}_{h(\boldsymbol{x})}g(\boldsymbol{x}) = \mathrm{diag}(\sigma(h(\boldsymbol{x})))$ where $\mathrm{diag}(\bullet)$ embeds a vector $\bullet$ into the diagonal of a matrix. The Hessian of component $i$ is $\partial^2 [g(h(\boldsymbol{x}))]_i / \partial h(\boldsymbol{x})^2 = [\sigma''(h(\boldsymbol{x}))]_i \boldsymbol{e}_i \boldsymbol{e}_i^\top$. Inserting Equation (C22) into Equation (C25) and using the Jacobian and Hessian expressions of the element-wise activation function yields the following forward Laplacian forward propagation:

$$\boldsymbol{w}_0^f = \sigma(\boldsymbol{w}_0^h)\,, \tag{C26a}$$

$$\left\{\boldsymbol{w}_{1,i}^f\right\} = \left\{\sigma'(\boldsymbol{w}_0^h) \odot \boldsymbol{w}_{1,i}^h\right\}_{i=1,\ldots,d}\,, \tag{C26b}$$

$$\sum_{i=1}^d \boldsymbol{w}_{2,i}^f = \frac{1}{2}\sigma''(\boldsymbol{w}_0^h) \odot \left(\sum_{i=1}^d \left(\boldsymbol{w}_{1,i}^h\right)^{\odot 2}\right) + \sigma'(\boldsymbol{w}_0^h) \odot \left(\sum_{i=1}^d \boldsymbol{w}_{2,i}^h\right)\,. \tag{C26c}$$

### C.2.2 Forward Laplacian for Linear Layers

Now, let $g : \mathbb{R}^{D_{\text{in}}} \to \mathbb{R}^{D_{\text{out}}}$ be a linear layer with weight matrix $\boldsymbol{W} \in \mathbb{R}^{D_{\text{out}} \times D_{\text{in}}}$ and bias vector $\boldsymbol{b} \in \mathbb{R}^{D_{\text{out}}}$. Its Jacobian is $\mathrm{J}_{h(\boldsymbol{x})}(\boldsymbol{W}h(\boldsymbol{x}) + \boldsymbol{b}) = \boldsymbol{W}$ and the second-order derivative is zero. Hence, Equation (C25) for linear layers becomes

$$\boldsymbol{w}_0^f = \boldsymbol{W}\boldsymbol{w}_0^h + \boldsymbol{b}\,, \tag{C27a}$$

$$\left\{\boldsymbol{w}_{1,i}^f\right\}_{i=1,\ldots,d} = \left\{\boldsymbol{W}\boldsymbol{w}_{1,i}^h\right\}_{i=1,\ldots,d}\,, \tag{C27b}$$

$$\sum_{i=1}^d \boldsymbol{w}_{2,i}^f = \boldsymbol{W}\left(\sum_{i=1}^d \boldsymbol{w}_{2,i}^h\right)\,. \tag{C27c}$$

We can summarize Equation (C27) in a single equation by grouping all quantities that are multiplied by $\boldsymbol{W}$ into a single matrix, and appending a single row of ones or zeros to account for the bias:

$$\underbrace{\begin{pmatrix}\boldsymbol{w}_0^f & \boldsymbol{w}_{1,1}^f & \cdots & \boldsymbol{w}_{1,d}^f & \sum_{i=1}^D \boldsymbol{w}_{2,i}^f\end{pmatrix}}_{:=\boldsymbol{T}^f \in \mathbb{R}^{D_{\text{out}} \times (d+2)}} = \begin{pmatrix}\boldsymbol{W} & \boldsymbol{b}\end{pmatrix} \underbrace{\begin{pmatrix}\boldsymbol{w}_0^h & \boldsymbol{w}_{1,1}^h & \cdots & \boldsymbol{w}_{1,d}^h & \sum_{i=1}^d \boldsymbol{w}_{2,i}^h \\ 1 & 0 & \cdots & 0 & 0\end{pmatrix}}_{:=\boldsymbol{T}^h \in \mathbb{R}^{(D_{\text{in}}+1) \times (d+2)}}\,,$$

or, in compact form,

$$\boldsymbol{T}^f = \tilde{\boldsymbol{W}}\boldsymbol{T}^h\,. \tag{C28}$$

Equation (C28) shows that the weight matrix $\tilde{\boldsymbol{W}}^{(l)} = (\boldsymbol{W}^{(l)}\ \boldsymbol{b}^{(l)})$ of a linear layer $f^{(l)}$ inside a neural network $f^{(L)} \circ \ldots \circ f^{(1)}$ is applied to a matrix $\boldsymbol{T}^{(l-1)} \in \mathbb{R}^{D_{\text{in}} \times (d+2)}$ during the computation of the net's prediction and Laplacian via the forward Laplacian framework and yields another matrix $\boldsymbol{T}^{(l)} \in \mathbb{R}^{D_{\text{out}} \times (d+2)}$.

## C.3 Generalization of the Forward Laplacian to Weighted Sums of Second Derivatives

The Laplacian is of the form $\Delta_{\boldsymbol{x}} f = \sum_i \partial^2 f(\boldsymbol{x})[\boldsymbol{e}_i, \boldsymbol{e}_i]$ and we previously described the forward Laplacian framework of Li et al. [29] as a consequence of pulling the summation into Taylor-mode's forward propagation. Here, we derive the forward propagation to more general operators of the form $\sum_{i,j} c_{i,j} \partial^2 f(\boldsymbol{x})[\boldsymbol{e}_i, \boldsymbol{e}_j]$, which contain the Laplacian for $c_{i,j} = \delta_{i,j}$.

As mentioned in §C.1, this requires a generalization of Taylor-mode which computes derivatives of the form $\partial^K f(\boldsymbol{x})[\boldsymbol{v}, \dots, \boldsymbol{v}]$, where the directions $\boldsymbol{v}$ must be identical. We start with the formulation in [25] which expresses the $K$-th multi-directional derivative of a function $f = g \circ h$ through the composites' derivatives (all functions can be vector-to-vector)

$$\partial^K f(\boldsymbol{x})[\boldsymbol{v}_1, \dots, \boldsymbol{v}_K] = \sum_{\sigma \in \text{part}(\{1,\dots,K\})} \partial^{|\sigma|} g(h(\boldsymbol{x})) \left[ \otimes_{\eta \in \sigma} \partial^{|\eta|} h(\boldsymbol{x}) \left[ \otimes_{l \in \eta} \boldsymbol{v}_l \right] \right] . \tag{C29}$$

Here, $\text{part}(\{1, \dots, K\})$ denotes the set of all set partitions of $\{1, \dots, K\}$ ($\sigma$ is a set of sets). E.g.,

$$\text{part}(\{1\}) = \{\{\{1\}\}\},$$
$$\text{part}(\{1, 2\}) = \{\{\{1, 2\}\}, \{\{1\}, \{2\}\}\},$$
$$\text{part}(\{1, 2, 3\}) = \{\{\{1, 2, 3\}\}, \{\{1\}, \{2, 3\}\}, \{\{1, 2\}, \{3\}\}, \{\{1, 3\}, \{2\}\}, \{\{1\}, \{2\}, \{3\}\}\}.$$

To make this more concrete, let's consider Equation (C29) for first- and second-order derivatives,

$$\partial f(\boldsymbol{x})[\boldsymbol{v}] = \partial g(h(\boldsymbol{x}))[\partial h(\boldsymbol{x})[\boldsymbol{v}]], \tag{C30a}$$

$$\partial^2 f(\boldsymbol{x})[\boldsymbol{v}_1, \boldsymbol{v}_2] = \partial g^2(h(\boldsymbol{x}))[\partial h(\boldsymbol{x})[\boldsymbol{v}_1], \partial h(\boldsymbol{x})[\boldsymbol{v}_2]] + \partial g(h(\boldsymbol{x}))[\partial h^2(\boldsymbol{x})[\boldsymbol{v}_1, \boldsymbol{v}_2]]. \tag{C30b}$$

From Equation (C30), we can see that if we want to compute a weighted sum of second-order derivatives $\sum_{i,j} c_{i,j} \partial^2 f(\boldsymbol{x})[\boldsymbol{v}_i, \boldsymbol{v}_j]$, we can pull the sum inside the second equation,

$$\sum_{i,j} c_{i,j} \partial^2 f(\boldsymbol{x})[\boldsymbol{v}_i, \boldsymbol{v}_j] = \sum_{i,j} c_{i,j} \partial^2 g(h(\boldsymbol{x}))[\partial h(\boldsymbol{x})[\boldsymbol{v}_i], \partial h(\boldsymbol{x})[\boldsymbol{v}_j]]$$
$$+ \partial g(h(\boldsymbol{x})) \left[ \sum_{i,j} c_{i,j} \partial^2 h(\boldsymbol{x})[\boldsymbol{v}_i, \boldsymbol{v}_j] \right] . \tag{C31}$$

Hence, we can propagate the collapsed second-order derivatives, together with all first-order derivatives along $\boldsymbol{v}_1, \boldsymbol{v}_2, \dots$. The only difference to the forward Laplacian is how second-order effects of an operation are incorporated (first term in Equation (C31)).

We now specify Equations (C29) and (C31) for linear layers and element-wise activation functions.

For a linear layer $g : h(\boldsymbol{x}) \mapsto \boldsymbol{W} h(\boldsymbol{x}) + \boldsymbol{b}$, we have $\partial^{m>1} g(h(x))[\boldsymbol{v}_1, \dots, \boldsymbol{v}_m] = \boldsymbol{0}$, and thus

$$\partial f(x)[\boldsymbol{v}] = \boldsymbol{W} \partial h(x)[\boldsymbol{v}], \tag{C32a}$$

$$\partial^2 f(x)[\boldsymbol{v}_1, \boldsymbol{v}_2] = \boldsymbol{W} \partial^2 h(x)[\boldsymbol{v}_1, \boldsymbol{v}_2], \tag{C32b}$$

$$\partial^K f(x)[\boldsymbol{v}_1, \dots, \boldsymbol{v}_K] = \boldsymbol{W} \partial^K h(x)[\boldsymbol{v}_1, \dots, \boldsymbol{v}_K]. \tag{C32c}$$

The last equation is because only the set partition $\{1, \dots, K\}$ contributes to Equation (C29).

For elementwise activations $g : h(x) \mapsto \sigma(h(x))$ with $\sigma : \mathbb{R} \to \mathbb{R}$ applied component-wise, we have the structured derivative tensor $[\partial^m g(h(x))]_{i_1,\dots,i_m} = \partial^m \sigma(h(x)_{i_1}) \delta_{i_1,\dots,i_m}$ and multi-directional derivative $\partial^K g(h(\boldsymbol{x}))[\boldsymbol{v}_1, \dots, \boldsymbol{v}_K] = \partial^K \sigma(\boldsymbol{x}) \odot \boldsymbol{v}_1 \odot \cdots \odot \boldsymbol{v}_K$. Equation (C30) becomes

$$\partial f(x)[\boldsymbol{v}] = \sigma'(h(x)) \odot \partial h(x)[\boldsymbol{v}], \tag{C33a}$$

$$\partial^2 f(x)[\boldsymbol{v}_1, \boldsymbol{v}_2] = \sigma''(h(x)) \odot \partial h(x)[\boldsymbol{v}_1] \odot \partial h(x)[\boldsymbol{v}_2] + \sigma'(h(x)) \odot \partial^2 h(x)[\boldsymbol{v}_1, \boldsymbol{v}_2]. \tag{C33b}$$

As shown in Equation (C30b), for both Equations (C32) and (C33), we can pull the summation inside the propagation scheme. Specifically, to compute $\sum_{i,j} c_{i,j} \partial^2 f(\boldsymbol{x})[\boldsymbol{e}_i, \boldsymbol{e}_j]$, we have for linear layers

$$f(\boldsymbol{x}) = g(h(\boldsymbol{x})), \tag{C34a}$$

$$\partial f(\boldsymbol{x})[\boldsymbol{e}_i] = \boldsymbol{W} \partial h(\boldsymbol{x})[\boldsymbol{e}_i], \qquad i = 1, \dots, d, \tag{C34b}$$

$$\sum_{i,j} c_{i,j} \partial^2 f(\boldsymbol{x})[\boldsymbol{e}_i, \boldsymbol{e}_j] = \boldsymbol{W} \left( \sum_{i,j} c_{i,j} \partial^2 h(\boldsymbol{x})[\boldsymbol{e}_i, \boldsymbol{e}_j] \right) . \tag{C34c}$$

and for activation layers

$$f(\boldsymbol{x}) = \sigma(h(\boldsymbol{x})) \,, \tag{C34d}$$

$$\partial f(\boldsymbol{x})[\boldsymbol{e}_i] = \sigma'(h(\boldsymbol{x})) \odot \partial h(\boldsymbol{x})[\boldsymbol{e}_i] \,, \qquad i = 1, \ldots, d \,, \tag{C34e}$$

$$\sum_{i,j} c_{i,j} \partial^2 f(\boldsymbol{x})[\boldsymbol{e}_i, \boldsymbol{e}_j] = \sum_{i,j} c_{i,j} \sigma''(h(\boldsymbol{x})) \odot \partial h(\boldsymbol{x})[\boldsymbol{e}_i] \odot \partial h(\boldsymbol{x})[\boldsymbol{e}_j]$$

$$+ \sigma'(h(\boldsymbol{x})) \odot \left( \sum_{i,j} c_{i,j} \partial^2 h(\boldsymbol{x})[\boldsymbol{e}_i, \boldsymbol{e}_j] \right) \,. \tag{C34f}$$

(the summed second-order derivatives that are forward-propagated are highlighted). This propagation reduces back to the forward Laplacian Equations (C26) and (C27) when we set $c_{i,j} = \delta_{i,j}$. In contrast to other attempts to compute such a weighted sum of second-order derivatives by reducing it to (multiple) partial standard forward Laplacians [30], we do not need to diagonalize the coefficient matrix and can compute the linear operator in one forward propagation.

### C.4 Comparison of Forward Laplacian and Autodiff Laplacian

**Setup** We compare the efficiency of the forward Laplacian, that we use in all our experiments, to an off-the shelve solution. We consider two Laplacian implementations:

1. *Autodiff Laplacian.* Computes the Laplacian with PyTorch's automatic differentiation (`functorch`) by computing the batched Hessian trace (via `torch.func.hessian` and `torch.func.vmap`). This is the standard approach in many PINN implementations.
2. *Forward Laplacian.* Computes the Laplacian via the forward Laplacian framework. We used this approach for all PDEs and optimizers, except ENGD, presented in the experiments.

We use the biggest network from our experiments (the $D_\Omega \to 768 \to 768 \to 512 \to 512 \to 1$ MLP with tanh-activations from Figure 3), then measure run time and peak memory of computing the net's Laplacian on a mini-batch of size $N = 1024$ with varying values of $D_\Omega$. To reduce measurement noise, we repeat each run over five independent Python sessions and report the smallest value (using the same GPU as in all other experiments, an NVIDIA RTX 6000 with 24 GiB memory).

**Results** The following tables compare run time and peak memory between the two approaches:

| $D_\Omega$ | Autodiff Laplacian [s] | Forward Laplacian [s] | $D_\Omega$ | Autodiff Laplacian [GiB] | Forward Laplacian [GiB] |
|---|---|---|---|---|---|
| 1 | 0.051 (1.6x) | 0.033 (1.0x) | 1 | 0.21 (0.96x) | 0.22 (1.0x) |
| 10 | 0.20 (2.0x) | 0.10 (1.0x) | 10 | 0.98 (1.6x) | 0.61 (1.0x) |
| 100 | 1.7 (2.0x) | 0.84 (1.0x) | 100 | 8.8 (1.9x) | 4.6 (1.0x) |

We observe that the forward Laplacian is roughly twice as fast as the `functorch` Laplacian, and that it uses significantly less memory for large input dimensions, up to only one half when $D_\Omega = 100$. We visualized both tables using more values for $D_\Omega$, see Figure C16. In the shown regime, we find that the MLP's increasing cost in $D_\Omega$ (due to the growing first layer) is negligible as we observe linear scaling in both memory and run time. For extremely large $D_\Omega$, it would eventually become quadratic.

## D  Backpropagation Perspective of the Laplacian

Here, we derive the computation graphs for the Laplacian and its associated Gramian when using reverse-mode AD, aka backpropagation. In contrast to the Taylor-mode perspective, the resulting expressions cannot be interpreted as simple weight-sharing. This complicates defining a Kronecker-factored approximation for the Gramian without introducing new approximations that are different from Eschenhagen et al. [17], rendering the Taylor-mode perspective advantageous.

We start by deriving the Laplacian $\Delta u := \mathrm{Tr}(\nabla_{\boldsymbol{x}}^2 u)$ of a feed-forward NN (see §2.1), assuming a single data point for simplicity (see §D.1) and abbreviating $u_{\boldsymbol{\theta}}$ as $u$. The goal is to make the Laplacian's dependence w.r.t. a weight $\boldsymbol{W}^{(i)}$ in one layer of the network explicit. Then, we can write

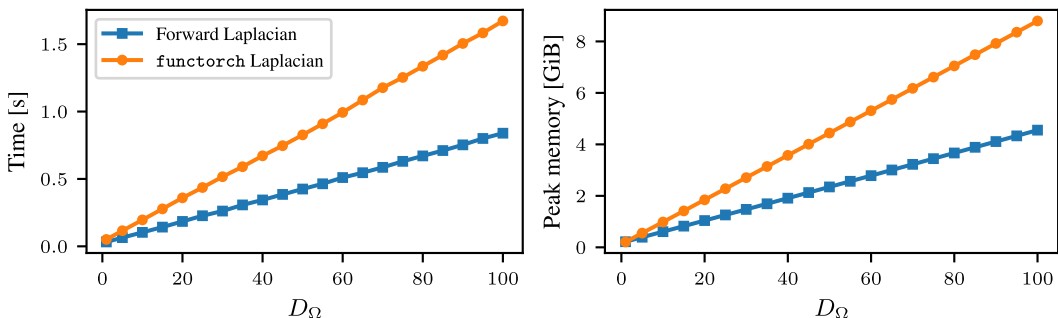

Figure C16: Time (left) and memory (right) required with the forward Laplacian used in our implementation and the `functorch` implementation.

down the Jacobian $\mathrm{J}_{\boldsymbol{W}^{(i)}} \Delta u$ (see §D.2) which is required for the Gramian in Equation (2) (see §D.3). We do this based on the concept of *Hessian backpropagation* [10, HBP,] which yields a recursion for the Hessian $\nabla_{\boldsymbol{x}}^2 u$. The Laplacian follows by taking the trace of the latter. Finally, we use the chain rule express the Laplacian's Jacobian $\mathrm{J}_{\boldsymbol{W}^{(i)}} \Delta u$ in terms of $\boldsymbol{W}^{(i)}$'s children in the compute graph.

### D.1  Hessian Backpropagation and Backward Laplacian

Gradient backpropagation describes a recursive procedure to compute gradients by backpropagating a signal via vector-Jacobian products (VJPs). A similar procedure can be derived to compute Hessians w.r.t. nodes in a graph ($\boldsymbol{z}^{(i)}$ or $\boldsymbol{\theta}^{(i)}$). We call this recursive procedure Hessian backpropagation [10].

**Gradient backpropagation**   As a warm-up, let's recall how to compute the gradient $\nabla_{\boldsymbol{\theta}} u = (\nabla_{\boldsymbol{\theta}^{(1)}} u, \ldots, \nabla_{\boldsymbol{\theta}^{(L)}} u)$. We start by setting $\nabla_{\boldsymbol{z}^{(L)}} u = \nabla_u u = 1$ (assuming $u$ is scalar for simplicity), then backpropagate the error via VJPs according to the recursion

$$
\begin{aligned}
\nabla_{\boldsymbol{z}^{(i-1)}} u &= \left( \mathrm{J}_{\boldsymbol{z}^{(i-1)}} \boldsymbol{z}^{(i)} \right)^{\top} \nabla_{\boldsymbol{z}^{(i)}} u \, , \\
\nabla_{\boldsymbol{\theta}^{(i)}} u &= \left( \mathrm{J}_{\boldsymbol{\theta}^{(i)}} \boldsymbol{z}^{(i)} \right)^{\top} \nabla_{\boldsymbol{z}^{(i)}} u
\end{aligned}
\tag{D35}
$$

for $i = L, \ldots, 1$. This yields the gradients of $u$ w.r.t. all intermediate representations and parameters.

**Hessian backpropagation**   Just like gradient backpropagation, we can derive a recursive scheme for the Hessian. Recall the Hessian chain rule

$$
\nabla^2 (f \circ g) = (\mathrm{J}g)^{\top} \nabla^2 f(g)(\mathrm{J}g) + \sum_k (\nabla_g f)_k \cdot \nabla^2 g_k,
\tag{D36}
$$

where $g_i$ denotes the individual components of $g$, see [53]. The recursion for computing Hessians of $u$ w.r.t. intermediate representations and parameters starts by initializing the recursion with $\nabla_{\boldsymbol{z}^{(L)}}^2 u = \nabla_u^2 u = 0$, and then backpropagating according to (see Dangel et al. [10] for details)

$$
\begin{aligned}
\nabla_{\boldsymbol{z}^{(i-1)}}^2 u &= \left( \mathrm{J}_{\boldsymbol{z}^{(i-1)}} \boldsymbol{z}^{(i)} \right)^{\top} \nabla_{\boldsymbol{z}^{(i)}}^2 u \left( \mathrm{J}_{\boldsymbol{z}^{(i-1)}} \boldsymbol{z}^{(i)} \right) + \sum_{k=1}^{h^{(i)}} \left( \nabla_{\boldsymbol{z}^{(i-1)}}^2 [\boldsymbol{z}^{(i)}]_k \right) [\nabla_{\boldsymbol{z}^{(i)}} u]_k \, , \\
\nabla_{\boldsymbol{\theta}^{(i)}}^2 u &= \left( \mathrm{J}_{\boldsymbol{\theta}^{(i)}} \boldsymbol{z}^{(i)} \right)^{\top} \nabla_{\boldsymbol{z}^{(i)}}^2 u \left( \mathrm{J}_{\boldsymbol{\theta}^{(i)}} \boldsymbol{z}^{(i)} \right) + \sum_{k=1}^{h^{(i)}} \left( \nabla_{\boldsymbol{\theta}^{(i)}}^2 [\boldsymbol{z}^{(i)}]_k \right) [\nabla_{\boldsymbol{z}^{(i)}} u]_k
\end{aligned}
\tag{D37}
$$

for $i = L, \ldots, 1$. The first term takes the incoming Hessian (w.r.t. a layer's output) and sandwiches it between the layer's Jacobian. It can be seen as backpropagating curvature from downstream layers. The second term adds in curvature introduced by the current layer. It is only non-zero if the layer is nonlinear. For linear layers, convolutional layers, and ReLU layers, it is zero.

Following the Hessian backpropagation procedure of Equation (D37) yields the per-layer parameter and feature Hessians $\nabla_{\boldsymbol{z}^{(i)}}^2 u, \nabla_{\boldsymbol{\theta}^{(i)}}^2 u$. In Figure D17 we depict the dependencies of intermediate gradients and Hessians for computing $\nabla_{\boldsymbol{x}}^2 u = \nabla_{\boldsymbol{z}^{(0)}}^2 u$:

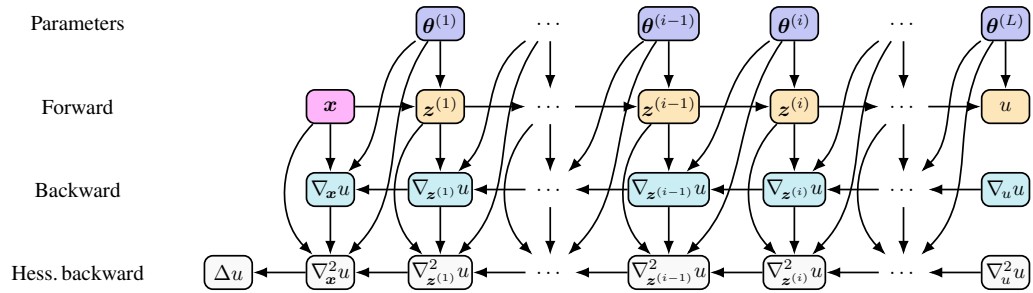

Figure D17: Computation graph of a sequential neural network's Laplacian $\Delta u$ when using (Hessian) backpropagation. Arrows indicate dependencies between intermediates. Note that $z^{(0)} := x$, $z^{(L)} := u$, $\nabla_u u = 1$, and $\nabla_u^2 u = 0$. For the Gramian, we are interested in how the neural network parameters enter the Laplacian's computation. Each parameter is used three times: during (i) the forward pass, (ii) the backward pass for the gradient, and (iii) the backward pass for the Hessian.

- $\nabla_{z^{(i-1)}} u$ depends on $\nabla_{z^{(i)}} u$ due to the recursion in Equation (D35), and on $z^{(i-1)}, \theta^{(i)}$ due to the Jacobian $J_{z^{(i-1)}} z^{(i)}$ in the gradient backpropagation Equation (D35).

- $\nabla_{z^{(i-1)}}^2 u$ depends on $\nabla_{z^{(i)}}^2 u$ and $\nabla_{z^{(i)}} u$ due to the recursion in Equation (D37), and on $z^{(i-1)}, \theta^{(i)}$ due to the Jacobian $J_{z^{(i-1)}} z^{(i)}$ and Hessian $\nabla_{z^{(i-1)}}^2 [z^{(i)}]_k$ in the Hessian backpropagation Equation (D35).

The Laplacian $\Delta u$ follows by taking the trace of $\nabla_x^2 u$ from above, and is hence recursively defined. To make these expressions more concrete, we now recap the HBP equations for fully-connected layers and element-wise nonlinear activations.

**Hessian backpropagation through nonlinear layers** We mostly consider nonlinear layers without trainable parameters and consist of a componentwise nonlinearity $z \mapsto \sigma(z)$ for some $\sigma \colon \mathbb{R} \to \mathbb{R}$. The Jacobian of such a nonlinear layer is given by $J_{z^{(i-1)}} z^{(i)} = \mathrm{diag}(\sigma'(z^{(i-1)}))$ and the Hessian terms are given by $\nabla_{z^{(i-1)}}^2 [z^{(i)}]_k = \sigma''(z_k^{(i-1)}) e_k e_k^\top$ where $e_k$ is the unit vector along coordinate $k$. With these two identities we can backpropagate the input Hessian through such layers via

$$
\begin{aligned}
\nabla_{z^{(i-1)}}^2 u = {} & \left( \mathrm{diag}(\sigma'(z^{(i-1)})) \right)^\top \nabla_{z^{(i)}}^2 u \left( \mathrm{diag}(\sigma'(z^{(i-1)})) \right) \\
& + \sum_{k=1}^{h^{(i)}} \sigma''(z_k^{(i-1)}) e_k e_k^\top [\nabla_{z^{(i)}} u]_k .
\end{aligned}
\tag{D38}
$$

**Hessian backpropagation through a linear layer** To de-clutter the dependency graph of Figure D17, we will now consider the dependency of $\Delta u$ w.r.t. the weight of a single layer. We assume this layer $i$ to be a linear layer with parameters $W^{(i)}$ such that $\theta^{(i)} = \mathrm{vec}(W^{(i)})$,

$$
z^{(i)} = W^{(i)} z^{(i-1)} .
\tag{D39}
$$

For this layer, the second terms in Equation (D37) disappears because the local Hessians are zero, that is $\nabla_{z^{(i-1)}}^2 [z^{(i)}]_k = 0$ and $\nabla_{W^{(i)}}^2 [z^{(i)}]_k = 0$. Also, the Jacobians are $J_{W^{(i)}} z^{(i)} = z^{(i-1)\top} \otimes I$ and $J_{z^{(i-1)}} z^{(i)} = W^{(i)}$ and hence only depend on one of the two layer inputs. This simplifies the computation graph. Figure D18 shows the dependencies of $W^{(i)}$ on the Laplacian, highlighting its three direct children,

$$
\begin{aligned}
z^{(i)} &= W^{(i)} z^{(i-1)} , \\
\nabla_{z^{(i-1)}} u &= W^{(i)\top} \left( \nabla_{z^{(i)}} u \right) , \\
\nabla_{z^{(i-1)}}^2 u &= W^{(i)\top} \left( \nabla_{z^{(i)}}^2 u \right) W^{(i)} .
\end{aligned}
\tag{D40}
$$

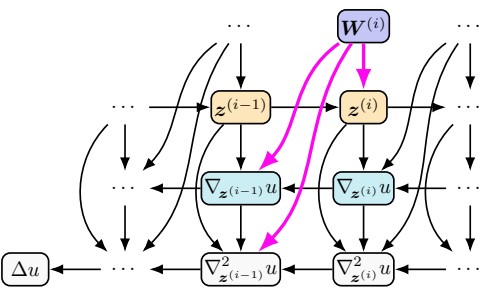

Figure D18: Direct dependencies of a linear layer's weight matrix $\boldsymbol{W}^{(i)}$ in the Laplacian's computation graph. There are three direct children: (i) the layer's output from the forward pass, (ii) the Laplacian's gradient w.r.t. the layer's input from the gradient backpropagation, and (iii) the Laplacian's Hessian w.r.t. the layer's input from the Hessian backpropagation. The Jacobians $\mathrm{J}_{\boldsymbol{W}^{(i)}}\Delta u$ required for the Gramian are the vector-Jacobian products accumulated over those children.

## D.2   Parameter Jacobian of the Backward Laplacian

Recall that the entries of the Gramian are composed from parameter derivatives of the input Laplacian, see Equation (2). We have identified the direct children of $\boldsymbol{W}^{(i)}$ in the Laplacian's compute graph, see Equation (D40). This allows us to compute the Jacobian $\mathrm{J}_{\boldsymbol{W}^{(i)}}\Delta u$ by the chain rule, i.e. by accumulating the Jacobians over all direct children,

$$
\begin{aligned}
\mathrm{J}_{\boldsymbol{W}^{(i)}}\Delta u &= \sum\nolimits_{\bullet \in \left\{\boldsymbol{z}^{(i)},\nabla_{\boldsymbol{z}^{(i-1)}}u,\nabla^2_{\boldsymbol{z}^{(i-1)}}u\right\}} \left(\mathrm{J}_{\boldsymbol{W}^{(i)}}\bullet\right)^\top \nabla_\bullet \Delta u \\
&= \left(\mathrm{J}_{\boldsymbol{W}^{(i)}}\boldsymbol{z}^{(i)}\right)^\top \nabla_{\boldsymbol{z}^{(i)}}\Delta u \\
&\quad + \left(\mathrm{J}_{\boldsymbol{W}^{(i)}}\nabla_{\boldsymbol{z}^{(i-1)}}u\right)^\top \nabla_{\nabla_{\boldsymbol{z}^{(i-1)}}u}\Delta u \\
&\quad + \left(\mathrm{J}_{\boldsymbol{W}^{(i)}}\nabla^2_{\boldsymbol{z}^{(i-1)}}u\right)^\top \nabla_{\nabla^2_{\boldsymbol{z}^{(i-1)}}u}\Delta u\,.
\end{aligned}
\tag{D41}
$$

The terms $\nabla_\bullet \Delta u$ can be computed with gradient backpropagation to the respective intermediates.

## D.3   Gramian of the Backward Laplacian

With the Laplacian's Jacobian from Equation (D41), we can now write down the Gramian block of the interior loss (up to summation over the data) for $\boldsymbol{W}^{(i)}$ as

$$
\begin{aligned}
\boldsymbol{G}_\Omega^{(i)} &= \left(\mathrm{J}_{\boldsymbol{W}^{(i)}}\Delta u\right)\left(\mathrm{J}_{\boldsymbol{W}^{(i)}}\Delta u\right)^\top \\
&= \sum\nolimits_{\bullet,\bullet \in \left\{\boldsymbol{z}^{(i)},\nabla_{\boldsymbol{z}^{(i-1)}}u,\nabla^2_{\boldsymbol{z}^{(i-1)}}u\right\}} \underbrace{\left(\mathrm{J}_{\boldsymbol{W}^{(i)}}\bullet\right)^\top \left[\left(\nabla_\bullet \Delta u\right)\left(\nabla_\bullet \Delta u\right)^\top\right]\left(\mathrm{J}_{\boldsymbol{W}^{(i)}}\bullet\right)}_{=:\boldsymbol{G}_{\Omega,\bullet,\bullet}^{(i)}}\,.
\end{aligned}
\tag{D42}
$$

The Gramian consists of nine different terms, see Figure D19 for a visualization which shows not only the diagonal blocks $\boldsymbol{G}_\Omega^{(i)}$, but also the full Gramian $\boldsymbol{G}_\Omega$ which decomposes in the same way. The terms $\nabla_\bullet \Delta u$ are automatically computed when computing the gradient of the loss via backpropagation. We will now proceed and simplify the terms by inserting the Jacobians into Equation (D41) and studying the Gramian's block diagonal, which is approximated by KFAC, in more detail.

**Computing** $\mathrm{J}_{\boldsymbol{W}^{(i)}}\bullet$   Let us first compute the Jacobians $\mathrm{J}_{\boldsymbol{W}^{(i)}}\bullet$ in Equation (D41). The Jacobian of the linear layer's forward pass is

$$
\mathrm{J}_{\boldsymbol{W}}\left(\boldsymbol{W}\boldsymbol{x}\right) = \boldsymbol{x}^\top \otimes \boldsymbol{I}\,.
\tag{D43a}
$$

The Jacobian from the gradient backpropagation is

$$
\mathrm{J}_{\boldsymbol{W}}\left(\boldsymbol{W}^\top \boldsymbol{x}\right) = \boldsymbol{I} \otimes \boldsymbol{x}^\top\,,
\tag{D43b}
$$

and the Jacobian from the Hessian backpropagation is

$$
\mathrm{J}_{\boldsymbol{W}}\left(\boldsymbol{W}^\top \boldsymbol{X}\boldsymbol{W}\right) = \boldsymbol{I} \otimes \boldsymbol{W}^\top \boldsymbol{X} + \boldsymbol{K}\left(\boldsymbol{I} \otimes \boldsymbol{W}^\top \boldsymbol{X}^\top\right)\,,
\tag{D43c}
$$

where $\boldsymbol{K} \in \mathbb{R}^{\dim(\boldsymbol{Z})\times\dim(\boldsymbol{Z})}$ (denoting $\boldsymbol{Z} := \boldsymbol{W}^\top \boldsymbol{X}\boldsymbol{W}$) is a permutation matrix that, when multiplied onto a vector whose basis corresponds to that of the flattened output $\boldsymbol{Z}$, modifies the order from first-varies-fastest to last-varies-fastest, i.e.

$$
\boldsymbol{K}\,\mathrm{vec}(\boldsymbol{Z}) = \mathrm{vec}(\boldsymbol{Z}^\top)\,.
$$

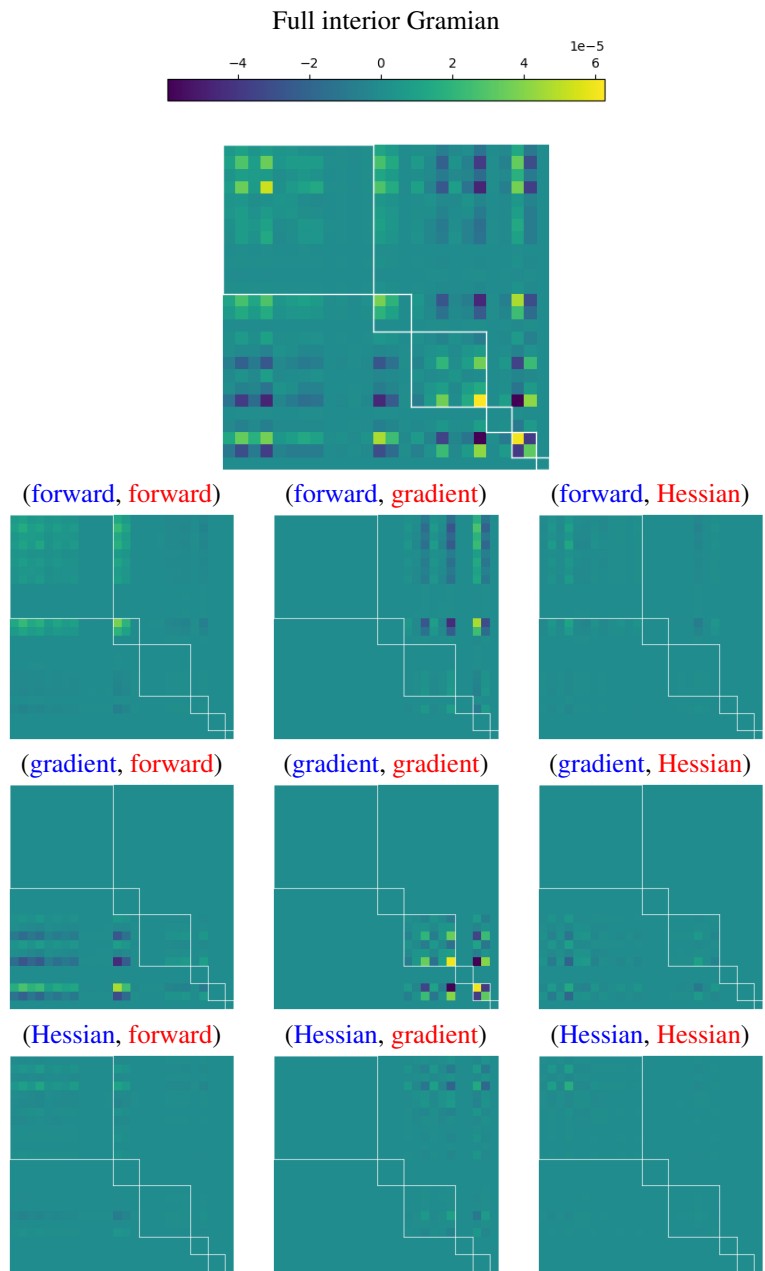

Figure D19: Contributions $G_{\Omega,\bullet,\bullet}$ to the Laplacian's Gramian $G_\Omega$ from different children in the computation graph on a synthetic toy problem. We use a $4 \to 3 \to 2 \to 1$ sigmoid-activated MLP and 10 randomly generated inputs. The contributions are highlighted as in Equation (D42).

Re-introducing the layer indices, the expressions in Equation (D40) become

$$
\begin{aligned}
\mathrm{J}_{\boldsymbol{W}^{(i)}} \boldsymbol{z}^{(i)} &= \boldsymbol{z}^{(i-1)^\top} \otimes \boldsymbol{I} \\
\mathrm{J}_{\boldsymbol{W}^{(i)}} \nabla_{\boldsymbol{z}^{(i-1)}} u\,, &= \boldsymbol{I} \otimes \nabla_{\boldsymbol{z}^{(i)}} u \\
\mathrm{J}_{\boldsymbol{W}^{(i)}} \nabla^2_{\boldsymbol{z}^{(i-1)}} u\,, &= \boldsymbol{I} \otimes \left[ \boldsymbol{W}^{(i)^\top} \left( \nabla^2_{\boldsymbol{z}^{(i)}} u \right) \right] + \boldsymbol{K} \left( \boldsymbol{I} \otimes \left[ \boldsymbol{W}^{(i)^\top} \left( \nabla^2_{\boldsymbol{z}^{(i)}} u \right)^\top \right] \right)\,.
\end{aligned}
\tag{D44}
$$

We will now use symmetries in the objects used during Hessian backpropagation to simplify this further. At a first glance, it looks like the Gramian consists of 16 terms, as there are 4 summands from the Jacobians in Equation (D43). However, we can simplify into 9 terms:

First, $\nabla^2_{\boldsymbol{z}^{(i)}} u$ is symmetric, that is

$$\mathrm{J}_{\boldsymbol{W}^{(i)}} \left( \boldsymbol{W}^{(i)^\top} \left( \nabla^2_{\boldsymbol{z}^{(i)}} u \right) \boldsymbol{W}^{(i)} \right) = \boldsymbol{I} \otimes \left[ \boldsymbol{W}^{(i)^\top} \left( \nabla^2_{\boldsymbol{z}^{(i)}} u \right) \right] + \boldsymbol{K} \left( \boldsymbol{I} \otimes \left[ \boldsymbol{W}^{(i)^\top} \left( \nabla^2_{\boldsymbol{z}^{(i)}} u \right) \right] \right) \,,$$

and the transposed Jacobian is

$$\boldsymbol{I} \otimes \left[ \left( \nabla^2_{\boldsymbol{z}^{(i)}} u \right) \boldsymbol{W}^{(i)} \right] + \left( \boldsymbol{I} \otimes \left[ \left( \nabla^2_{\boldsymbol{z}^{(i)}} u \right) \boldsymbol{W}^{(i)} \right] \right) \boldsymbol{K}^\top \,.$$

Second, we multiply the transpose Jacobian onto $\nabla_{\nabla^2_{\boldsymbol{z}^{(i-1)}} u} \Delta u$, which inherits symmetry from the Hessian, $[\nabla_{\nabla^2_{\boldsymbol{z}^{(i-1)}} u} \Delta u]_{j,k} = [\nabla_{\nabla^2_{\boldsymbol{z}^{(i-1)}} u} \Delta u]_{k,j}$. Due to this symmetry, the action of $\boldsymbol{K}$ (or $\boldsymbol{K}^\top$) does not alter it,

$$\boldsymbol{K}^\top \left( \nabla_{\nabla^2_{\boldsymbol{z}^{(i-1)}} u} \Delta u \right) = \nabla_{\nabla^2_{\boldsymbol{z}^{(i-1)}} u} \Delta u \,.$$

In other words, it does not matter how we flatten (first- or last-varies-fastest). This simplifies the VJP (last line in Equation (D42)) to

$$\left( \boldsymbol{I} \otimes \left[ \left( \nabla^2_{\boldsymbol{z}^{(i)}} u \right) \boldsymbol{W}^{(i)} \right] \right) \nabla_{\nabla^2_{\boldsymbol{z}^{(i-1)}} u} \Delta u + \left( \boldsymbol{I} \otimes \left[ \left( \nabla^2_{\boldsymbol{z}^{(i)}} u \right) \boldsymbol{W}^{(i)} \right] \right) \boldsymbol{K}^\top \nabla_{\nabla^2_{\boldsymbol{z}^{(i-1)}} u} \Delta u$$
$$= 2 \left( \boldsymbol{I} \otimes \left[ \left( \nabla^2_{\boldsymbol{z}^{(i)}} u \right) \boldsymbol{W}^{(i)} \right] \right) \nabla_{\nabla^2_{\boldsymbol{z}^{(i-1)}} u} \Delta u \,.$$

We can now write down the simplified Jacobian from Equation (D41), whose self-outer product forms the Gramian block for a linear layer's weight matrix,

$$\mathrm{J}_{\boldsymbol{W}^{(i)}} \Delta u = \underbrace{\left( \boldsymbol{z}^{(i-1)^\top} \otimes \boldsymbol{I} \right)^\top \nabla_{\boldsymbol{z}^{(i)}} \Delta u}_{(1)}$$
$$+ \underbrace{\left( \boldsymbol{I} \otimes \nabla_{\boldsymbol{z}^{(i)}} u \right)^\top \nabla_{\nabla_{\boldsymbol{z}^{(i-1)}} u} \Delta u}_{(2)} \tag{D45}$$
$$+ \underbrace{2 \left( \boldsymbol{I} \otimes \left[ \left( \nabla^2_{\boldsymbol{z}^{(i)}} u \right) \boldsymbol{W}^{(i)} \right] \right) \nabla_{\nabla^2_{\boldsymbol{z}^{(i-1)}} u} \Delta u}_{(3)} \,,$$

where (1) is the contribution from the forward pass, (2) is the contribution from the gradient back-propagation, and (3) is the contribution from the Hessian backpropagation. The Jacobians from Equation (D43) allow to express the Gramian in terms of Kronecker-structured expressions consisting of 9 terms in total. Figure D19 shows the 6 contributions from different children pairs.

**Conclusion** One problem of computing the Laplacian and its Jacobian with backpropagation according to Equation (D45) is that if we write out the Gramian's block $\boldsymbol{G}_\Omega^{(i)} = \mathrm{J}_{\boldsymbol{W}^{(i)}} \Delta u (\mathrm{J}_{\boldsymbol{W}^{(i)}} \Delta u)^\top$, we obtain 9 terms of different structure. Defining a single Kronecker product approximation would involve introducing new approximations on top of those employed by Eschenhagen et al. [17]. Therefore, the forward Laplacian, or Taylor-mode, perspective we choose in the main text is advantageous as it allows to define KFAC without introducing new approximations.

