# OpenReview forum: "Kronecker-Factored Approximate Curvature for Physics-Informed Neural Networks"
_NeurIPS.cc/2024/Conference — NeurIPS 2024 poster_

### Official Review · Reviewer_aiiA · 2024-06-27

**Soundness:** 3
**Presentation:** 3
**Contribution:** 3
**Rating:** 7
**Confidence:** 4

**Summary:**

This work develops a KFAC approximation for approximate second order optimization of physics-informed neural network (PINN) losses, in order to address optimization difficulties of PINNs.The first key idea is to use forward mode automatic differentiation for the different derivatives of the PDE and notice that it corresponds to a forward pass with additional inputs (corresponding to the different derivatives) and weight sharing. The second key idea is to then tie this to KFAC approximations for weight sharing architectures.
Both of these ideas are already known previous work. However, the combination of these two ideas in the context of PINN optimization in order to obtain a scalable approximate second order method is novel and promising. Experimental evaluation involves only training loss curves for 2 PDEs (heat equation and poisson equation) as a function of time or iterations. This has been done only for a single run (no error bars), hyper-parameters were tuned with different HPO methods.

**Strengths:**

- The paper is generally very well written: The introduction, contributions and related work section is kept short and on the point, providing a clear explanation of the problem setting and motivating the approach of this paper.
- The paper is technically solid. The authors are very precise in terms of the derivations and the notation. I could not spot any mistakes.
- Viewing the derivatives as forward propagation of additional inputs with weight sharing and tying this to recently developed KFAC approximations is a neat insight, even if both on their own are not novel.

**Weaknesses:**

- While the authors are very rigorous in their notation (which I listed as a strength), it often took me a very long time to parse everything due to the overwhelming information for simple expressions. For instance, even the simple expression of the chain rule that has just two terms can look quite complicated when writing the Jacobians using an operator applied to a vector and both having multiple indices that correspond to indices for layer, data point, derivative. Again, this makes the math precise and avoids confusion. But I think that some parts could be simplified to help the reader to focus their attention on what is actually new throughout all the respective parts of the paper, rather than having to slowly parse every detail and remembering what each index and variable corresponds to. I am wondering if some of this could be simplified without much loss of precision. Perhaps, one quick win would be to remove layer indices (superscript l): Every layer is approximated independently (factorizing). The indices would only be important if you had cross-layer interactions, but I do not see this. Secondly, I wish you could write the Jacobians directly with a single symbol rather than operator applied to a vector or matrix (but I guess this is more difficult, because you need to differentiate between weights and neurons, etc.).

- My main criticism is that the choice of experiments is unfortunately lacking similar rigor as the rest of the paper. The loss curves do show that the KFAC approximation is often favorable, but only for a single run. It would be important to do multiple runs to average out the noise from different initializations and mini-batching, and also include error bars. Furthermore, the loss curves only show the training loss and it is thus not clear if KFAC leads to more overfitting compared to other optimisers. I would have also liked to see some visualizations for some low-dimensional problems that show failure-cases of SGD and how KFAC fixes this. Or some other informative visualizations other than loss curves.

---------------------
- Minor: You noted that "Taylor-mode" AD is synonym to forward-mode AD and refer to related work in 3.1. However, "Taylor-mode" was mentioned a few times before, e.g. in the last sentence of related work, it is mentioned without a reference or explanation that it is also known as forward mode. Just put this information earlier.

**Questions:**

- I noticed that the EMA update rate hyper-parameter is often super large, even up to 0.988 (for heat equation, appendix A9), which effectively looks only at the most recent batch. Do you have an explanation for why this is? What happens if you use a much smaller mini-batch size?
- Is it correct that for linear PDEs, the approximation boils down to simply averaging over the additional forward passes corresponding to the additional derivatives on top of the average over the data points? So in comparison to the standard KFAC, you only have on additional average for these derivatives?
- For the non-linear PDEs, you had to make an additional approximation in C26 - C27 of the supplementary material. Does this correspond to linearizing the non-linear PDE? Is this analogous to a last-layer non-linearity, e.g. if we had some additional tanh activation before the regression outputs in order to bound the output values between -1 and 1? If I understood C24 correctly, the Psi term appears just due to chain rule, but I did not understand why it was or had to be grouped with the Hessian of the loss w.r.t. outputs. Could you expand on this?
- It seems very odd that KFAC performs better if the random mini-batch are *not* sampled (new) at every step. Have you noticed any such thing for KFAC on standard problems (not PINN losses)? Are the matrices in KFAC inverted every iteration or also just every T-th iteration?

**Limitations:**

The authors addressed the limitations, but I think the answer in (7) Experiment Statistical Significance should be No, as there are no error bars.

---

> ### Author Rebuttal · Authors · 2024-08-06
>
> Dear Reviewer aiiA,
>
> thanks a lot for the time and effort you put into reviewing our work.
>
> Regarding readability, we agree that layer indices can be omitted and will do so in the updated version. We will also think about other ways to make the notation lighter.
>
> **Weaknesses**
>
> > [...] the loss curves only show the training loss and it is thus not clear if KFAC leads to more overfitting compared to other optimisers.
>
> We respectfully disagree. *We report relative L2 error rather than training loss, which avoids overfitting*. The relative L2 error is defined as $\frac{\lVert u_\theta - u^\star\rVert_{L^2(\Omega)}}{\lVert u^\star\rVert_{L^2(\Omega)}}$, where $u^\star$ is the true solution of the PDE. In PINNs, the relative L2 error is usually considered as the relevant quantity and we estimate it on held-out points that are not used during training, similar to the test loss in supervised learning.
>
> We apologize for not explicitly introducing the relative L2 error and have adjusted this in our revised version.
>
> > The loss curves do show that the KFAC approximation is often favorable, but only for a single run. It would be important to do multiple runs to average out the noise from different initializations and mini-batching, and also include error bars.
>
> It is true that our plots show the performance of a single run. However, this run was obtained through extensive hyper-parameter tuning, and we found our results to be consistent when using a different search strategy (random or Bayesian). We think this already supports the robustness of our results, but understand your concern.
>
> Since our computational ressources are limited, would you be satisfied if we presented an additional experiment with error bars for one of the sub-experiments?
>
> > I would have also liked to see some visualizations for some low-dimensional problems that show failure-cases of SGD and how KFAC fixes this. Or some other informative visualizations other than loss curves.
>
> This is a great idea. We already have the code to do such visualizations because it was helpful to debug all optimizers. We will add such visualizations for the low-dimensional PDEs in the appendix.
>
> **Explicit questions**
>
> > I noticed that the EMA update rate hyper-parameter is often super large, even up to 0.988 (for heat equation, appendix A9), which effectively looks only at the most recent batch. Do you have an explanation for why this is? What happens if you use a much smaller mini-batch size?
>
> High EMA factors correspond to *slowly* forgetting pre-conditioners from previous iterations and are common in KFAC. For example, [Martens and Grosse](https://arxiv.org/pdf/1503.05671) uses an EMA factor increasing to 0.95 (page 19), which is also the [default value in KFAC's JAX implementation](https://github.com/google-deepmind/kfac-jax/blob/755f647d85252423f7dca6a14cf101735b5c46d8/kfac_jax/_src/optimizer.py#L298-L299).
>
> > Is it correct that for linear PDEs, the approximation boils down to simply averaging over the additional forward passes corresponding to the additional derivatives on top of the average over the data points? So in comparison to the standard KFAC, you only have on additional average for these derivatives?
>
> Yes, this is correct and very concise. We have added a clarifying sentence using your suggested phrasing.
>
> > For the non-linear PDEs, you had to make an additional approximation in C26 - C27 of the supplementary material. [...] Could you expand on this?
>
> Yes, the GN-preconditioner corresponds to linearizing the PDE. The approximations (C26)-(C27) are analoguous to the linear case and do not require further approximations compared to the linear case.
>
> The interpretation of $\Psi$ as a nonlinear output layer is correct. As $\Psi$ is not part of the convex loss, we have to linearize it for the GN-preconditioner. We group the Jacobian of $\Psi$ with the Hessian of the sample loss as it does not depend on the trainable parameters. It can also be grouped with the Jacobians $J_{n, \alpha}^{(l)}$, however, this results in the same approximation. We hope that this clarifies the question, but please do not hesitate to let us know whether we should elaborate on this any further.
>
> > It seems very odd that KFAC performs better if the random mini-batch are not sampled (new) at every step. Have you noticed any such thing for KFAC on standard problems (not PINN losses)? Are the matrices in KFAC inverted every iteration or also just every T-th iteration?
>
> We are not aware of any findings in the literature that have observed something like this. To make our comparison fair, we allowed the number of iterations a mini-batch is used as a hyper-parameter on which we performed Bayesian optimization. We found that for all optimizers it was favorable to keep a mini-batch for significant number of iterations. Thus, we regard this finding as something specific to the PINN problem rather than to KFAC. Our methods update the Kronecker matrices and inverses at every step.
> ___
> Thanks once more for your constructive comments. We hope our responses addressed them to your satisfaction. We remain attentive to your feedback!

---

> > ### Author Response · Authors · 2024-08-09
> > **Additional experiment with error bars**
> >
> > Dear Reviewer aiiA,
> >
> > We are happy to inform you that we can report the results of an additional experiment testing the optimizers for a variety of initializations. For this, we took the optimized hyper-parameters and run the different optimizers for 10 different initializations and mini-batches for the 4+1d heat equation for the MLP with 10 065 parameters (Figure 2, middle plot). Unfortunately, we are not allowed to share figures during the discussion period. The table below summarizes variability of final performance. We will add the corresponding figure to the appendix.
> >
> > We find that all optimizers except LBFGS perform stably when using different data/model initialization. When taking the average performance, the same trend as shown in the original submission is visible where KFAC* outperforms the competing optimizers. Please let us know in case that you have any further questions.
> >
> >
> > | Optimizer | Relative L2 error |
> > |-----------|--------------------|
> > | SGD |  $(6.77\pm2.36)\cdot 10^{-3}$ |
> > | Adam | $(3.10\pm2.45)\cdot 10^{-3}$ |
> > | Hessian-free | $(1.97\pm0.47)\cdot 10^{-5}$ |
> > | ENGD (full) | $(4.99\pm9.25)\cdot 10^{-2}$ |
> > | ENGD (layer-wise) | $(1.68\pm0.26)\cdot 10^{-4}$ |
> > | KFAC | $(6.50\pm0.84)\cdot 10^{-5}$ |
> > | KFAC* | $(1.46\pm0.48)\cdot 10^{-5}$ |
> >
> > Let us know if you would like to see similar results for other sub-experiments.

---

> ### Comment · Reviewer_aiiA · 2024-08-11
> **Answer to Rebuttal**
>
> Thank you for your response and for answering my questions.
>
> > We respectfully disagree. We report relative L2 error rather than training loss [...]
>
> Thank you for the clarification, this was a misunderstanding on my side. This is indeed mentioned in Sec. 4.
>
> > It is true that our plots show the performance of a single run. However, this run was obtained through extensive hyper-parameter tuning [...]
>
> I was about to suggest running 10 different seeds and data splits, which is what you have reported in the comment below.
>
> > High EMA factors correspond to slowly forgetting pre-conditioners [...]
>
> Thanks, this makes sense, of course, it was a misunderstanding on my side.
>
> > We found that for all optimizers it was favorable to keep a mini-batch for significant number of iterations.
>
> This is quite interesting and weird at the same time. If you find the time, an ablation for this would be quite interesting.

---

### Official Review · Reviewer_YHUU · 2024-07-11

**Soundness:** 3
**Presentation:** 2
**Contribution:** 3
**Rating:** 6
**Confidence:** 4

**Summary:**

This paper generalizes the K-FAC method (which is well-known in optimization for deep learning) to enable it to train PINNs. The key idea is to combine Taylor-mode autodifferentiation with recent work on K-FAC with weight sharing layers. The proposed method is evaluated on several PDEs, where it outperforms first-order methods like SGD and Adam and quasi-Newton methods like L-BFGS, while performing comparably to (matrix-free) ENGD.

**Strengths:**

* The paper makes a novel contribution by extending K-FAC to PINNs, which had not been previously done in the literature.

* The paper does a good job of showing how the Kronecker-factored preconditioner can be extended to the residual portion of the PINN loss. I really appreciated the example derivations in sections 3.2 and 3.3!

* The paper empirically evaluates their methods against other second-order/quasi-Newton methods used in training PINNs, such as ENGD, matrix-free ENGD, and L-BFGS. The comparison appears to be fair, since all methods receive equal amounts of hyperparameter tuning.

**Weaknesses:**

* The paper is missing more recent work on difficulties of training PINNs, such as [1, 2].

* K-FAC/K-FAC* do not reach the lowest error obtained by running ENGD with $D = 449$ (Figure 2). Why is this the case?

* I don’t think the PDE settings tested in this paper are particularly challenging. For example, the 2d Poisson equation and (4+1)d heat equation have solutions that have relatively low frequencies, but PINNs are known to struggle more with learning solutions containing high frequencies [3, 4]. How would K-FAC/K-FAC* fair in more challenging settings [5, 6]?

* The approach for computing the Kronecker-factored approximation uses “a larger net with shared weights”. Could this cause out-of-memory issues with larger networks? How much additional memory is required?

* I think notation is being reused too much in equations (5) and (6). For example, I don’t think it’s a good idea to have $z^{(l)} = W^{(l)} z^{(l - 1)}$ and $z^{(l)} = \sigma(z^{(l - 1)})$.

* I’d recommend defining the symbol $\odot$ as the Hadamard product before it appears in equation (6).

* Equation below line 168: $N_\Omega$ in the sum should be $N$.

* I believe the outer product decomposition in line 200 is missing a transpose on the second $l_{n, m}$ in the summation.

* There is no per-step computational complexity provided for the proposed methods.

* No open-source implementation is provided for the proposed methods.

Minor comments:

* The sentence containing equation (3) is a bit confusing. Saying “$u_n = u_\theta(x_n)$, and it coincides with the classical Gauss-Newton matrix” makes it sound like $u_n$ itself is the classical Gauss-Newton matrix (although I can see that this is not what the authors meant).

* For consistency with existing literature, I would recommend writing “KFAC” as “K-FAC” and “LBFGS” as “L-BFGS”.

[1] “On the Role of Fixed Points of Dynamical Systems in Training Physics-Informed Neural Networks.” Rohrhofer et al. (2023)

[2] “Challenges in Training PINNs: A Loss Landscape Perspective.” Rathore et al. (2024)

[3] “On the eigenvector bias of Fourier feature networks: From regression to solving multi-scale PDEs with physics-informed neural networks.” Wang et al. (2021)

[4] “When and why PINNs fail to train: A neural tangent kernel perspective.” Wang et al. (2022)

[5] “PINNacle: A Comprehensive Benchmark of Physics-Informed Neural Networks for Solving PDEs.” Hao et al. (2023)

[6] “PDEBENCH: An Extensive Benchmark for Scientific Machine Learning.” Takamoto et al. (2022)

**Questions:**

* Line 64: Could the authors please elaborate on why the preconditioner not having a Laplacian term is beneficial?

* Line 131: Could the authors please elaborate on why the interior Gramian cannot be approximated with the existing K-FAC? Is this due to computational and/or autodiff limitations?

* Is using a general loss $\ell$ in equation (11) necessary? For PINNs, $\ell$ is almost always the least-squares loss, but I could also understand if the authors wanted to show that their method works in a more general setting.

* Does the proposed approach generalize to inverse problems involving PINNs? If it does generalize, how does it affect the calculation of the Kronecker-factored approximation?

* Are there operators besides the Laplacian for which the weight sharing can be reduced?

**Limitations:**

Please see "Weaknesses"

---

> ### Author Rebuttal · Authors · 2024-08-06
>
> Dear Reviewer YHUU,
>
> we would like to thank you for the time and effort you put into reviewing our work. In the following, address some of the points that were rightfully raised by you.
>
> > How much additional memory is required [for our KFAC]?
> > There is no per-step computational complexity provided
>
> In our general response, we compare memory requirements and computation time of the proposed scheme. We find that the viewpoint of 'a larger net with shared weights' alias the forward Laplacian graph saves both memory and compute time when compared to the standard approach relying on nested backpropagation, as implemented in `functorch`. If there are any further questions, please do not hesitate to contact us.
>
> > No open-source implementation is provided
>
> We will release the code after acceptance with the camera ready version.
>
> > K-FAC/K-FAC* do not reach the lowest error obtained by running ENGD [...]. Why is this the case?
>
> As KFAC is designed as an approximation of ENGD we do not expect KFAC to provide results superior to ENGD. Note however that ENGD requires the solution of a system of linear equations in the parameter space of the network. Hence, while we can use it as baseline for a small network, it does not scale to larger nets. Our updated discussion of Figure 2 elaborates on this.
>
> > How would K-FAC/K-FAC* fair in more challenging settings [5, 6]?
>
> We believe that PINNs should be employed for high-dimensional PDE problems where classical methods are intractable and are currently working on the log-Fokker-Planck equation which is a challenging nonlinear PDE. We aim to present first results in the discussion period and refined results for a camera-ready version.
>
> > The paper is missing more recent work on difficulties of training PINNs, such as [1, 2]
>
> Thank you for the pointers to these references, which we have included into our introduction.
>
> **Notation/typos**
>
> > I think notation is being reused too much in equations (5) and (6)
>
> Thank you for this feedback. We are currently looking into ways to streamline and improve our notation for the camera-ready version.
>
> > I’d recommend defining the symbol as the Hadamard product before
>
> Thanks for the pointer, we have added the definition.
>
> > Equation below line 168
>
> Thanks for spotting this, indeed, we have been slightly inconsistent in carrying the subscript $N_\Omega$ for the number of interior integration points. We have adjusted this and have made our notation consistent.
>
> > the outer product decomposition in line 200 is missing a transpose
>
> Thanks for catching this typo, which we have corrected.
>
> **Minor comments:**
>
> > The sentence containing equation (3) is a bit confusing.
>
> Thanks for the feedback. To our knowledge the matrix $J_\theta^\top u J_\theta u$ is usually called Gauß-Newton matrix, where $J$ is the Jacobian with respect to the model parameters. This is exactly the form of the matrix defined in (3) if $u = (u_1, \dots, u_n)^\top$. Please let us know, whether you have any further questions and we are willing to adjust our wording to reduce ambiguities.
>
> > I would recommend writing “K-FAC” and “L-BFGS”.
>
> Thank you for the feedback, of course we are willing to follow the common nomenclature.
>
> **Questions**
>
> > Line 64: Could the authors please elaborate on why the preconditioner not having a Laplacian term is beneficial?
>
> We are deeply convinced that when optimizing a PINN, preconditioners with a PDE term, e.g., a Laplacian, are much more meaningful than preconditioners without a PDE term. In line 64 we mention a line of works that use KFAC for PINN, but only to approximate a preconditioner that does not include PDE terms. Hence, we regard these approaches as not satisfactory which motivated us to develop a KFAC method for preconditioners including PDE terms. We have slightly changed our wording in line 64 to improve clarity.
>
> > Line 131: Could the authors please elaborate on why the interior Gramian cannot be approximated with the existing K-FAC? Is this due to computational and/or autodiff limitations?
>
> K-FAC has only been proposed when the loss and therefore the pre-conditioner only incorporates function evaluations. This excludes PDE terms in the pre-conditioner, which appear naturally when incorporating second-order information in PINNs. Hence, the reason existing KFAC approaches can't be used is not a computational limitation but quite simply the lack of a KFAC for problems including PDE terms.
>
> > Is using a general loss in equation (11) necessary?
>
> We decided to work with a general loss as losses other than the least-square problem appear in practice. This is for example the case for variational approaches including the [*deep Ritz method*](https://link.springer.com/article/10.1007/s40304-018-0127-z) and [*consistent PINNs*](https://arxiv.org/pdf/2406.09217) which  use Lp-norms. Of course, there is a trade-off between generality and accessibility. We decided to provide the general version as we already provide a didactic specific example with the Poisson equation and as we found the general form to be not much more complex. If you strongly disagree, we can also offer a simplified version in the main body and defer the fully general case to the appendix.
>
> > Does the proposed approach generalize to inverse problems involving PINNs?
>
> We have not yet considered inverse problems. In general, we see no conceptual problem, however, one needs to take into account two (or more) neural networks when dealing with inverse problems. We leave this for future work.
>
> > Are there operators besides the Laplacian for which the weight sharing can be reduced?
>
> A similar reduction can be obtained for linear scalar valued PDE operators, meaning whenever the PDE of consideration is a linear function of the partial derivatives of some order.
> ___
> Thanks once more for your comments that have helped use improving our manuscript and we hope our responses could address your comments to your satisfaction. We remain attentive to your feedback!

---

> > ### Comment · Reviewer_YHUU · 2024-08-11
> > **Thanks for the rebuttal**
> >
> > Thank you for addressing most of my comments. I would have liked to see the results on the log-Fokker-Planck equation, but it is ok if this is not available in time (I understand the discussion period is rather short). I will raise my score accordingly.
> >
> > Minor question (does not affect my score): why would solving an inverse problem with this approach require two (or more) neural networks? I thought the standard formulation for the PINN inverse problem uses a single neural network with the inverse coefficients as inputs.

---

> > > ### Author Response · Authors · 2024-08-14
> > >
> > > Thank you for your question. Assume we are given an inverse problem (let's say a source recovery problem for simplicity) where we are looking to find both the function $u$ and the right-hand side/source $f$ given some observations of $u$ and a PDE that $u$ should satisfy. In this case, the loss function contains a term for the PDE residual and boundary values, and additionally a data fidelity term that incorporates the observations of $u$. A canonical approach would be to parametrize both $u$ and $f$ by neural networks, which explains the two neural networks in our answer above.

---

### Official Review · Reviewer_TQnM · 2024-07-15

**Soundness:** 3
**Presentation:** 3
**Contribution:** 3
**Rating:** 6
**Confidence:** 4

**Summary:**

This work considers the problem of optimising partial differential equations (PDE) with neural networks, in particular second-order optimization.
Even if simple models (multi-layer perceptrons) are used, for which KFAC approximations of the curvature matrix are well-known, it needs to derive new approximations because the loss function used for solving PDEs is different than previous work (square loss or cross entropy), in particular it includes derivatives of the model with respect to the input, which is interpreted as a larger model with weight sharing. Therefore, it uses the previously (but recently) developed technique of KFAC with weight sharing. Experiments show that the new method is successful when applied to a few example PDEs.

**Strengths:**

The paper is clearly written and seems correct, both the theory and experiment. It seems to be the first application of KFAC to PDEs and has the potential of a high impact.

**Weaknesses:**

Even if experiments with a relatively large number of parameters are shown (10^5), the paper does not provide the complexity of the algorithm, in terms of relevant quantities, for example the input dimension, output dimension, batch size, number of parameters, etc. For example, does the method scale poorly with the input dimension? I would expect quadratic scaling with input dimension. I have not read the forward laplacian paper, but I have the feeling that improves scaling by only a constant factor. https://arxiv.org/abs/1206.6464 argued that it may not exist an algorithm for computing the diagonal of the Hessian that is linear in the dimension using automatic differentiation. Also, as with other KFAC approximations, it remains unclear how general is the approach, e.g. how to apply it to higher order PDEs or other neural network models.

**Questions:**

NA

---

> ### Author Rebuttal · Authors · 2024-08-06
>
> Dear Reviewer TQnM,
>
> we would like to thank you for the time and effort you put into reviewing our work. In the following, we want to address some of the points that you rightfully raised.
>
> ---
>
> > paper does not provide the complexity of the algorithm
>
> > does the method scale poorly with the input dimension? I would expect quadratic scaling with input dimension.
>
> You are right that we currently do not discuss this; we address this in detail in our global rebuttal and will add a discussion to the main text.
>
> The quick summary is that empirically we observe linear run time and memory scaling in the input dimension for computing the Laplacian either via `functorch`'s nested first-order autodiff or the forward Laplacian framework. This linear scaling is intuitive from the forward Laplacian perspective, which performs `input_dimension + 2` forward passes of the net (assuming the network's forward pass is independent of the input dimension, which is a good approximation for the MLP we used, but in general will depend on the architecture). In Landau notation, the scaling with the input dimension will depend on the choice of architecture.
>
> Moreover, we observe some practical advantages of the forward Laplacian over the nested first-order autodiff approach:
> - The constants in the scaling of the forward Laplacian are smaller than those in the `functorch` implementation. The forward Laplacian is roughly 2x faster. We think this is because its computation graph is less sequential. Also, we observe that the forward Laplacian uses roughly half the memory. These constants matter in practise.
> - Backpropagation on the forward Laplacian graph (which we need to compute the gradient of the loss function) allows us to populate the K-FAC matrices without much re-computation.
>
> Overall, it seems that the forward Laplacian is the current state-of-the-art method to compute input Laplacians, comes with advantages over nested first-order autodiff, and integrates nicely with our proposed K-FAC approximation.
>
> ---
>
> > it remains unclear how general is the approach, e.g. how to apply it to higher order PDEs or other neural network models.
>
> Our approach is general:
>
> - The proposed K-FAC approximation generalizes to arbitrary PDEs, both in the PINN formulation (i.e. least squares of the strong form) and for variational formulations (deep Ritz). We discuss this in Section 3.3, see also equation (14).
> - The Taylor-mode AD needed to assemble the K-FAC matrices works for general neural network architectures. For instance, JAX offers a general purpose Taylor-mode implementation ([`jax.experimental.jet`](https://jax.readthedocs.io/en/latest/jax.experimental.jet.html)) onto which our proposed K-FAC framework could be built on.
> - Like the traditional KFAC, our KFAC approximation is architecture-dependent. However, thanks to the recent formulation of KFAC for linear layers with weight sharing [2], a wide range of architectures is covered, e.g. fully-connected/attention/convolution layers.
>
> ---
>
> Thanks again for helping us to improve our manuscript! We hope our responses address your comments. Please let us know if you have follow-up questions. We would be happy to discuss.
>
> ## References
>
> [2] Eschenhagen, R., Immer, A., Turner, R. E., Schneider, F., & Hennig, P. (NeurIPS 2023). Kronecker-factored approximate curvature for modern neural network architectures.

---

> > ### Comment · Reviewer_TQnM · 2024-08-09
> >
> > Thank you for the response.
> > How is the forward pass independent on the input dimension?
> > In an MLP, you must compute a matrix-vector product, where the input vector has size d, and the matrix has size d' x d (hidden layer size d'), this has complexity o(d'd), linear in d. Maybe d is not your bottleneck now, but if you increase it at some point you should suffer from it.
> > I'm asking also because I have been working on score matching, which requires computing the trace of the Hessian of the log-likelohood, different setting but same computational problem (see for example https://arxiv.org/abs/1905.07088). That is impossible to scale up when d is in the range of millions or higher.

---

> ### Author Response · Authors · 2024-08-09
>
> Thanks for your follow-up question!
>
> You are of course right that the MLP's cost depends linearly on $d_\Omega$ due to the first layer's matrix multiply ($d_\Omega \to 768$). For the values of $d_\Omega$ we experimented with, the evaluation cost is dominated by the sub-sequent matrix multiplies (e.g. $d_\Omega \le 100$ but the second layer is $768 \to 768$). Therefore we empirically observe linear scaling in this regime, i.e. 'approximately' no dependency of the MLP w.r.t. $d_\Omega$ and linear scaling from the number of forward passes for the Laplacian.
>
> You are completely right though that if we let $d_\Omega \to \infty$, e.g. for score matching with $d_\Omega \sim 10^6$, the empirical behavior would eventually become quadratic on this model. However, for the PDE applications we target in this paper, we think that practically relevant values are $d_\Omega \sim 10^2$ (at most $10^3$). Therefore, we believe the empirical linear scaling we measured in this regime is representative for our method's scaling on PINN problems.
>
> We will add a sentence to clarify this limitation for extremely high dimensions of $d_\Omega$ which are (1) out of scope for PINNs and (2) for which the autodiff-based approach suffers from the same limitation.
>
> Please let us know if you have any follow-up question.

---

### Author Rebuttal · Authors · 2024-08-06

We want to thank all reviewers for their thorough evaluation of our submission and provide extensive answers to the individual points raised by the reviewers below.

Here, we want to elaborate on our method's per-iteration complexity, both theoretically and empirically, as this concern was raised by multiple reviewers. We will add the following discussion to the main text.

On top of evaluating the loss and computing the gradient, there are two parts of KFAC that require additional resources; the assembly and inversion of the Kronecker factors:

1. **Inversion of Kronecker factors:** Inverting layer $l$'s Kronecker approximation of the Gramian requires $O(h_l^3+h_{l+1}^3)$ time and $O(h_l^2+h_{l+1}^2)$ storage, where $h_l$ is the number of neurons in the $l$-th layer.  Note that inverting the exact block for layer $l$ (as done by 'ENGD (layer-wise)') requires $O(h_l^3h_{l+1}^3)$ time and $O(h_l^2 h_{l+1}^2)$ memory. In general, the improvement from the Kronecker factorization depends on how close to square the weight matrices of a layer are, and therefore on the architecture. In practise, the Kronecker factorization usually significantly reduces memory and run time.
Further improvements can be achieved by using structured Kronecker factors, e.g. diagonal or block-diagonal matrices as proposed in [1], which are beyond the scope of our paper.

2. **Assembly of Kronecker factors:** To compute the Kronecker factors, we use the layer inputs from evaluating the loss (for free), and need to backpropagate through its graph. The loss is computed with the forward Laplacian framework, which is the state-of-the-art method to compute input Laplacians of neural networks. The additional backpropagation is similar to the one performed by SGD with the only difference that we differentiate w.r.t. layer outputs rather than parameters.

We want to stress that evaluating the Laplacian and parameter derivatives is essential for *all* gradient-based PINN optimizers, not only for KFAC. To provide concrete insights into the computation time and memory of this routine, we ran an additional experiment and compared the forward Laplacian we use with the alternative `functorch` implementation.

**We find that the forward Laplacian is roughly 2x faster and uses roughly half the memory compared to PyTorch's automatic differentiation. Further, the computation time for both methods scales linearly in the input dimension in our experiments.**

We provide a detailed description of our findings below and in the attached PDF and will add it to the updated version.

**Conclusion**

Thanks again for your helpful comments and questions. We are convinced that they have helped us to substantially strengthen the manuscript and hope we addressed them to your satisfaction.

We remain attentive to your feedback!

## Experimental details

Due to the limited length of the discussion period, we present a short experiment. We will add a more detailed comparison to the manuscript.

**Setup:** We consider two Laplacian implementations:

1. **Autodiff Laplacian:** Computes the Laplacian with PyTorch's automatic differentiation (`functorch`) by computing the batched Hessian trace (via `torch.func.hessian` and `torch.func.vmap`). This is the standard approach in many PINN implementations.

2. **Forward Laplacian:** Computes the Laplacian via the forward Laplacian framework. We used this approach for all PDEs and optimizers presented in the experiments.

We use the biggest network from our experiments (the $D_\Omega \to 768 \to 768 \to 512 \to 512 \to 1$ MLP with tanh-activations from Fig. 3 right), then measure run time and peak memory consumption of computing the net's Laplacian on a mini-batch of size $N=1024$ with varying values of $D_\Omega$. To reduce measurement noise, we repeat each run over five independent Python sessions and report the smallest value. We use the same GPU as all other experiments, i.e. an NVIDIA RTX 6000 with 24 GiB memory.

**Results:** The following tables compare run time and peak memory between the two approaches:

| $D_\Omega$ | Autodiff Laplacian [s] | Forward Laplacian [s] |
|-----------|--------------------|-------------------|
| 1 | 0.051 (1.6x) | 0.033 (1.0x) |
| 10 | 0.20 (2.0x) | 0.10 (1.0x) |
| 100 | 1.7 (2.0x) | 0.84 (1.0x) |

We observe that the forward Laplacian is roughly twice as fast as the `functorch` Laplacian.

| $D_\Omega$ | Autodiff Laplacian [GiB] | Forward Laplacian [GiB] |
|-----------|--------------------|-------------------|
| 1 | 0.21 (0.96x) | 0.22 (1.0x) |
| 10 | 0.98 (1.6x) | 0.61 (1.0x) |
| 100 | 8.8 (1.9x) | 4.6 (1.0x) |

We observe that the forward Laplacian uses significantly less memory for large input dimensions, up to only one half when $D_\Omega = 100$.

We visualized both tables using more values for $D_\Omega$ and observed linear scaling in both memory and run time; see the attached PDF.

**Other scalings:**
- Both the forward Laplacian and nested backpropagation scale linearly w.r.t. batch size.
- Scaling w.r.t. output dimension (as asked by TQnM) is less relevant as many high-dimensional PDE problems are scalar-valued.
- Scaling w.r.t. number of trainable parameters is highly architecture-dependent. This is also true for simple backpropagation in standard deep learning applications and we believe this question needs to be answered on a case-by-case basis.

## References

[1] Lin, W., Dangel, F., Eschenhagen, R., Neklyudov, K., Kristiadi, A., Turner, R. E., & Makhzani, A. (ICML 2024). Structured inverse-free natural gradient descent: memory-efficient & numerically-stable KFAC.

---

### Decision · Program_Chairs · 2024-09-25

**Decision:**

Accept (poster)

**Comment:**

All three reviewers agree this is a nice contribution to the literature of optimization for PINNs. The noted strengths are

* novel extension of KFAC to PINNs which leverages nontrivial forward-mode autodiff techniques.
* rigorous and clear derivations
* empirical results demonstrating advantages over SGD, Adam, and L-BFGS

There are a few weaknesses such as computational complexity and robustness of experiments that are well addressed in rebuttal. Therefore, I'm recommending acceptance.